# Quantile Reward Policy Optimization: Alignment with Pointwise Regression and Exact Partition Functions

**Simon Matrenok**[*]
CLAIRE, EPFL

**Skander Moalla**[*]
CLAIRE, EPFL

**Caglar Gulcehre**
CLAIRE, EPFL

## Abstract

Aligning large language models with pointwise absolute rewards has so far required online, on-policy algorithms such as PPO and GRPO. In contrast, simpler methods that can leverage offline or off-policy data, such as DPO and REBEL, are limited to learning from preference pairs or relative signals. To bridge this gap, we introduce *Quantile Reward Policy Optimization* (QRPO), which learns from pointwise absolute rewards while preserving the simplicity and offline applicability of DPO-like methods. QRPO uses quantile rewards to enable regression to the closed-form solution of the KL-regularized RL objective. This reward yields an analytically tractable partition function, removing the need for relative signals to cancel this term. Moreover, QRPO scales with increased compute to estimate quantile rewards, opening a new dimension for pre-computation scaling. Empirically, QRPO consistently achieves top performance on chat and coding evaluations—reward model scores, AlpacaEval 2, and LeetCode—compared to DPO, REBEL, and SimPO across diverse datasets and 8B-scale models. Finally, we find that training with robust rewards instead of converting them to preferences induces less length bias.

**Table 1:** Policy fitting methods are popular for their **simplicity** and ability to work **offline**, but **cannot** fully use the signal from **pointwise absolute rewards** (e.g., strong reward models, verifiable rewards); they learn from **relative rewards** (preferences or reward differences). **QRPO fills this important gap.**

|  | Offline (& Online) | Pointwise Rewards |
|---|:---:|:---:|
| *Policy Improvement:* PPO, GRPO, RLOO, etc. | ✗ | ✓ |
| *Policy Fitting:* DPO, REBEL, SimPO, etc. | ✓ | ✗ |
| *Policy Fitting:* **QRPO (Ours)** | ✅ | ✅ |

**Figure 1:** QRPO outperforms policy fitting methods that learn from relative signals in downstream general chat tasks.

**Table 2:** QRPO is more suitable for code generation tasks such as LeetCode, where the reward is canonically pointwise, expressing the test-case pass rate.

| Method | Avg. Pass (%)↑ |
|---|:---:|
| Llama 8B Tülu 3 SFT | $20.9 \pm 0.5$ |
| + SFT on Correct | $18.8 \pm 1.1$ |
| + REBEL | $26.1 \pm 1.8$ |
| + DPO | $30.2 \pm 1.4$ |
| + SimPO | $22.3 \pm 1.4$ |
| + QRPO (Ours) | $\mathbf{32.7} \pm 1.0$ |

---

[*]Shared first authorship and equal contributions. Correspondence to `skander.moalla@epfl.ch`.

39th Conference on Neural Information Processing Systems (NeurIPS 2025).

**Figure 2: QRPO uses quantile rewards**, which makes the **exact expression** of the **partition function** $Z$ **tractable** and allows fitting the solution of the KL-regularized objective with a **pointwise regression**, i.e., using a **single sample with its reward instead of a pair of preferences.** In the pre-computation phase, we generate reference completions from the reference model and compute their rewards. For training, we then use these reference rewards to compute the quantile reward of training samples. In its simplest form, QRPO optimizes the quantile reward; however, **a family of transformations can be applied on top of the quantile reward to recover a desired reward shape**, while still having a tractable exact expression for $Z$ (see Table 4).

## 1 Introduction

Alignment methods are highly effective in fine-tuning large language models (LLMs), improving their conversational abilities (Ouyang et al., 2022; Stiennon et al., 2020; Rafailov et al., 2023), safety (Bai et al., 2022), and reasoning capabilities (Lambert et al., 2024; Shao et al., 2024a; Guo et al., 2025). They are applied as the final stage of the LLM training pipeline after pre-training the LLM on a large corpus of data (Brown et al., 2020) and fine-tuning the LLM to follow instructions (Ouyang et al., 2022). However, while simple and efficient methods, such as DPO (Rafailov et al., 2023), relying on a *preference reward signal*, achieve strong performance in conversational tasks, more complex and computationally intensive methods, such as GRPO (Shao et al., 2024b), have been the only alternative to effectively optimize a *pointwise reward signal*.

Alignment methods are often referred to by different names, such as reinforcement learning from human feedback (RLHF) (Ouyang et al., 2022), reinforcement learning from verifiable rewards (RLVR) (Lambert et al., 2024), and direct alignment algorithms (DAA) (Rafailov et al., 2024), but the underlying algorithms in most of these methods optimize a common objective from regularized reinforcement learning (RL) (Jaques et al., 2017). This nomenclature distinguishes two dimensions: whether the reward is optimized explicitly (RLHF, RLVR) or implicitly (DAA) and whether it is modeled from (human) preference feedback (RLHF, DAA) or objectively defined (RLVR). Yet, another crucial dimension is the data regime required by the underlying algorithm, which significantly impacts the computational cost of the methods. *Policy improvement (PI) methods*, such as PPO (Ouyang et al., 2022; Schulman et al., 2017), GRPO (Shao et al., 2024b), and RLOO (Ahmadian et al., 2024), require online sampling as they perform online policy improvement, thus increasing the complexity and cost of distributing large networks across training and inference engines. In contrast, *policy fitting (PF) methods*, such as DPO (Rafailov et al., 2023) and REBEL (Gao et al., 2024), can leverage any data distribution because they aim to fit a closed-form solution of the regularized RL objective valid for all data, giving them the reputation for being simpler and more stable, as usually used with offline data.[2]

Existing PF methods rely on *relative rewards*, i.e., differences between rewards, to tractably fit the closed-form solution, as estimating it for absolute rewards is known to be intractable due to the difficulty of estimating the partition function. This feature has contributed to their popularity, as human feedback can be expressed as preferences and modeled as relative rewards by preference models. However, relative rewards can provide a suboptimal signal, for example, when both the chosen and rejected completions are better than the training model. This leads to conflicting dynamics with updates from samples that could both improve the model. Moreover, human feedback can be more easily collected as absolute feedback in the form of a rating scale. More pressingly, with the emergence of powerful absolute reward models for human alignment trained with regression

---

[2]In the LLM literature, the terms are "RL" methods and "DAA" methods, but we feel that this is not accurate as all methods optimize an RL objective, and methods like REBEL do not fit within the "direct" naming which refers to directly using a preference model, which is not necessary to target the closed-form solution optimization.

such as ArmoRM (Wang et al., 2024) and Nemotron-Reward (Wang et al., 2025) and successful absolute-reward applications such as verifiable rewards (Lambert et al., 2024), PF methods are losing applicability and practitioners have to resort to complex PI methods to use those rewards.

To overcome this limitation, we propose *Quantile Regression Policy Optimization (QRPO)*. This new RL fine-tuning algorithm retains the simplicity and benefits of policy fitting methods while enabling the use of absolute rewards to overcome their limitations and broaden their applicability. QRPO is theoretically sound, simple, and effective at scale. We make the following contributions:

1. **QRPO addresses the problem of estimating the partition function in policy fitting methods** QRPO is able to fit the closed-form optimal policy of the KL-regularized RL objective for individual points with a pointwise absolute reward signal using a simple supervised regression. The key insight to making this possible is using a reward distribution for which the exact expression of the partition function is tractable. This optimization strategy has not been widely explored due to the common concern that partition functions are generally intractable. QRPO relies on quantile rewards and their distribution to derive this expression. This gives a partition function equal to $\beta(\exp(1/\beta) - 1)$. Furthermore, we frame QRPO as a framework in which applying additional transformations on top of the quantile reward still yields tractable partition functions.

2. **QRPO scales with a larger pre-compute budget to estimate quantile rewards** We demonstrate that generating more reference rewards to estimate quantile rewards improves performance.

3. **QRPO consistently achieves top performance in conversational and coding tasks** We show through an extensive and fair experimental protocol that QRPO consistently obtains the best rewards and AlpacaEval (Li et al., 2023) scores. We compared our method to DPO, REBEL, and SimPO, where each was trained in more than 200 settings of models, datasets, hyperparameters, and distribution shifts, including Llama 8B and Mistral 7B as models and Magpie-Air (Xu et al., 2024b) and UltraFeedback (Cui et al., 2024) as datasets. We also demonstrate that QRPO is effective in a LeetCode (Xia et al., 2025) coding task, where the reward is canonically pointwise, expressing the test-case pass rate and loses signal when expressed as a preference.

4. **Methods that directly use a robust reward model exhibit less length bias than methods that have to convert this reward to preferences** We show that SimPO, which tries to overcome length bias by incorporating an inductive bias using length normalization, reduces absolute length but still shows strongly length-biased policies like DPO, while QRPO and REBEL do not.

## 2 Preliminaries

**RL fine-tuning for LLM Alignment** The underlying RL fine-tuning objective in alignment methods is formulated as (Jaques et al., 2017; Stiennon et al., 2020; Rafailov et al., 2023)

$$\max_{\pi_\theta} \mathbb{E}_{x \sim \mathcal{D}} \left[ \mathbb{E}_{y \sim \pi_\theta(\cdot|x)} \left[ \mathcal{R}(x, y) \right] - \beta \mathbb{D}_{KL}(\pi_\theta(\cdot \mid x) \| \pi_{ref}(\cdot \mid x)) \right]. \tag{1}$$

The goal is to maximize the expected reward $\mathcal{R}$ of $\pi_\theta$ over prompts sampled from a distribution of user queries $x \sim D$, while enforcing a constraint to keep $\pi_\theta$ close to $\pi_{ref}$, the instruction fine-tuned pre-trained LLM, using a KL divergence term, weighted by $\beta$. $\pi_\theta$ is initialized from $\pi_{ref}$, and the regularization prevents the model from over-optimizing the reward function by ensuring it stays close to the original distribution, maintaining fluency and diversity.

**Policy improvement (PI) methods** PPO (Schulman et al., 2017) and their variants rewrite the RL fine-tuning objective by moving the KL penalty to the reward (Stiennon et al., 2020, PPO), or consider it as an auxiliary loss (Shao et al., 2024b, GRPO), to then effectively optimize a standard RL objective by taking policy gradient steps to *improve the policy online*. Hence, these methods require sampling before each gradient step, requiring extra complexity to host inference instances of the policy, and making them more costly in terms of time and computational resources.

**Policy fitting (PF) methods** Objective 1 admits a closed-form solution, expressed as

$$\pi^*(y \mid x) = \frac{1}{Z(x)} \pi_{ref}(y \mid x) \exp \left( \frac{1}{\beta} \mathcal{R}(x, y) \right); \quad Z(x) = \sum_y \pi_{ref}(y \mid x) \exp \left( \frac{1}{\beta} \mathcal{R}(x, y) \right). \tag{2}$$

*Policy fitting* methods leverage this expression to *fit the optimal policy*. By substituting the optimal policy $\pi^*$ with a parameterized model $\pi_\theta$ in Equation 2, we can formulate a distribution matching or

regression task. However, estimating $\pi^*$ directly is known to be intractable due to the difficulty in estimating the partition function $Z$ (Rafailov et al., 2023). Hence, current methods resort to relative signals between two completions to eliminate the absolute partition function. Indeed, we have:

$$\mathcal{R}(x, y) = \beta \log \frac{\pi^*(y \mid x)}{\pi_{ref}(y \mid x)} + \beta \log Z(x), \tag{3}$$

and we see that the difference in rewards between two completions gives a relative expression between two policies that is independent of the partition function:

$$\mathcal{R}(x, y^+) - \mathcal{R}(x, y^-) = \beta \left( \log \frac{\pi^*(y^+ \mid x)}{\pi_{ref}(y^+ \mid x)} - \log \frac{\pi^*(y^- \mid x)}{\pi_{ref}(y^- \mid x)} \right). \tag{4}$$

When the reward is used to parameterize a preference model, such as the Bradley-Terry model (Bradley & Terry, 1952), this difference gives the preference between two completions $p(y^+ \succ y^-) = \sigma(\mathcal{R}(x, y^+) - \mathcal{R}(x, y^-))$, and direct preference optimization methods such as DPO (Rafailov et al., 2023) learn the policy by substituting the reward difference by the policy difference and fitting the preferences. Gao et al. (2024) propose REBEL, which fits the optimal policy by directly regressing the difference in $\log$-probability ratios to the difference in rewards (as shown in Equation 4). The REBEL algorithm was originally presented as an iterative method; however, in this work, we focus on the RL fine-tuning objective with a fixed reference policy, and as the first iteration of REBEL already fits the optimal policy relative to $\pi_{ref}$ we present REBEL as a policy fitting method.

Fitting a relative signal from two completions of a policy still remains a significant limitation, as two completions could both be better or worse than what the current model outputs, causing conflicting dynamics through updates to the policy to both increase and decrease the likelihood of two good or bad completions. The relative signal also naturally degrades the absolute information present in a single sample. In what follows, we present a novel algorithm that enables policy fitting with an absolute reward signal, addressing the abovementioned limitations.

## 3 Quantile Regression Policy Optimization (QRPO)

**A canonical objective**   We start by rearranging Equation 2 as

$$\mathcal{R}(x, y) - \beta \log Z(x) = \beta \log \frac{\pi^*(y \mid x)}{\pi_{ref}(y \mid x)}. \tag{5}$$

By substituting the optimal policy $\pi^*$ with a parameterized model $\pi_\theta$, we obtain the following regression objective for the RL fine-tuning task:

$$\min_{\pi_\theta} \mathbb{E}_{x,y} \left[ \left( \underbrace{\mathcal{R}(x, y) - \beta \log Z(x)}_{\text{calibrated target}} - \beta \log \frac{\pi_\theta(y \mid x)}{\pi_{ref}(y \mid x)} \right)^2 \right], \tag{6}$$

We choose the mean-squared error (MSE) as a loss, which is also a common choice in other LLM alignment algorithms (Azar et al., 2024). Moreover, as a consequence of the central limit theorem, the noise at the target value is likely to have a Gaussian distribution, which makes the MSE appropriate.

The objective in Equation 6 is the most straightforward regression objective to derive from Equation 2 and is aimed at by previous work (Gao et al., 2024; Pierre Harvey Richemond et al., 2024), but is still intractable due to the difficulty in estimating the partition function $Z$, leading to these previous attempts to resort to approximations (Pierre Harvey Richemond et al., 2024) or to forgo it entirely (Gao et al., 2024). However, this form has multiple benefits, and QRPO's main contribution is to transform the reward to maintain this regression objective while making the partition function tractable.

**Why this objective?**   First, unlike policy improvement methods such as PPO, the training completion $y$ does not have to be online with respect to $\pi_\theta$ and can be sampled from any distribution, such as an offline dataset to leverage pre-existing data (as in DPO originally), online data when novel exploration is needed, as in Online DPO (Qi et al., 2024), or a mix of offline, off-policy, and on-policy data. This is a significant benefit of policy fitting methods.

Second, the primary rationale for employing an absolute reward signal instead of a relative one (as in DPO or REBEL) is that an absolute signal not only contains strictly more information but also qualitatively influences the optimization dynamics when data coverage is limited. Relative reward

methods leverage limited data coverage by increasing the difference between paired samples, often simultaneously lowering the probabilities of both (Pal et al., 2024). This consistent reduction in probabilities leads to unintended probability mass drifting toward out-of-distribution samples, resulting in unstable or ineffective long-term training (Razin et al., 2025). In contrast, an absolute reward approach enforces model probabilities to match fixed, predefined targets, directing probability mass explicitly toward higher-reward samples rather than drifting towards out-of-distribution outcomes, enabling stable and effective optimization. See Appendix A for a comprehensive discussion.

**Insight 1: The partition function can be expressed in terms of rewards only**  The main argument commonly supporting the difficulty of estimating $Z(x)$ in Equations 6 and 2 is the intractability of the infinite sum over completions $y$ for an LLM. Our first insight is to show that the partition function can be expressed using only the distribution of reference rewards $\mathcal{R}(x, y)$ for completions $y$ drawn from the reference policy $\pi_{ref}(\cdot \mid x)$. Let $P_{ref}(r|x)$ be the probability of a reference reward $r$, and $M$ denote the moment generating function (MGF). We show that (proof in Appendix D):

$$Z(x) = \mathbb{E}_{r \sim P_{ref}(\cdot|x)} \left[ e^{\frac{1}{\beta} r} \right] = M_r \left( 1/\beta \right). \tag{7}$$

This derivation allows us to have a better intuition of the challenge of estimating the partition function: it is particularly challenging with reward distributions that have a right tail because the exponent of large rewards, which happen with low probability, makes the estimate very noisy. We support this intuition with a theoretical analysis of the noise amplitude of $Z(x)$ when rewards follow a standard normal distribution, which can be an example for typical LLM reward distributions, such as in our experiments, which do have a right tail. We show that with $\beta = 0.1$, approximately $10^{40}$ samples are required to achieve a reasonable signal-to-noise ratio in the estimate of $Z(x)$ (proof in Appendix H).

**Insight 2: We can derive Z analytically if we know the reward distribution**  Our rewriting of $Z(x)$ in Equation 7 opens the new alternative of computing $Z(x)$ analytically. In fact, the partition function takes the form of the moment generating function at $\frac{1}{\beta}$ for the reference reward distribution, which is known for common distributions or could be derived by computing the expectation.

**Insight 3: QRPO—the quantile-based reward gives a uniform distribution**  The reference reward distribution can be arbitrary, and assuming a specific distribution to leverage the above insight could introduce significant errors. However, we can transform the initial reward into one with a known distribution. QRPO uses the quantile transformation to achieve this. We begin by defining the cumulative distribution function (CDF) of rewards under the reference policy for a fixed input $x$:

$$F_{ref}(x, r) = \Pr_{y' \sim \pi_{ref}(\cdot|x)} \left\{ \mathcal{R}(x, y') \leq r \right\}. \tag{8}$$

This CDF intuitively represents the probability that a sample from the reference policy will have a reward less than or equal to a given value $r$. We then introduce the *quantile reward transformation*:

$$\mathcal{R}_q(x, y) = F_{ref}(x, \mathcal{R}(x, y)) = \Pr_{y' \sim \pi_{ref}(\cdot|x)} \left\{ \mathcal{R}(x, y') \leq \mathcal{R}(x, y) \right\}. \tag{9}$$

When the reference reward distribution can be treated as continuous, such as with a reward model or a coding test-case pass rate, as in our experiments, the resulting distribution of the quantile-transformed reference rewards $\mathcal{R}_q(x, y)$ for $y \sim \pi_{ref}(\cdot \mid x)$ is *uniform*, independently of the reference policy (Appendix E).[3] Therefore, using Eq. 7 which holds for any rewards, the corresponding partition function for this new reward is simply the MGF of the uniform distribution (details in Appendix F):

$$Z_q(x) = \beta \left( \exp \left( 1/\beta \right) - 1 \right). \tag{10}$$

Hence, we obtain a tractable regression form of Objective 6 (see Appendix A for an interpretation):

$$\mathcal{L}_{QRPO} = \mathbb{E}_{x,y} \left[ \left( \mathcal{R}_q(x, y) - \beta \log \left\{ \beta \left( \exp \left( \frac{1}{\beta} \right) - 1 \right) \right\} - \beta \log \frac{\pi_\theta(y \mid x)}{\pi_{ref}(y \mid x)} \right)^2 \right]. \tag{11}$$

For computational stability, we neglect the $-1$ term as $\exp(1/\beta)$ is typically very large for useful values of $\beta \leq 0.1$, and use $\log(\beta) + 1/\beta$ for $\log Z$, obtaining

$$\mathcal{L}_{QRPO} = \mathbb{E}_{x,y} \left[ \left( \mathcal{R}_q(x, y) - \beta \log(\beta) - 1 - \beta \log \frac{\pi_\theta(y \mid x)}{\pi_{ref}(y \mid x)} \right)^2 \right]. \tag{12}$$

---

[3] For completeness, we also derive QRPO for reward distributions with a small number of possible values in App. F by directly estimating the expectation for quantile rewards, as they do not result in a known distribution.

---

**Algorithm 1** (Offline) Quantile Reward Policy Optimization

---

**Require:** Dataset $\mathcal{D} = \{x_i, y_i\}$ of prompts and completions, reference policy $\pi_{ref}$, reward function $\mathcal{R}$, scaling factor $\beta$, number of reference rewards $n$ to generate, and number of training steps $T$.

> **Pre-computation phase:**
> *(Dataset rewards)* $\forall x_i, y_i \in \mathcal{D}$, pre-compute $\pi_{ref}(y_i \mid x_i)$ and $\mathcal{R}(x_i, y_i)$.
> *(Reference rewards)* $\forall x_i \in \mathcal{D}$, sample $n$ reference completions $y_{i,j} \sim \pi_{ref}(\cdot \mid x_i)$ and annotate them with $\mathcal{R}$ to obtain the reference rewards $\mathcal{S}_{ref,i} = \{\mathcal{R}(x_i, y_{i,j})\}$.

> **Training phase:**
> Initialize policy $\pi_\theta \leftarrow \pi_{ref}$ and minimize the following loss with gradient descent for $T$ steps.
>
> $$\mathbb{E}_{(x_i, y_i) \sim \mathcal{D}} \left[ \mathcal{R}_q(x_i, y_i) - \beta \log \beta - 1 - \beta \log \frac{\pi_\theta(y_i \mid x_i)}{\pi_{ref}(y_i \mid x_i)} \right]^2$$
>
> Where $\mathcal{R}_q(x_i, y_i) = \frac{1}{|\mathcal{S}_{ref,i}|} \sum_{\mathcal{R}(x_i, y_{i,j}) \in \mathcal{S}_{ref,i}} \mathbf{1}\{\mathcal{R}(x_i, y_{i,j}) \leq \mathcal{R}(x_i, y_i)\}$

---

**QRPO as a framework**   We can generalize the derivation of QRPO to more general reward transformations. Specifically, we can apply a wide range of functions $f$ on top of the quantile reward, while still retaining the ability to analytically derive the corresponding partition function when the reference rewards can be treated as continuous (proof in App. F):

$$\tilde{\mathcal{R}}(x, y) = f\left(\mathcal{R}_q(x, y)\right); \quad \tilde{Z}(x) = M_{\tilde{r}}\left(1/\beta\right) = \int_0^1 \exp\left(\tfrac{1}{\beta} f(t)\right) dt. \tag{13}$$

where $f$ represents any function defined on the interval $[0, 1]$. This formulation demonstrates a significant flexibility of QRPO, positioning it as a general framework and allowing the design of reward distributions with specific properties that shape the structure of the optimal policy.

**A new optimal policy**   The optimal policy of the RL fine-tuning objective depends on the reward. Therefore, optimizing the quantile reward (Eq. 9) or its generalized transformed version (Eq. 13) produces policies with different properties compared to optimizing the original reward. Notably, Balashankar et al. (2024) demonstrate that optimizing the quantile reward is equivalent to optimizing the win rate against the reference policy and further allows to optimize the policy performance for a specific inference-time scaling strategy by choosing a specific transformation $f$ on top of the quantile reward in Eq. 13. This highlights a key advantage of the QRPO framework.

**Foundations for theoretical analyses**   The analytical tractability of the partition function in QRPO is just one example of the broader opportunities for theoretical analysis enabled by quantile-based reward framework. To illustrate this, we conduct an initial investigation into the impact of different transformation functions in Eq. 13. In Appendix K, we show that for a class of functions $f(t)$ that are strictly increasing and finite at $t = 1$, the resulting optimal policy is invariant to the specific choice of $f$ when $\beta$ is sufficiently small. However, understanding the effects of other types of functions, the relationship between the optimal policy and the optimization dynamics, and several other related questions remain open problems and represent key directions for future research.

**Noise amplitude in the regression objective**   We have eliminated the noise in the partition function by using an exact expression computed analytically. However, we have introduced new noise in the regression objective, originating from the transformed reward $\mathcal{R}_q(x, y)$, as it requires estimating a quantile. We show in Appendix H, using a similar analysis as in Insight 1 with a Gaussian distribution, that the noise in the calibrated target (Eq. 6) computed by keeping the original reward and estimating $Z(x)$ is significantly higher than when estimating the quantile reward and using the exact expression of $Z(x)$, in particular for the order of magnitude of $\beta$ commonly used in RL fine-tuning (0.1 and lower). Hence, QRPO reduces the noise in the objective function dramatically, and we therefore enable the general approach of policy fitting with pointwise absolute rewards to work well in practice.

**A practical algorithm**   QRPO is implemented similarly to popular policy fitting algorithms primarily used in an offline regime for their computational simplicity, so we present an offline version of QRPO (although QRPO is not limited to the offline setting). QRPO does not use a pairwise preference dataset; it uses a dataset with prompts and completions (as in SFT), but also requires a reward model or function. A preliminary step (*pre-computation*) is to generate reference completions and annotate them with rewards denoted $\mathcal{S}_{ref,i} = \{\mathcal{R}(x_i, y_{i,j})\}$ to estimate the quantile rewards during training. We used between 1 and 20 reference completions in our experiments, and note that these generated completions can themselves be reused as the training completions to serve as

**Figure 3: Scaling** Performance of QRPO with a varying number of reference rewards generated during the pre-computation phase to estimate quantile rewards, in off-policy and offline distribution shifts with different values of KL regularization ($\beta$) for Llama on Magpie-Air with ArmoRM. We report the average online test reward with error bars indicating the standard deviation over three seeds. In the off-policy case (where training samples closely match the reference model's distribution, here generated by the reference model itself), performance steadily scales with more reference rewards, and is cost-effective for higher regularization ($\beta = 0.1$), typically used in large post-training pipelines. In contrast, in the offline case (high distribution shift due to training samples from a more performant model in this case) QRPO achieves near-optimal performance with very few reference rewards and has minimal gains from additional pre-computation, which may be more cost-effective for lower regularization ($\beta = 0.003$). This shows QRPO's pre-computation scalability and effectiveness in both scenarios.

an *off-policy* dataset. Then, in the training phase, QRPO minimizes Equation 12 with fully offline stochastic gradient descent, computing the quantile rewards using $\mathcal{S}_{ref}$. A summary of the algorithm is presented in Algorithm 5. A nice feature of the quantile reward in QRPO is that, as in DPO, it allows comparing $\beta$ values across domains independently of the original scale of rewards.

## 4    Experiments

**Setup**    We conduct our experiments in general chat and code generation. Our main setting is a single-turn general chat task, a common research testbed for offline RL fine-tuning in LLM alignment, and we add a code generation task where the model is asked to generate code that passes unit tests given a description. We focus on comparing QRPO with policy fitting methods offline, and choose DPO, REBEL, and SimPO as our baselines. Each captures a fundamental means of fitting the solution of the RL fine-tuning objective: DPO for relative preferences, REBEL for the relative reward difference, and QRPO for the absolute reward. We added SimPO to compare to methods that replace the canonical implicit reward with a reward derived from an inductive bias (length normalization). We select a comprehensive set of models, datasets, and distribution shift settings and sweep over the same comprehensive hyperparameter budget for each algorithm, resulting in 225 trained models for each algorithm. Additionally, we adopt a three-fold data split (train/validation/test) with three seeds to perform robust hyperparameter selection and report error bars. Our setting captures many of the experimental settings used in previous related work (Gao et al., 2024; Meng et al., 2024) and allows for broader comparisons. We present our experimental protocol in detail in Appendix M.

**QRPO does not require a large pre-computation budget and scales with a larger budget to estimate the quantile reward**    To practically validate our choice to estimate the quantile rewards instead of the partition function, we plot in Figure 3 QRPO trained with an increasing number of reference rewards generated in the pre-computation phase to estimate the quantile reward, for different values of the KL regularization hyperparameter $\beta$, with lower $\beta$ leading to less regularization and a larger change in the model. We distinguish two settings: the *off-policy* setting, where the training samples are close to the distribution of the reference model and initial training checkpoint which are the same, e.g., in our case the samples have been generated from the reference model, and the *offline* setting, where there is a high distribution shift between the training samples and the reference/initial model, e.g., collected from humans or in our case in Magpie-Air they have been generated from a more performant model. In the off-policy case, the performance of QRPO scales with a larger number of reference rewards generated in the pre-computation phase, which indeed gives a more precise signal to estimate the quantile reward of the training samples, as they lie in a dense region of the estimated reference distribution, thanks to the weak distribution shift. However, in the offline case, with a high-quality dataset, the performance of the trained model is very lightly impacted by varying the number of reference rewards, and only a few reference rewards, as few as 1 or 3, are enough to capture the information needed to train the model. Indeed, in this case with high distribution shift, pre-computing more reference rewards does not necessarily make the quantile signal more precise, as the rewards of the training samples lie in the sparse right tail of the estimated reference distribution.

**Table 3:** QRPO compared to baselines. We report original (Orig.) and *length-controlled* (LC) rewards (defined in App. M) on a held-out test split with means and standard deviations over three seeds, and the AlpacaEval 2 original and length-controlled win rates (WR and LC) with their standard errors. Higher is better. We bold the highest numbers with intersecting means $\pm$ one standard deviation after rounding. The best model for each algorithm is selected according to a validation set from training with 9 hyperparameter combinations (learning rate and $\beta$) across 6 distribution shift settings, including the option to first SFT on the chosen answer (Initial vs. SFT-chosen) and then the option to fine-tune on set dataset completions (offline) or generate new completions from the initial checkpoint (off-policy) and annotate them (using the best vs. the worst among 6 completions, or by ranking a random pair of samples). We rank the preference pairs used by DPO and SimPO using the same reward model (ArmoRM) used by REBEL and QRPO, ensuring an equal potential signal source for all methods.

| Model | Method | Magpie–Air Dataset | | | | UltraFeedback Dataset | | | |
|---|---|---|---|---|---|---|---|---|---|
| | | ArmoRM Reward | | AlpacaEval 2 | | ArmoRM Reward | | AlpacaEval 2 | |
| | | LC | Orig. | LC (%) | WR (%) | LC | Orig. | LC (%) | WR (%) |
| Llama 8B Tülu 3 SFT | Initial | $0.1518_{\pm 0.0011}$ | $0.1519_{\pm 0.0010}$ | $18.1_{\pm 0.4}$ | $10.8_{\pm 1.1}$ | $0.1294_{\pm 0.0006}$ | $0.1293_{\pm 0.0005}$ | $18.1_{\pm 0.4}$ | $10.8_{\pm 1.1}$ |
| | SFT-chosen | $0.1648_{\pm 0.0008}$ | $0.1649_{\pm 0.0008}$ | $23.0_{\pm 0.3}$ | $20.6_{\pm 1.4}$ | $0.1219_{\pm 0.0008}$ | $0.1219_{\pm 0.0007}$ | $9.8_{\pm 0.4}$ | $7.5_{\pm 0.9}$ |
| | DPO | $0.1904_{\pm 0.0003}$ | $0.1943_{\pm 0.0002}$ | $47.7_{\pm 0.1}$ | $\mathbf{53.7}_{\pm 1.8}$ | $0.1493_{\pm 0.0001}$ | $0.1491_{\pm 0.0001}$ | $39.4_{\pm 0.4}$ | $34.3_{\pm 1.7}$ |
| | SimPO | $0.1975_{\pm 0.0003}$ | $\mathbf{0.1976}_{\pm 0.0002}$ | $49.2_{\pm 0.1}$ | $49.6_{\pm 1.8}$ | $\mathbf{0.1535}_{\pm 0.0009}$ | $\mathbf{0.1539}_{\pm 0.0002}$ | $47.0_{\pm 0.2}$ | $39.2_{\pm 1.7}$ |
| | REBEL | $0.1889_{\pm 0.0012}$ | $0.1937_{\pm 0.0011}$ | $46.3_{\pm 0.1}$ | $45.5_{\pm 1.8}$ | $0.1488_{\pm 0.0004}$ | $0.1487_{\pm 0.0005}$ | $39.5_{\pm 0.4}$ | $37.8_{\pm 1.7}$ |
| | **QRPO** | $\mathbf{0.2005}_{\pm 0.0004}$ | $0.1972_{\pm 0.0003}$ | $\mathbf{50.6}_{\pm 0.1}$ | $\mathbf{56.5}_{\pm 1.7}$ | $\mathbf{0.1556}_{\pm 0.0017}$ | $0.1504_{\pm 0.0008}$ | $\mathbf{49.8}_{\pm 0.1}$ | $\mathbf{65.6}_{\pm 1.7}$ |
| Mistral 7B Instruct v0.2 | Initial | $0.1598_{\pm 0.0008}$ | $0.1598_{\pm 0.0006}$ | $27.6_{\pm 0.4}$ | $21.6_{\pm 1.5}$ | $0.1348_{\pm 0.0009}$ | $0.1349_{\pm 0.0009}$ | $27.2_{\pm 0.4}$ | $21.4_{\pm 1.4}$ |
| | SFT-chosen | $0.1633_{\pm 0.0004}$ | $0.1633_{\pm 0.0006}$ | $25.3_{\pm 0.3}$ | $26.2_{\pm 1.6}$ | $0.1254_{\pm 0.0015}$ | $0.1258_{\pm 0.0013}$ | $13.4_{\pm 0.4}$ | $10.2_{\pm 1.1}$ |
| | DPO | $\mathbf{0.1898}_{\pm 0.0003}$ | $\mathbf{0.1901}_{\pm 0.0001}$ | $42.1_{\pm 0.1}$ | $\mathbf{49.8}_{\pm 1.8}$ | $0.1465_{\pm 0.0008}$ | $\mathbf{0.1480}_{\pm 0.0007}$ | $\mathbf{38.8}_{\pm 0.2}$ | $\mathbf{39.0}_{\pm 1.7}$ |
| | SimPO | $0.1879_{\pm 0.0012}$ | $0.1884_{\pm 0.0001}$ | $44.0_{\pm 0.1}$ | $\mathbf{49.9}_{\pm 1.8}$ | $\mathbf{0.1478}_{\pm 0.0007}$ | $\mathbf{0.1472}_{\pm 0.0005}$ | $\mathbf{38.8}_{\pm 0.2}$ | $34.8_{\pm 1.7}$ |
| | REBEL | $0.1884_{\pm 0.0002}$ | $0.1864_{\pm 0.0002}$ | $40.7_{\pm 0.1}$ | $46.5_{\pm 1.8}$ | $0.1466_{\pm 0.0006}$ | $0.1457_{\pm 0.0007}$ | $31.5_{\pm 0.2}$ | $35.5_{\pm 1.7}$ |
| | **QRPO** | $0.1893_{\pm 0.0003}$ | $0.1886_{\pm 0.0002}$ | $\mathbf{44.4}_{\pm 0.1}$ | $\mathbf{46.6}_{\pm 1.8}$ | $0.1470_{\pm 0.0007}$ | $0.1469_{\pm 0.0007}$ | $\mathbf{38.8}_{\pm 0.2}$ | $\mathbf{36.7}_{\pm 1.7}$ |

Furthermore, by varying the KL regularization strength, we observe a tradeoff between using the off-policy versus the offline regime. For a relatively large regularization ($\beta = 0.1$), the off-policy regime outperforms the offline regime with a relatively low number of reference rewards and continues to surpass it with more reference rewards. This aligns well with modern post-training pipelines (Grattafiori et al., 2024; Lambert et al., 2024) which use a relatively large regularization to train on large datasets targeting multiple tasks and capabilities, and perform multiple iterations of policy fitting on newly sampled data from the latest model. The Llama 3 models (Grattafiori et al., 2024), for example, have been trained with DPO with $\beta = 0.1$ for several rounds, updating the reference model and sampling from it at each round; QRPO would be easily integrated into this off-policy pipeline to provide its top performance. Also note that in this case, all the previously generated reference completions can be reused as training samples to generate a growing training dataset. Yet, for a low regularization ($\beta = 0.003$), the offline regime with high-quality offline data and minimal KL constraints yields the highest performance overall across $\beta$s, and the off-policy regime requires a large number of reference rewards to reach comparable performance. Although this setting combined with offline data would not provide stable results for large post-training pipelines due a large test-time distribution shift (see Appendix A for a comprehensive discussion), it is the setting where research papers report their best results, as it eventually provides the best performance on narrow tasks, leveraging high variance combined with hyperparameter optimization. Most of the best models for all the algorithms we train also result from this low regularization offline setting, which at the same time allows QRPO to provide strong results with very few reference rewards, but we minimize the impact from a selection bias by adopting a robust model selection protocol.

**QRPO demonstrates top-tier performance** Table 3 shows that QRPO achieves notably strong results in optimizing the offline RL fine-tuning objective for general conversational tasks, translating effectively to downstream evaluations on AlpacaEval. QRPO models consistently rank among the top-performing models based on length-controlled metrics, suggesting that the approach also mitigates the well-known length bias issue, which we analyze in the next section. Notably, on the Magpie-Air dataset—which contains completions from Llama-8B-Instruct, which is stronger than our initial models—we only use a single reference reward in QRPO to estimate the quantile reward, corroborating our results from the scaling analysis (section 4). Indeed, most of the algorithms trained on Magpie-Air obtained their best performance in the offline distribution shift setting, leveraging the quality of the dataset. For the UltraFeedback dataset, which is significantly lower quality than Magpie-Air and the initial models, we still only used 3 reference rewards for QRPO, and most of the algorithms obtained their best performance in the off-policy distribution shift setting, replacing the completions from the dataset with the reference model completions. Appendix N shows the hyperparameters and distribution shift settings for which each algorithm obtained its best model.

Table 2 also shows that when the task is canonically expressed with rewards instead of preferences, QRPO is more effective than the baselines in optimizing these rewards, which for baselines are converted to preferences and lose their absolute signal. In our case, it's the performance on LeetCode problems where the reward of a solution to a problem is computed as the test-case pass rate over the test suite for the problem. This opens an avenue for novel applications using policy fitting methods for fine-tuning large language models. It is also interesting to note that SimPO, with its inductive bias made for conversational abilities, falls significantly short in optimizing this coding reward.

**QRPO with robust rewards is less prone to length bias than preference methods with labels from the same rewards** We analyze the best model for each algorithm and setting for signs of length bias.

Length bias (or correlation) is a spurious correlation between the output length and the performance of models resulting from RLHF and DAA training (Dubois et al., 2024; Singhal et al., 2024; Park et al., 2024). It's undesired as it increases the verbosity and inference cost of the models without necessarily improving their unbiased performance. It is inherited from training on (underspecified) preference labels. However, the reward model ArmoRM (Wang et al., 2024) we use in our experiments has been explicitly trained to counteract this length bias in its scalar value and we show that QRPO and REBEL trained with this scalar value exhibit less length bias than DPO and SimPO which convert it to preferences, potentially inheriting the length bias back. Similarly to Park et al. (2024), we study the completion length versus the implicit rewards. As we have generated reference completions for each prompt to train QRPO, we are in the position to leverage a better signal for length bias than previous work. We take a per-prompt standardized completion length, which for each prompt and completion generated by the trained model, is the length of the completion standardized by the distribution

**Figure 4: Length bias** Completion length difference from the initial checkpoint vs. the implicit reward of test completions generated by the best Llama model trained on Magpie-Air for each algorithm. Implicit rewards (colloquially, DPO rewards) are induced by the policy, which is optimal for these rewards according to the RL fine-tuning objective. We report a linear fit with a marker indicating the average completion length and Spearman rank correlation. SimPO reduces the average completion length compared to DPO (cloud shifted to the left), but its policy still exhibits as much length bias as the DPO policy. In contrast, QRPO and REBEL, which use the reward signal, do not exhibit a length bias trend.

of reference completions in that prompt (that we generated in the pre-computation phase). Hence, unlike prior work, our $x$-axis captures the length increase for each prompt independently. For implicit rewards, in the same manner as Park et al. (2024), for each model (policy), we consider the reward parameterized by this policy according to the RL fine-tuning objective (Eq. 3 without $Z$, colloquially the DPO reward). This is the reward for which the trained policy is optimal. Hence, if there is strong length bias or correlation in the reward, we can say that this is also a feature of the learned policy, as it is optimal for that reward. Results are shown in Figure 4 and Appendix N. We observe the typical positive slope for DPO reported by Park et al. (2024), and surprisingly, we observe the same slope for SimPO. Although SimPO decreases length overall (the cloud is shifted to the left compared to other models), the length bias trend is still present. On the contrary, we observe that QRPO and REBEL exhibit much less length bias. We conjecture that this improvement is due to effective training with the robust ArmoRM reward signal, which for SimPO and DPO is lost in the binary preferences.

## 5 Related work

**Preference optimization** A large body of research on policy fitting methods in RL fine-tuning has focused on developing preference-based algorithms to compete directly with DPO (Rafailov et al., 2023) by better learning from preferences. The list includes IPO (Azar et al., 2024), ORPO (Hong et al., 2024), CPO (Xu et al., 2024a), and SimPO (Meng et al., 2024). However, many of these methods deviate from optimizing the RL fine-tuning objective and attempt to better optimize human feedback for conversational abilities. With comprehensive results, SimPO positions itself as the most competitive method and shows that a well-tuned DPO is still superior to most other methods. Hence, we choose to compare to DPO to capture most of these methods, to SimPO to capture one competitive way to include inductive bias in the objective, and to REBEL to illustrate fundamentally different ways of optimizing the RL fine-tuning objective: preferences vs. reward differences.

**Offline alignment with a pointwise signal**    DRO (Pierre Harvey Richemond et al., 2024) formulates the same regression objective in Eq. 6, but proceeds to learn the partition function jointly with the policy. Although the approach is close to our motivation and derivation, DRO no longer benefits from the simplicity and stability of policy fitting methods, as it learns a policy and a value function through joint optimization. In this work, we evaluate simple policy fitting methods and thus do not compare against DRO (which also does not have an open implementation). More recently, the same objective has also been used in SPO (Cohen et al., 2025), AGRO (Tang et al., 2025), and TBA (Bartoldson et al., 2025) which like DRO get both the policy and the partition function from Eq. 5 but instead of learning the partition function they propose different ways of estimating it. These works mostly focus on semi-online training and do not demonstrate competitive performance in the offline regime. Alternatively, KTO (Ethayarajh et al., 2024) drops the partition function in the KL-regularized optimal reward (Eq. 3) and optimizes for human utility based on prospect theory (Kahneman & Tversky, 2013) instead of preferences, making the partition function irrelevant and the objective optimizable with pointwise binary human feedback. KTO could be considered as a baseline using a pointwise signal; but, since it excessively diverges from the RL fine-tuning objective, and Meng et al. (2024) show that empirically, using similar benchmarks to us, we do not KTO consider as a baseline.

**Quantile rewards**    The cumulative density function of the rewards under the reference policy has been used in both the vBoN (Amini et al., 2025) and BOND (Sessa et al., 2025) algorithms to derive the distribution of the best-of-N (BoN) policy and identify the BoN policy as induced by optimizing the $\log$-quantile rewards in the RL fine-tuning objective. However, both derivations focus specifically on the event defining the BoN probability and do not stem from the solution to the KL-regularized RL objective, which in these works remains intractable due to the partition function. QRPO instead uses the quantile reward as a universal solution to derive the partition function and the expression of the induced policy. In fact, we have the BoN policy as a special case of our framework using the $\log$-quantile transformation. Another difference with our work is that we opt for a regression loss rather than a distribution matching loss. Gulcehre et al. (2023, ReST) also experiment with an SFT loss on the best off-policy data according to a quantile reward threshold. We believe that comparing QRPO, BOND, vBoN, and ReST can be made in the scope of the more targeted open question of studying the BoN distillation problem, but is out of the scope of the claims made in our work.

**Offline & off-policy policy improvement**    There is a growing body of research attempting to make policy improvement methods work off-policy, such as TOPR (Roux et al., 2025), which uses truncated importance sampling ratios, and ReMix (Liang et al., 2025), which combines regularization techniques for data reuse. This is a promising direction, confirming that QRPO tackles an important gap in the literature to optimize pointwise absolute rewards with data from other distributions.

## 6   Conclusion and discussion

We have presented Quantile Reward Policy Optimization (QRPO), a novel RL fine-tuning algorithm that uses quantile rewards to derive a tractable expression of the optimal solution to the KL-regularized RL objective with its partition function, and fits it with a simple regression on individual samples; therefore, directly using the pointwise absolute reward of samples instead of relying on preferences. We have shown that QRPO provides strong empirical results on chat and coding tasks, scales with the pre-computation budget, and is less prone to length bias. We believe that QRPO unlocks a new avenue for policy fitting methods, which can combine any data distribution during training, benefit from offline scaling, and now use the signal from absolute pointwise rewards.

**Limitations and open questions**    We have proposed QRPO as a framework and shown that transformations can be applied on top of the quantile to reshape it into the desired form while still maintaining a tractable partition function. However, we have not explored these transformations empirically. We encourage future work to study them and provide further understanding in Appendix B. A notable limitation of the quantile reward is that it is based on the reference policy, and when the model improves drastically beyond the reference policy, the quantiles become less informative. Still, this can be overcome by performing iterative training and updating the reference policy, as is done in many state-of-the-art open-source post-training pipelines. Finally, QRPO and policy fitting methods in general can leverage any distribution, including online data, but we only focused on offline data to compare with other policy fitting methods. An open question is the impact of online data, potentially in combination with offline and off-policy data, and a comparison to policy improvement methods such as GRPO, which rely critically on online data.

## Acknowledgments and Disclosure of Funding

This work was supported as part of the Swiss AI Initiative by a grant from the Swiss National Supercomputing Centre (CSCS) under project ID a10 on Alps. We are grateful to Mikhail Pavlov for insightful comments on the theoretical aspects of the project, Mikhail Terekhov for discussions on the project theory and experiments, to Anja Surina, Mikhail Terekhov, and Roman Machacek for suggestions on datasets for the coding experiment, and to Ivan Pavlov and Juan Garcia Giraldo for support with some evaluations. We thank the CSCS and EPFL SCITAS staff for discussions on infrastructure engineering. Finally, we thank Karin Gétaz for the administrative support provided within EPFL.

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

# Appendices

# Contents

# A    QRPO loss and gradient interpretation

**Figure 5:** Distribution of the initial reward $\mathcal{R}$ for a given prompt under the reference ($\pi_{ref}$) and optimal ($\pi^*$) policy distributions for different values of $\beta$. The optimal policy is maximizing the RL fine-tuning objective (Equation 1) with the quantile reward $\mathcal{R}_q$. In this plot, the reference reward distribution is assumed to be Gaussian, which can serve as a good example for rewards obtained from a reward model in a general chat task. With this assumption, we can compute both the reference and optimal reward distributions analytically. Refer to the final paragraph for the derivation. **Left:** Gradient update direction for samples with different reward values. The quantile reward $\mathcal{R}_q = \beta \log Z_q$ corresponds to the reward $\mathcal{R}$ at the intersection point of the densities. This value plays the role of a threshold, which separates the samples with rewards below the threshold that should have their probability decreased, and samples with rewards above the threshold that should have their probability increased. **Right:** Position of the optimal policy reward distribution for different values of $\beta$. A smaller $\beta$ leads to a larger target distribution shift, resulting in a gradient that decreases the probability of the majority of samples around the reference policy (see left plot for the intuition).

The QRPO loss has the form: $\quad \mathcal{L}_{QRPO} = \mathop{\mathbb{E}}_{x,y} \left[ \left( \mathcal{R}_q(x,y) - \beta \log Z_q - \beta \log \frac{\pi_\theta(y \mid x)}{\pi_{ref}(y \mid x)} \right)^2 \right]$

with $\quad\quad Z_q = \beta \left( \exp\left( \frac{1}{\beta} \right) - 1 \right)$

or with an additional transformation of the quantile reward $\tilde{\mathcal{R}} = f(\mathcal{R}_q)$ and its corresponding $\tilde{Z}$ as described in Equation 13 and Appendix B.

The loss has a simple and intuitive meaning of pushing the probability of each sample in the training dataset (offline or online) towards the probabilities of the optimal policy that maximizes the RL fine-tuning objective (Equation 1) with the transformed reward $\mathcal{R}_q$ (or $\tilde{\mathcal{R}}$). Writing the model updates given by the QRPO loss gradient and highlighting terms in the equation we get:

$$-\nabla_\theta \mathcal{L}_{QRPO} = 2\beta \mathop{\mathbb{E}}_{x,y} \left[ \left( \Big[ \underbrace{\mathcal{R}_q(x,y) - \beta \log Z_q}_{\textit{calibrated target}} \Big] - \beta \log \frac{\pi_\theta(y \mid x)}{\pi_{ref}(y \mid x)} \right) \nabla_\theta \log \pi_\theta(y \mid x) \right].$$

(14)

To minimize the loss, starting from an initial checkpoint equal to the reference policy, the update increases (respectively decreases) the policy's probability for samples with a positive (respectively negative) calibrated target. This adjustment continues from an initial log-ratio of zero until the log-ratio aligns with the calibrated target.

In light of known issues with existing policy fitting methods—such as their tendency to decrease the probabilities of both chosen and rejected samples (as seen in DPO-like approaches)—it is particularly interesting to analyze QRPO from the same perspective. As shown above, at the start of training, the QRPO update increases the probability of a sample only if its calibrated target is positive. Since the term $\beta \log Z_q$ is constant across all samples (i.e., independent of $x$ and $y$ as the quantile reward normalizes the signal in each prompt), it acts as a threshold: a sample's reward $\mathcal{R}_q(x, y)$ must exceed this value for the probability to increase. For example:

- with $\beta = 0.1$, the threshold is $\beta \log Z_q \approx 0.77$,
- with $\beta = 0.01$, the threshold is $\approx 0.95$,
- and with $\beta = 0.001$, it reaches $\approx 0.99$.

**Figure 6:** $\log \frac{\pi_\theta(y|x)}{\pi_{ref}(y|x)}$ dynamics for chosen and rejected responses when training with QRPO on a preference dataset in the offline regime with $\beta = 0.1$. We plot the average completion $\log$-ratio over all the prompts in the evaluation set. $\log$-ratios start at 0 when the trained policy is initialized $\pi_\theta(y \mid x) = \pi_{ref}(y \mid x)$, and during training become positive if $\pi_\theta(y \mid x) > \pi_{ref}(y \mid x)$ and negative if $\pi_\theta(y \mid x) < \pi_{ref}(y \mid x)$. As desired, we observe that they become positive for chosen completions and negative for rejected completion. Indeed, with $\beta = 0.1$ the quantile threshold for samples to increase their probabilities is at $\mathcal{R}_q \approx 0.77$ and in this dataset chosen samples have an average $\mathcal{R}_q$ of approximately 0.91, while rejected samples have an average of around 0.45. The initial decrease of the chosen $\log$-ratios can be attributed to generalization caused by the gradient being larger for rejected completions, until a clearer separation between them is learned by the model.

Recalling that $\mathcal{R}_q$ corresponds to the CDF value of the reference policy's reward distribution (Definition 9), we can interpret these thresholds in terms of quantiles. Specifically, if the training data points are sampled from the reference policy (off-policy data):

- for $\beta = 0.1$, only the top 23% of the samples will have their probabilities increased (see Figure 5 Left),
- for $\beta = 0.01$, only the top 5%,
- and for $\beta = 0.001$, only the top 1%.

A similar pattern holds in the offline case: for a sample from an offline dataset to be favored (i.e., to have its probability increased), its reward must lie within the top-performing percentile of completions under the reference policy, with the required percentile depending on the value of $\beta$.

To better understand why only a small fraction of samples see their probabilities increased, it is helpful to visualize the reward distribution and examine the position of the optimal policy distribution for different values of $\beta$. Figure 5 illustrates how the optimal policy distribution shifts as $\beta$ changes and indicates the update direction for samples based on their rewards. We can see that for small values of $\beta$, only a very small subset of samples from the reference policy will have their probabilities increased. Consequently, when training with off-policy data (i.e., samples drawn from the reference policy), it's normal to observe an overall decrease in the probabilities of training samples—except when using very large $\beta$. In the offline setting, a sample's probability will increase if it has a sufficiently high reward, specifically if it lies in the region where the optimal policy places a high probability mass.

We observe exactly this behavior for sample probabilities in our experiments. In the off-policy regime, we consistently see an overall decrease in the probabilities of training samples for all tested values of $\beta \in [0.0001, 0.1]$. In the offline regime, we use a preference dataset constructed for preference training (e.g., for DPO), which includes "chosen" and "rejected" samples for each prompt. Even though QRPO does not require preference pairs, we can use this separation to track sample probabilities dynamics for "better" and "worse" examples. In this dataset, chosen samples have an average $\mathcal{R}_q$ of approximately 0.91, while rejected samples have an average of around 0.45. We observe a decrease in probabilities for both chosen and rejected samples when $\beta \in [0.0001, 0.01]$. However, for $\beta = 0.1$, we see an increase in the probabilities of chosen samples and a decrease for rejected ones (see Figure 6). These results align precisely with the $\mathcal{R}_q = \beta \log Z_q$ threshold values discussed above.

It is important to highlight that even though training with a small $\beta$ results in updates that predominantly decrease the probabilities of training samples, this regime can still lead to effective reward optimization. Empirically, we observe that due to the policy's generalization capacity, reducing the probability of lower-reward samples more than that of higher-reward ones implicitly increases the

probability assigned to even better completions. Notably, our best results on the general chat task were achieved under such "aggressive" training regime with $\beta = 0.0003$. However, this strategy relies entirely on implicit generalization to shift probability mass toward higher-reward regions, without any explicit control over where that mass moves; thus, potentially resulting in inaccurate policy updates. In practice, we observed signs of such inaccuracies when training with small $\beta$: while the model initially improves rapidly, it later undergoes sharp degradation, likely due to the cumulative effect of these uncontrolled updates disrupting performance.

While this form of low-regularization training may work well for single-task settings (as in this and most other research papers), where the input domain is narrow and potential side effects of inaccurate policy updates on other domains go unnoticed, it becomes risky in broader contexts. In large-scale post-training pipelines aimed at improving multi-task performance, such uncontrolled drift may degrade results on tasks not represented in the training data. For this reason, we consider the "aggressive" small-$\beta$ training regime unsuitable for scenarios where robust generalization across tasks is critical. Conversely, training with a larger $\beta$ constrains the optimal policy to a lower reward region compared to small-$\beta$ regimes, potentially limiting overall performance. However, large-scale post-training pipelines typically mitigate this limitation by performing a few iterations of training with relatively high $\beta$, interleaved with collecting fresh on-policy data. This iterative approach enables gradual improvement of the policy while maintaining explicit control over probability mass drift (by leveraging on-policy data for the intermediate checkpoints), ultimately allowing the model to approach the performance levels achievable with small-$\beta$ training, but in a safer and more robust manner. It can be shown theoretically that performing $N$ iterations of training with $\beta$ while updating the reference policy to the latest policy between iterations leads to the same optimal policy as training in a single stage with $\beta/N$. Indeed, to get the optimal policy after the second iteration, we can use the closed-form solution (Equation 2) for the optimal policy and replace the reference policy with the optimal policy from the previous step:

$$
\begin{aligned}
\pi_{(2)}^*(y \mid x) &= \frac{1}{Z_{(2)}(x)} \pi_{(1)}^*(y \mid x) \exp\left(\frac{1}{\beta}\mathcal{R}(x,y)\right) \\
&= \frac{1}{Z_{(2)}(x)} \frac{1}{Z_{(1)}(x)} \pi_{ref}(y \mid x) \exp\left(\frac{1}{\beta}\mathcal{R}(x,y)\right) \exp\left(\frac{1}{\beta}\mathcal{R}(x,y)\right) \\
&= \frac{1}{Z_{(1,2)}(x)} \pi_{ref}(y \mid x) \exp\left(\frac{2}{\beta}\mathcal{R}(x,y)\right).
\end{aligned}
$$

Eventually, if one does $N$ iterations with $\beta$ the final optimal policy is

$$
\pi_{(N)}^* = \frac{1}{Z(x)} \pi_{ref}(y \mid x) \exp\left(\frac{N}{\beta}\mathcal{R}(x,y)\right),
$$

which equals the optimal policy in the closed-form solution for $\beta/N$, providing a principled explanation for why large-$\beta$ iterative schemes can reach similar final performance to single-stage small-$\beta$ training, but in a more stable manner.

**Interpretation of the gradient in the on-policy case**    Observe that in the derivation of the gradient in Equation 14, we pulled the derivative through the expectation without taking into account that $y$ can be online (as it would impact the gradient). Indeed, in the case of using online data in QRPO, we do not need to take a derivative of the expectation over $y \sim \pi_\theta(\cdot \mid x)$ like it is usually done in RL as we want the equality in Equation 5 to hold for all available $y$, whereas taking a derivative of the expectation over $y$ will reduce the support of its distribution to the cases where the equality in Equation 5 is easy to achieve. Hence, for the online case, we have the same gradient expression, but we can interpret it differently, as now the rewards are changing with the online sampling from the policy:

$$
-\nabla_\theta \mathcal{L}_{QRPO} = 2\beta \underset{\substack{x \sim \mathcal{D} \\ y \sim \pi_\theta(\cdot \mid x)}}{\mathbb{E}} \left[ \left( \left[ \underbrace{\mathcal{R}_q(x,y) - \beta \log \frac{\pi_\theta(y \mid x)}{\pi_{ref}(y \mid x)}}_{\text{reward with KL penalty}} \right] - \beta \log Z_q \right) \nabla_\theta \log \pi_\theta(y \mid x) \right].
$$

Notice that it can be seen as a standard *policy gradient* with an advantage $\left( \left[\mathcal{R}_q(x,y) - \beta \log \frac{\pi_\theta(y|x)}{\pi_{ref}(y|x)}\right] - \beta \log Z_q \right)$ made from a reward that includes a KL penalty

term, typically included in the reward in online RL methods, and a baseline term. One may argue that the baseline term $\beta \log Z_q$ is a constant, thus, may not be a good substitute for the baseline term which is typically taken as the average performance of the training policy for each prompt, such as the critic for PPO and the Monte-Carlo average return for GRPO. However, the baseline signal that QRPO brings is in two folds: the first fold already comes from the transformation to a quantile reward, which re-centers with respect to the performance of a *reference* policy and normalizes its variance; the second fold comes with $\beta \log Z_q$ which is subtracted to reflect the difference to the *optimal* policy. Indeed, we show below that $\beta \log Z_q = \beta \log Z_q(x) = \mathbb{E}_{y \sim \pi^*(\cdot|x)} \left[ \mathcal{R}_q(x,y) - \beta \log \frac{\pi^*(y|x)}{\pi_{ref}(y|x)} \right]$ is the average performance of the optimal policy with the reward including the KL penalty. Thus, $\beta \log Z_q$ is a precise baseline estimate when the model approaches the optimal policy.

From Equation 5: $\forall y, \qquad \mathcal{R}(x,y) - \beta \log Z(x) = \beta \log \frac{\pi^*(y \mid x)}{\pi_{ref}(y \mid x)}$

$$\Rightarrow \qquad \beta \log Z(x) = \mathcal{R}(x,y) - \beta \log \frac{\pi^*(y \mid x)}{\pi_{ref}(y \mid x)}$$

$$\Rightarrow \qquad \mathbb{E}_{y \sim \pi^*(\cdot|x)} \left[ \beta \log Z(x) \right] = \mathbb{E}_{y \sim \pi^*(\cdot|x)} \left[ \mathcal{R}(x,y) - \beta \log \frac{\pi^*(y \mid x)}{\pi_{ref}(y \mid x)} \right]$$

$$\Rightarrow \qquad \beta \log Z(x) = \mathbb{E}_{y \sim \pi^*(\cdot|x)} \left[ \mathcal{R}(x,y) - \beta \log \frac{\pi^*(y \mid x)}{\pi_{ref}(y \mid x)} \right]$$

And when using the reward $\mathcal{R}_q$ $\qquad \beta \log Z_q(x) = \mathbb{E}_{y \sim \pi^*(\cdot|x)} \left[ \mathcal{R}_q(x,y) - \beta \log \frac{\pi^*(y \mid x)}{\pi_{ref}(y \mid x)} \right]$

**On-policy training further improves QRPO performance**  It is well established that on-policy training improves the performance of policy optimization methods (Lanchantin et al., 2025; Guo et al., 2024), as it allows for better control over output probability mass drift. Although on-policy training is beyond the main scope of this work, we nevertheless verify that QRPO also benefits from this regime. Specifically, we conduct ablations where online data is incorporated alongside offline data during QRPO training on the Magpie-Air dataset. This results in a 26% greater reward improvement compared to training with offline data alone (see Figure 7), highlighting the strong potential of QRPO in on-policy settings.

**Figure 7:** Performance of QRPO under two training regimes: using only offline completions and using a mix of offline and online completions.

**Dynamic-reference KL for stable and flexible QRPO**  QRPO inherently optimizes a KL-regularized reinforcement learning objective (Eq. 1). While there is ongoing debate about the necessity of explicit KL penalties, particularly in the context of verifiable rewards (Yu et al., 2025), recent work presents a more nuanced perspective: Liu et al. (2025) demonstrate that combining KL regularization with periodic updates of the reference policy to recent checkpoints both stabilizes training and prevents stagnation, thereby enabling continued improvement during prolonged RL. Consequently, we do not view the inclusion of KL regularization as a limitation of QRPO; rather, we recommend adopting a dynamic-reference approach, where the reference policy is periodically updated to facilitate further reward optimization. Moreover, this strategy helps prevent saturation of the training signal when the trained policy approaches the top quantile, ensuring the quantile reward remains informative throughout training.

**Figure 5: Deriving the reward distribution for the optimal policy analytically**  To derive the distribution of the initial reward $\mathcal{R}(x,y)$ under the optimal policy when optimizing the quantile reward $\mathcal{R}_q(x,y)$, we start from the closed-form solution in Equation 2:

$$\pi^*(y \mid x) = \frac{1}{Z_q} \pi_{ref}(y \mid x) \exp \left( \frac{1}{\beta} \mathcal{R}_q(x,y) \right). \tag{15}$$

Next, we sum both sides over all completions $y$ corresponding to a given reward $r = \mathcal{R}(x, y)$ for a particular prompt $x$ to obtain the probability $P^*(r \mid x)$ of observing reward $r$ under the optimal policy:

$$
\begin{aligned}
P^*(r \mid x) &= \sum_{y:\ \mathcal{R}(x,y)=r} \pi^*(y \mid x) = \sum_{y:\ \mathcal{R}(x,y)=r} \frac{1}{Z_q} \pi_{ref}(y \mid x) \exp\left(\frac{1}{\beta} \mathcal{R}_q(x, y)\right) \\
&= \sum_{y:\ \mathcal{R}(x,y)=r} \frac{1}{Z_q} \pi_{ref}(y \mid x) \exp\left(\frac{1}{\beta} F_{ref}(x, r)\right) \\
&= \frac{1}{Z_q} \exp\left(\frac{1}{\beta} F_{ref}(x, r)\right) \sum_{y:\ \mathcal{R}(x,y)=r} \pi_{ref}(y \mid x) \\
&= \frac{1}{Z_q} \exp\left(\frac{1}{\beta} F_{ref}(x, r)\right) P_{ref}(r \mid x),
\end{aligned}
$$

where $P_{\mathrm{ref}}(\cdot \mid x)$ denotes the reward distribution induced by $y \sim \pi_{ref}(\cdot \mid x)$. We used the definition of the quantile reward from Equation 9, namely $\mathcal{R}_q(x, y) = F_{\mathrm{ref}}(x, \mathcal{R}(x, y))$ which is also equal to $F_{\mathrm{ref}}(x, r)$ in the conditioned sum, to express the exponential term as a function of $r$.

In case we can approximate the reward with a continuous distribution (e.g., in the case of using a continuous reward model), we can use the probability densities $p_{\mathrm{ref}}(\cdot \mid x)$ and $p^*(\cdot \mid x)$ to express the reference and the optimal policy reward distributions:

$$
p^*(r \mid x) = \frac{1}{Z_q} p_{ref}(r \mid x) \exp\left(\frac{1}{\beta} F_{ref}(x, r)\right). \tag{16}
$$

Hence, assuming that $p_{\mathrm{ref}}(\cdot \mid x)$ is standard normal we can illustrate the shift in reward distribution of the optimal policy relative to that of the reference policy in Figure 5. This figure should be interpreted qualitatively, as the actual distribution $P_{\mathrm{ref}}(\cdot \mid x)$ can be arbitrary in real-world scenarios.

## B  Guidelines for using QRPO with quantile transformation functions

**Table 4:** Closed-form partition functions for common reward transformations. erfi is the imaginary error function and $\mathrm{CDF}^{-1}_{\mathcal{N}(0,1)}$ is the inverse CDF of the standard normal distribution.

| Transformation $f(t)$ | Partition function $\tilde{Z}_f(\beta)$ |
|---|---|
| $t$ | $\beta\left(e^{1/\beta} - 1\right)$ |
| $\log t$ | $\dfrac{\beta}{\beta + 1}$ |
| $t^2$ | $\dfrac{\sqrt{\pi\beta}}{2}\, \mathrm{erfi}\left(1/\sqrt{\beta}\right)$ |
| $\sqrt{t}$ | $2\beta\left[\beta + (1 - \beta)\,e^{1/\beta}\right]$ |
| $\mathrm{CDF}^{-1}_{\mathcal{N}(0,1)}(t)$ | $\exp\left(\frac{1}{2\beta^2}\right)$ |
| $\mu + \sigma\,\mathrm{CDF}^{-1}_{\mathcal{N}(0,1)}(t)$ | $\exp\left(\frac{\mu}{\beta} + \frac{\sigma^2}{2\beta^2}\right)$ |

The generalized transformed reward for QRPO has the following expression (Equation 13):

$$
\tilde{\mathcal{R}}(x, y) = f\left(\mathcal{R}_q(x, y)\right).
$$

In this section, we discuss the common questions that need to be addressed when using a custom quantile transformation function $f$, notably

- how to choose it,
- how to compute its associated partition function, and
- how to choose a suitable scale for the hyperparameter $\beta$ to balance it.

**The impact of a quantile transformation function**    The optimal solution for the KL-regularized RL fine-tuning objective (Eq. 2) depends on the shape of the reward distribution; therefore, the primary function of a quantile transformation $f$ is to modify the transformed reward distribution to have some useful properties. One can easily obtain any desired distribution by using its inverse CDF as a quantile transformation function (since for any distribution $Y$, applying its inverse CDF to a uniform sample, $x \sim \texttt{Uniform}[0, 1]$, results in $\texttt{CDF}_Y^{-1}(x) \sim Y$). For example, one might wish the reward distribution to be Gaussian with mean $\mu$ and variance $\sigma^2$. However, after transforming it into a quantile reward $\mathcal{R}_q$, the resulting reward is uniformly distributed. In this case, the following transformation can be applied to recover the desired distribution: $\tilde{\mathcal{R}} = f(\mathcal{R}_q) = \sigma \cdot \texttt{CDF}_{\mathcal{N}(0,1)}^{-1}(\mathcal{R}_q) + \mu$.

**Choosing a quantile transformation function**    Generally, it is natural to require a transformation function $f$ to be monotonically increasing to ensure a meaningful transformed reward. This guarantees that higher original rewards correspond to higher transformed rewards, preserving the reward signal's intended structure. Furthermore, there are only two requirements for the transformation function $f$ to be suitable for QRPO:

1. it must be defined on the interval $[0, 1]$, and
2. its corresponding partition function must be finite.

The first condition is straightforward, and we provide some guidelines to find functions that meet the second condition.

**Using moment generating functions**    One possible approach to address the finiteness condition is to get insights from the distribution of transformed rewards $\tilde{\mathcal{R}}$. The partition function $\tilde{Z}(x)$ is simply a moment generating function (MGF) for the distribution of transformed rewards under the reference model: $\tilde{Z}(x) = \mathbb{E}_{\tilde{r}}\left[e^{t\tilde{r}}\right] = M_{\tilde{r}}(t)$, where $t = \frac{1}{\beta}$, $\tilde{r} = \tilde{\mathcal{R}}(x, y)$ for $y \sim \pi_{ref}(\cdot \mid x)$ (see Section D for the derivation). Thus, to check the finiteness of $\tilde{Z}(x)$, one can derive the distribution of the transformed rewards for completions sampled from the reference model, and then check the value of the corresponding MGF for $t = \frac{1}{\beta}$.

**Sufficient conditions**    There is a sufficient condition for the finiteness of $\tilde{Z}$: if the transformation function $f$ is upper-bounded on the interval $[0, 1]$, then the corresponding partition function is guaranteed to be finite (proof in Appendix G). However, it is very important to note that this condition is not necessary, i.e., there exist some unbounded functions (we consider the upper bound) such that the corresponding $\tilde{Z}$ is finite. An important example of such an unbounded function is the inverse CDF for the normal distribution (the transformation we considered above to obtain a Gaussian distribution of transformed rewards).

**Computing a partition function**    After applying a quantile transformation to the initial reward values, the distribution of the resulting quantile reward $\mathcal{R}_q$ is known; therefore, the corresponding partition function $Z_q$ can be calculated analytically. Moreover, starting from a quantile reward $\mathcal{R}_q$, one can apply any additional transformation $f(\mathcal{R}_q)$ to it while still preserving the ability to compute the corresponding partition function: the partition function corresponding to a specific transformation function $f$ can be computed as a simple integral (derivation in Appendix F):

$$\tilde{Z}(x) = \int_0^1 \exp\left(\frac{1}{\beta} f(t)\right) dt.$$

If the integral cannot be computed analytically, one can compute it numerically for each specific value of $\beta$ that one plans to use during the training as the value of $\tilde{Z}(x)$ is constant. We provide a table 4 with some transformation functions and their corresponding partition function values

**Scale of the $\beta$ hyperparameter**   $\beta$ is one of the most important hyperparameters when optimizing a KL-constrained RL objective (see Objective 1), and like other alignment methods, QRPO requires this parameter to be properly tuned. As can be seen in Objective 1, if the reward is multiplied by a factor $\alpha$, then $\beta$ should also be multiplied by $\alpha$ to preserve the "signal-to-regularization" ratio.

A feature of the quantile reward is that it normalizes the reward between $[0, 1]$, making $\beta$ independent of the original reward scale and transferable across domains. However, an additional transformation on top of the quantile can change the reward scale again. Thus, if a transformation function changes the typical reward scale by some factor, then, roughly speaking, $\beta$ should be scaled by the same factor. Hence, the typical range for $\beta$ depends on the transformation applied to the quantile. This is rather a rule of thumb, as the shape of the reward can also matter. This aspect is important to keep in mind when trying different transformation functions.

We can make a more rigorous observation regarding the correct scaling of $\beta$ in the case where the quantile transformation function $f$ is upper-bounded on the interval $[0, 1]$ (see Appendix K for the intuition). The key factor determining the relative scale of the transformed reward is the left derivative of the quantile transformation function $f$ at $x = 1$. Therefore, when comparing two upper-bounded quantile transformation functions, the ratio between their respective $\beta$ values should be approximately equal to the ratio of their left derivatives at $x = 1$. If both left first derivatives at $x = 1$ are equal to zero, move to higher orders until you find the first non-zero left derivative common to both functions. Let $k$ be that order. Then the appropriate $\beta$-scaling factor is the ratio of their $k$-th left derivatives. If the first non-zero left derivative appears at different orders for the two functions, no single constant can scale $\beta$ consistently.

## C   Statistical Consistency and Convergence Guarantees

In this section, we analyze the QRPO objective in Eq. (11) (and its practical variant Eq. (12)) through the lens of (i) exactness of the partition function, (ii) statistical properties of the empirical quantile reward, and (iii) optimization with stochastic gradients.

**Notation.**   Write the calibrated target as $s(x, y) := \mathcal{R}_q(x, y) - \beta \log Z_q$ and the model log–ratio as $g_\theta(x, y) := \beta \log \frac{\pi_\theta(y|x)}{\pi_{\text{ref}}(y|x)}$. The population QRPO loss is $\mathcal{L}(\theta) = \mathbb{E}\big[(s(x, y) - g_\theta(x, y))^2\big]$.

### C.1   Exactness of the partition function

Assume that, for each fixed $x$, the reference reward distribution $r = \mathcal{R}(x, y)$ for $y \sim \pi_{\text{ref}}(\cdot \mid x)$ is continuous (as with standard regression reward models). Then the quantile reward $\mathcal{R}_q(x, y) = F_{\text{ref}}(x, \mathcal{R}(x, y))$ is uniformly distributed on $[0, 1]$ for every $x$. Therefore the partition function is *analytically exact and $x$-independent*:

$$Z_q(x) = \mathbb{E}\left[\exp\left(\tfrac{1}{\beta}\mathcal{R}_q\right)\right] = \int_0^1 e^{t/\beta}\, dt = \beta\left(e^{1/\beta} - 1\right). \tag{17}$$

Hence QRPO introduces *no approximation or modeling bias* through $Z_q$.

**Remark (discrete rewards)**   When $\mathcal{R}$ takes a small number of values, $\mathcal{R}_q$ is no longer uniform; however $Z_q(x) = \sum_k p_k(x)\, e^{t_k/\beta}$ remains exactly computable as a finite expectation over the atoms $\{t_k\}$ with masses $\{p_k(x)\}$ (see App. F). Thus $Z_q$ is still exact.

**Practical constant used in Eq.** (12)   If one replaces $\beta \log Z_q$ by $\beta \log \beta + 1$ for stability, the calibration error is $\Delta_\beta = \beta \log Z_q - (\beta \log \beta + 1) = \beta \log(1 - e^{-1/\beta}) < 0$. For $\beta \le 0.1$, $|\Delta_\beta| \le 2\beta e^{-1/\beta} < 10^{-5}$, is negligible.

### C.2   Empirical quantile reward: unbiasedness and concentration

For each $x$, let $y_1, \ldots, y_n \sim \pi_{\text{ref}}(\cdot \mid x)$ be i.i.d. reference samples with rewards $r_i = \mathcal{R}(x, y_i)$, and define the empirical quantile estimator $\widehat{\mathcal{R}}_q(x, y) = \frac{1}{n} \sum_{i=1}^n \mathbf{1}\{r_i \le \mathcal{R}(x, y)\}$.

**Unbiasedness** For all $(x, y)$, $\mathbb{E}\left[\widehat{\mathcal{R}}_q(x, y) \mid x, y\right] = \mathcal{R}_q(x, y)$.

**Uniform consistency and finite-sample deviation** By Glivenko–Cantelli, $\sup_r |\widehat{F}_n(x, r) - F_{\text{ref}}(x, r)| \xrightarrow{\text{a.s.}} 0$ as $n \to \infty$. Moreover, the Dvoretzky–Kiefer–Wolfowitz inequality yields, for any $\varepsilon > 0$, the following distribution-free bound:

$$\Pr\left( |\widehat{\mathcal{R}}_q(x, y) - \mathcal{R}_q(x, y)| > \varepsilon \mid x \right) \leq 2 e^{-2n\varepsilon^2}.$$

**Expected error**

$$\text{Var}\left( \widehat{\mathcal{R}}_q(x, y) \mid x, y \right) = \frac{\mathcal{R}_q(x, y)\left(1 - \mathcal{R}_q(x, y)\right)}{n} \leq \frac{1}{4n}, \qquad \Rightarrow \mathbb{E}\left| \widehat{\mathcal{R}}_q - \mathcal{R}_q \right| \leq \frac{1}{2\sqrt{n}}.$$

Thus the only stochasticity introduced by QRPO is *zero-mean label noise* on the target, with sub-Gaussian tails (from the exponential bound) and $O(1/\sqrt{n})$ rate.

## C.3 Population argmin is unchanged by unbiased quantile noise

Let $\xi(x, y) := \widehat{\mathcal{R}}_q(x, y) - \mathcal{R}_q(x, y)$, so that $\mathbb{E}[\xi(x, y) \mid x, y] = 0$. Define $\widehat{s}(x, y) = s(x, y) + \xi(x, y)$, where $s(x, y) = \mathcal{R}_q(x, y) - \beta \log Z_q$ and $g_\theta(x, y) = \beta \log \frac{\pi_\theta(y|x)}{\pi_{\text{ref}}(y|x)}$. Then the population loss when training against $\widehat{s}$ is

$$\widehat{\mathcal{L}}(\theta) = \mathbb{E}\left[ \left( \widehat{s}(x, y) - g_\theta(x, y) \right)^2 \right] = \mathcal{L}(\theta) + \mathbb{E}\left[ \xi(x, y)^2 \right],$$

because the cross term vanishes: $\mathbb{E}\left[ \xi(x, y) \left( s(x, y) - g_\theta(x, y) \right) \right] = \mathbb{E}\left[ \mathbb{E}\left[ \xi(x, y) \mid x, y \right] (s - g_\theta) \right] = 0$.

**Oracle risk invariance** The set of minimizers of $\widehat{\mathcal{L}}(\theta)$ equals that of $\mathcal{L}(\theta)$. Unbiased quantile noise adds only a $\theta$–independent constant to the population risk.

**Gradient perturbation bound (per sample)** With the same $\xi(x, y)$ as above, the per-sample gradient difference is

$$\nabla_\theta \widehat{\ell} - \nabla_\theta \ell = 2 \xi(x, y) \nabla_\theta g_\theta(x, y).$$

If $\|\nabla_\theta g_\theta(x, y)\| \leq G$ (e.g., via gradient clipping), then for any $\varepsilon > 0$,

$$\Pr\left( \|\nabla_\theta \widehat{\ell} - \nabla_\theta \ell\| \leq 2G\varepsilon \mid x \right) \geq 1 - 2e^{-2n\varepsilon^2}.$$

Equivalently, with probability at least $1 - \delta$, $\|\nabla_\theta \widehat{\ell} - \nabla_\theta \ell\| \leq 2G \sqrt{\frac{1}{2n} \log \frac{2}{\delta}}$.

## C.4 Convergence of (stochastic) gradient descent

The QRPO loss is nonconvex in $\theta$, but the preceding results ensure that the *population* objective is unaffected by empirical-quantile noise. Assume (A1) the loss is lower-bounded and $L$-smooth in $\theta$; (A2) stochastic gradients have bounded second moment; (A3) either the empirical quantiles are resampled independently across iterations (or $n \to \infty$ so that $\xi \to 0$), or one optimizes the fixed empirical objective with standard SGD. Then:

1. With resampling (or $n \to \infty$), SGD produces gradients that are unbiased for $\nabla \mathcal{L}(\theta)$ and converges to a stationary point of the *oracle* loss $\mathcal{L}(\theta)$ under standard step-size conditions $\sum_t \eta_t = \infty, \ \sum_t \eta_t^2 < \infty$.

2. With fixed precomputed $\widehat{\mathcal{R}}_q$ (Algorithm 5), SGD converges to a stationary point of the *empirical* QRPO loss. By DKW and Glivenko–Cantelli, as $n \to \infty$ the empirical objective converges uniformly to the oracle objective, so the stationary points approach those of $\mathcal{L}(\theta)$.

**Summary.** (i) $Z_q$ is known in closed form (or as an exact finite sum in the discrete case), so QRPO introduces *no bias* via the partition function; the practical constant in Eq. (12) differs from $\beta \log Z_q$ by a negligible $O(\beta e^{-1/\beta})$ shift. (ii) The empirical quantile reward is an *unbiased* estimator with exponential concentration $e^{-2n\varepsilon^2}$ and expected error $O(1/\sqrt{n})$. (iii) This induces only zero-mean label noise, leaving the *population argmin* unchanged; SGD converges to a stationary point of the same oracle objective under standard assumptions.

# D The partition function is a moment generating function at $t = 1/\beta$

In this section, we show how to express a partition function $Z(x)$ in terms of a reward distribution for completions from the reference model. Assume that $\mathcal{R}(x, y)$ is the reward optimized in the objective 1, it can be an initial reward or a transformed one. Then, according to the definition of the partition function (Equation 2):

$$
\begin{aligned}
Z(x) &= \sum_y \pi_{\text{ref}}(y \mid x) \exp\left(\tfrac{1}{\beta}\mathcal{R}(x, y)\right) \\
&= \sum_r \sum_{y:\,\mathcal{R}(x,y)=r} \pi_{\text{ref}}(y \mid x) \exp\left(\tfrac{1}{\beta}\mathcal{R}(x, y)\right) \quad \textit{\scriptsize [Stratify the summation of y by possible reward values, see discussion below regarding the validity of this reordering]} \\
&= \sum_r \sum_{y:\,\mathcal{R}(x,y)=r} \pi_{\text{ref}}(y \mid x) \exp\left(\tfrac{1}{\beta}r\right) \quad \textit{\scriptsize [Change $\mathcal{R}(x,y) \to r$ since all y's in the inner sum have reward $\mathcal{R}(x,y)=r$]} \\
&= \sum_r \exp\left(\tfrac{1}{\beta}r\right) \sum_{y:\,\mathcal{R}(x,y)=r} \pi_{\text{ref}}(y \mid x) \quad \textit{\scriptsize [Pull $\exp\left(\tfrac{1}{\beta}r\right)$ out of the inner sum as it doesn't depend on y]} \\
&= \sum_r \exp\left(\tfrac{1}{\beta}r\right) \underbrace{\Pr_{y\sim\pi_{\text{ref}}(\cdot|x)}\{\mathcal{R}(x, y) = r\}}_{P_{\text{ref}}(r|x)} \quad \textit{\scriptsize [The inner sum becomes a cumulative probability of all y's with the reward $r$ $\Rightarrow$ it's a probability of obtaining the reward r]} \\
&= \sum_r \exp\left(\tfrac{1}{\beta}r\right) P_{\text{ref}}(r \mid x) \\
&= \mathbb{E}_{r\sim P_{\text{ref}}(\cdot|x)}\left[\exp\left(\tfrac{1}{\beta}r\right)\right] = M_r\left(t = \frac{1}{\beta}\right). \quad \textit{\scriptsize [Use the definition of MGF]}
\end{aligned}
$$

Here we use a notation $P_{ref}(r|x)$ that represents a probability of obtaining a reward $r$ when sampling a completion $y$ for a reference model $\pi_{ref}(y \mid x)$ and computing the corresponding reward $\mathcal{R}(x, y)$.

In the last line, we applied a definition of a moment generating function (MGF) $M_X(t) = \mathbb{E}_X[\exp(tX)]$.

**Note on the validity of the summation reordering** Although the sets of possible $y$ and $r$ can be infinite, the reordering of the summation is legitimate for the following reasons. A language model by definition can generate only finite-length strings, with generation halting with probability 1 (a property often referred to as the *tightness* of a language model). Consequently, although the sets of possible $y$ and $r$ are infinite, they are countable. Moreover, since all terms in the sum are non-negative, the series converges absolutely, and the reordering of terms is therefore justified.

We derived an expression for the partition function that depends only on the reward distribution and is independent of the probability distribution over completions. This result greatly simplifies both the analysis and computation of the partition function. Specifically, if the reward distribution is known and the moment generating function (MGF) is finite at the chosen value of $\beta$, the partition function can always be computed analytically or numerically. Conversely, if the partition function is known as a function of a variable $\beta$ and after the change of variables $t = \frac{1}{\beta}$ is still defined on an open interval containing $t = 0$, then the reward distribution is uniquely determined (E.g., using the inverse Laplace transform) and can be recovered analytically or numerically.

This leads to an important insight: in practice, knowing the reward distribution implies knowing the partition function (provided it is finite), and vice versa. Therefore, it is not possible to obtain an analytical expression for the partition function if the reward distribution can be arbitrary. This insight is central to the QRPO method, which uses a quantile transformation to ensure the reward distribution is known and thus the partition function becomes analytically tractable.

# E    Distribution of the quantile reward $\mathcal{R}_q$

Let $y \sim \pi_{\text{ref}}(\cdot \mid x)$. We want to show that the random variable

$$\mathcal{R}_q(x, y) = \Pr_{y' \sim \pi_{ref}(\cdot|x)} \{\mathcal{R}(x, y') \leq \mathcal{R}(x, y)\}$$

is uniformly distributed on $[0, 1]$ for $y \sim \pi_{ref}(\cdot \mid x)$ in case $\mathcal{R}(x, y)$ can be treated as a continuous variable. This is a well-known result; we include it for the completeness of the theoretical part.

The cumulative distribution function (CDF) of rewards under the reference policy (Eq. 8) is

$$F_{\text{ref}}(x, r) = \Pr_{y' \sim \pi_{\text{ref}}(\cdot|x)} \{\mathcal{R}(x, y') \leq r\}.$$

Hence, the quantile reward (Eq. 9) is also

$$\mathcal{R}_q(x, y) = F_{\text{ref}}(x, \mathcal{R}(x, y)).$$

**Proof intuition**    There is a simple intuition to see that $\mathcal{R}_q(x, y)$ is uniformly distributed for $y \sim \pi_{ref}(\cdot \mid x)$ by thinking about its cumulative distribution. We pick a value $\alpha \in [0, 1]$ and ask what is the probability that $\mathcal{R}_q(x, y) \leq \alpha$, but since $\mathcal{R}_q(x, y) = F_{\text{ref}}(x, \mathcal{R}(x, y))$, this is equivalent to asking what is the probability to cumulate $\alpha$ probability under the reference reward distribution, which is by definition $\alpha$.

**Formally**    Assume that, for the fixed input $x$, the map $r \mapsto F_{\text{ref}}(x, r)$ is continuous and strictly increasing (i.e., $\mathcal{R}(x, y)$ has a continuous distribution). For any $\alpha \in [0, 1]$,

$$\begin{aligned}
\Pr\{\mathcal{R}_q(x, y) \leq \alpha\} &= \Pr\{F_{\text{ref}}(x, \mathcal{R}(x, y)) \leq \alpha\} \\
&= \Pr\{\mathcal{R}(x, y) \leq F_{\text{ref}}^{-1}(x, \alpha)\} \\
&= F_{\text{ref}}(x, F_{\text{ref}}^{-1}(x, \alpha)) \\
&= \alpha,
\end{aligned}$$

which is the CDF of $\text{Uniform}(0, 1)$. Consequently,

$$\mathcal{R}_q(x, y) \sim \text{Uniform}(0, 1) \quad \text{for } y \sim \pi_{\text{ref}}(\cdot \mid x).$$

As an illustration, we show in Figure 8 an empirical distribution of the quantile reward.

**Figure 8:** Empirical distribution of the quantile reward $\mathcal{R}_q(x, y)$ for completions generated from a reference model $\pi_{ref}$.

# F  Partition function for the quantile reward $\mathcal{R}_q$ and the generalized transformed reward $\tilde{\mathcal{R}}$

Here we compute the partition function $\tilde{Z}$ for the case of a generalized transformed reward $\tilde{\mathcal{R}}(x,y) = f(\mathcal{R}_q(x,y))$ (as introduced in Equation 13). To get $Z_q$ for the quantile reward $\mathcal{R}_q$, one can just take $f(t) = t$.

We use the same notation as in Equation 9:

$$\mathcal{R}_q(x,y) = \Pr_{y' \sim \pi_{ref}(\cdot|x)} \{\mathcal{R}(x,y') \leq \mathcal{R}(x,y)\} = F_{ref}(x, \mathcal{R}(x,y)),$$

$$\tilde{\mathcal{R}}(x,y) = f(\mathcal{R}_q(x,y))$$

for any function $f$ defined on the interval $[0,1]$.

Following the results obtained in Appendix D, we can express the partition function in terms of the moment generating function (MGF) for the generalized transformed reward $\tilde{\mathcal{R}}(x,y)$ distribution for $y \sim \pi_{ref}(\cdot \mid x)$:

$$\tilde{Z} = \mathbb{E}_{\tilde{r} \sim \tilde{P}_{\mathrm{ref}}(\cdot|x)} \left[\exp\left(\tfrac{1}{\beta}\tilde{r}\right)\right] = M_{\tilde{r}}\left(t = \frac{1}{\beta}\right),$$

where, similar to Appendix D, we introduce $\tilde{P}_{\mathrm{ref}}(\cdot \mid x)$—the probability distribution of the generalized transformed reward $\tilde{\mathcal{R}}(x,y)$ induced by $y \sim \pi_{ref}(\cdot \mid x)$.

If one knows the distribution of the transformed reward, the partition function can be computed directly by evaluating its MGF at $t = 1/\beta$. For instance, in Appendix E, we show that the quantile reward $\mathcal{R}_q(x,y)$, when treated as a continuous variable, follows a $\mathrm{Uniform}(0,1)$ distribution. The MGF for $\mathrm{Uniform}(0,1)$ distribution is given by $M_{r_q}(t) = \tfrac{1}{t}(e^t - 1)$, and thus the corresponding partition function is

$$Z_q = M_{r_q}\left(t = \frac{1}{\beta}\right) = \beta(e^{\frac{1}{\beta}} - 1).$$

**More generally, when rewards can be approximated by a continuous distribution**  Note that, generally speaking, when the reward can be treated as a continuous variable (such as in the case of a reward model or a coding test-case pass rate), the distribution of the generalized transformed reward $\tilde{\mathcal{R}}$ can, in principle, always be computed. This is because the distribution of $\mathcal{R}_q$ is known, and $\tilde{\mathcal{R}}$ is defined as a known function $f$ of $\mathcal{R}_q$. Moreover, it is not even necessary to explicitly derive this distribution in order to compute the partition function; instead, one can directly apply the change-of-variables formula for expectations (also known as the "Law of the Unconscious Statistician" (LOTUS)):

$$\tilde{Z} = \mathbb{E}_{\tilde{r} \sim \tilde{P}_{\mathrm{ref}}(\cdot|x)} \left[\exp\left(\tfrac{1}{\beta}\tilde{r}\right)\right] = \mathbb{E}_{r_q \sim \mathrm{U}[0,1]} \left[\exp\left(\tfrac{1}{\beta}f(r_q)\right)\right] = \int_0^1 \exp\left(\tfrac{1}{\beta}f(t)\right) dt.$$

We obtained a 1-D integral that can be computed for any transformation function $f$ (either analytically or numerically for some specific values of $\beta$). However, note that QRPO requires this integral to be finite, thus introducing some basic restrictions for the transformation function $f$.

**If the rewards cannot be well-approximated by a continuous distribution, we directly use the discrete distribution of the rewards**  This situation typically arises when the reward distribution has only a small number of possible values. In this case, the expression for the partition function cannot be computed in the same way as in the continuous case because we can no longer rely on the fact that the quantile reward $\mathcal{R}_q$ follows a uniform distribution. Consequently, the chain of reasoning used in the continuous setting—starting from the known distribution of $\mathcal{R}_q$, then obtaining the distribution of $\tilde{\mathcal{R}}$, and finally computing the exact expression for the partition function—breaks down.

Instead, we need to explicitly approximate or estimate the distribution of rewards. Specifically, we need to know the distribution of either the initial reward $\mathcal{R}$, the quantile reward $\mathcal{R}_q$, or the generalized transformed reward $\tilde{\mathcal{R}}$. In practice, it is usually easiest to work with the distribution of the initial reward $\mathcal{R}$.

The good news is that in such discrete settings with a small number of possible values, the distribution of the initial rewards can often be estimated much more easily and reliably, for example by simple empirical counting of occurrences in a finite sample. Once we have estimated probabilities for the initial reward values $\{\hat{P}_{\text{ref}}(r_i \mid x)\}_i$ for each prompt by sampling from the reference model $y \sim \pi_{ref}(\cdot \mid x)$, we can, similarly to the continuous case, apply the change-of-variables formula for expectations to compute the partition function:

$$\tilde{Z}(x) = \mathbb{E}_{\tilde{r} \sim \tilde{P}_{\text{ref}}(\cdot|x)} \left[ \exp\left( \tfrac{1}{\beta} \tilde{r} \right) \right] = \mathbb{E}_{r \sim P_{ref}(\cdot|x)} \left[ \exp\left( \tfrac{1}{\beta} f(F_{ref}(x, r))) \right) \right]$$

$$\approx \sum_{i=1}^{K} \hat{P}_{ref}(r_i \mid x) \exp\left( \frac{1}{\beta} f\left( \sum_{r_j \leq r_i} \hat{P}_{ref}(r_j \mid x) \right) \right).$$

Here, $P_{\text{ref}}(\cdot \mid x)$ denotes the probability distribution of the initial reward $\mathcal{R}(x, y)$ induced by $y \sim \pi_{ref}(\cdot \mid x)$, as introduced in Appendix D and similar to $\tilde{P}_{\text{ref}}(\cdot \mid x)$ introduced earlier in this section; $\hat{P}_{\text{ref}}(\cdot \mid x)$ is an empirical estimate for $P_{\text{ref}}(\cdot \mid x)$; $F_{ref}(x, r)$ is a CDF of initial rewards under the reference policy as introduced in Definition 8. $K$ denotes the number of possible initial reward values.

The expression for the discrete case derived above is intended to be used in the case when the number of possible reward values is small (i.e. $< 10$). For other cases, the continuous version is likely to be a decent approximation for the partition function, and is more preferable, since it does not require one to estimate the probabilities $\hat{P}_{\text{ref}}(r_i \mid x)_i$.

## G  Sufficient conditions for the function in the reward transformation for the existence of a finite partition function

We derive a sufficient condition on the function $f$ in the proposed reward transformation (Eq. 13) that ensures the finiteness of the corresponding partition function. The partition function is given by:

$$\tilde{Z}(x) = \sum_y \pi_{ref}(y \mid x) \exp\left( \frac{1}{\beta} \tilde{\mathcal{R}}(x, y) \right).$$

Recall that the partition function can be expressed as a moment generating function for the distribution of the reward used in the objective 1 (see Appendix D for the derivation). Here we consider objective 1 with the generalized transformed reward $\tilde{\mathcal{R}}$:

$$\tilde{Z} = \mathbb{E}_{\tilde{r} \sim \tilde{P}_{\text{ref}}(\cdot|x)} \left[ \exp\left( \tfrac{1}{\beta} \tilde{r} \right) \right],$$

where $\tilde{P}_{\text{ref}}(r \mid x)$ denotes the probability distribution of the generalized transformed reward $\tilde{\mathcal{R}}(x, y)$ induced by the reference policy $y \sim \pi_{ref}(\cdot \mid x)$.

By applying the change-of-variables formula for expectations (also known as the "Law of the Unconscious Statistician" (LOTUS)) we obtain:

$$\tilde{Z} = \mathbb{E}_{\tilde{r} \sim \tilde{P}_{\text{ref}}(\cdot|x)} \left[ \exp\left( \tfrac{1}{\beta} \tilde{r} \right) \right] = \mathbb{E}_{r \sim P_{ref}(\cdot|x)} \left[ \exp\left( \tfrac{1}{\beta} f\left( F_{\text{ref}}(x, r) \right) \right) \right]$$

$$= \sum_r \exp\left( \tfrac{1}{\beta} f\left( F_{\text{ref}}(x, r) \right) \right) P_{\text{ref}}(r \mid x),$$

where $P_{\text{ref}}(r \mid x)$ denotes the probability distribution of the initial reward $\mathcal{R}(x, y)$ induced by the reference policy $y \sim \pi_{ref}(\cdot \mid x)$.

Then we can derive a very simple sufficient criterion for the finite partition function existence:

$$\tilde{Z}(x) = \sum_r \exp\left(\tfrac{1}{\beta} f\big(F_{\text{ref}}(x, r)\big)\right) P_{\text{ref}}(r \mid x) \leq \sum_r \exp\left(\tfrac{1}{\beta} \max_{u^* \in [0,1]} f(u^*)\right) P_{\text{ref}}(r \mid x)$$

$$= \exp\left(\frac{1}{\beta} \max_{u^* \in [0,1]} f(u^*)\right) < \infty,$$

if the function $f$ is upper-bounded on the interval $[0, 1]$.

**Other transformations**  The condition for a transformation function to be upper-bounded is sufficient for a finite partition function existence, but is not necessary. Thus, one can also consider other transformation functions that are not upper-bounded on the interval $[0, 1]$. However, this would require a more careful consideration of the finiteness conditions of a partition function.

## H  Noise tradeoff in the calibrated target: estimating the partition function vs. estimating the quantile reward

**Figure 9:** The noise magnitude in the regression objective in a log-scale when the reward is left unchanged and $Z(x)$ is estimated directly (initial calibrated target) vs. when the quantile reward is estimated (modified calibrated target) and $Z(x)$ is computed analytically. The $x$-axis represents the true quantile of the reward value relative to the reward distribution of a reference model. We use $n = 10$ samples to estimate both the values of the empirical partition function and the transformed reward in this plot.

**Noise amplitude and intractability of partition function estimation for the initial reward**  Direct regression methods described above require estimating a partition function $Z$ that is computationally very expensive. To support this empirical observation, we provide a theoretical analysis of the noise amplitude in its empirical estimate, assuming that completion rewards for a given prompt follow a Gaussian distribution. The standard deviation of a logarithm of an empirical estimate of partition function $\hat{Z}$ has the following expression for a large number of samples $n \gg e^{\frac{\sigma^2}{\beta^2}}$ (derivation in Appendix I):

$$\text{std}(\beta \log \hat{Z}(x)) = \frac{\beta}{\sqrt{n}} \sqrt{e^{\frac{\sigma^2}{\beta^2}} - 1} \tag{18}$$

where $n$ is the number of samples that we use for the estimation of $Z(x)$ for each prompt $x$, $\sigma^2$ is a variance of the reward $\mathcal{R}(x, y)$ for $y \sim \pi_{ref}(\cdot \mid x)$, and $\beta$ is the parameter from the RL fine-tuning objective. The derived formula shows that, for example, when $\sigma = 1$ and $\beta = 0.1$, approximately $n = e^{100} \sim 10^{40}$ samples per prompt $x$ are required to achieve a reasonable signal-to-noise ratio in the estimated partition function that is an entirely intractable number. In fact, while the actual reward distribution may not be perfectly Gaussian in practice, any distribution with a right tail will face the same issue due to the $\exp(\tfrac{1}{\beta}\mathcal{R})$ term in the partition function expression. A good workaround would be to compute $Z(x)$ analytically, but this requires us to know the exact distribution of the reward. Hence, we need a transformation for the reward that leads to a known final reward distribution.

**Noise amplitude in the regression objective**  We showed that the regression objective in Equation 6 suffers from high noise due to the estimation of the partition function $\hat{Z}$. By introducing a transformed reward, we have completely eliminated this noise in the partition function by using an exact expression computed analytically. However, some noise still remains in the regression objective, now originating from the transformed reward $\mathcal{R}_q(x, y)$ itself, as it requires estimating a quantile.

To assess the impact of this change, we compare the noise level in the objective 6 after applying the transformation with the initial noise level. The standard deviation of $\beta \log \hat{Z}(x)$ for the initial reward was previously shown in Equation 18. The standard deviation of $\mathcal{R}_q(x, y)$ is given by the following expression (proof in Appendix J):

$$\text{std}\left(\mathcal{R}_q(x, y)\right) = \frac{1}{\sqrt{n}} \sqrt{\mathbb{E}\left[\mathcal{R}_q(x, y)\right]\left(1 - \mathbb{E}\left[\mathcal{R}_q(x, y)\right]\right)}.$$

Note that the standard deviation of the transformed reward is not constant and depends on the initial reward value through its exact quantile, $\mathbb{E}\left[\mathcal{R}_q(x, y)\right]$. Figure 9 compares the noise magnitude before and after the reward transformation. As shown, the noise in the initial reward is significantly higher, even for moderate values of $\beta$. This is particularly relevant given that typical $\beta$ values in RL fine-tuning methods are around 0.1 or lower.

This theoretical analysis demonstrates that the proposed approach of introducing a reward transformation reduces the noise in the objective function dramatically[4], which enables the general approach of pointwise policy fitting to work well in practice.

# I  Noise amplitude in the partition function estimate for the initial reward

Here we compute the variance of the $\log \hat{Z}(x)$ estimate for the general case of using the initial reward for the policy fitting regression objective. Recall that the partition function can be expressed as a moment generating function for the distribution of the reward used in the objective 1 (see Appendix D for the derivation):

$$Z = \mathbb{E}_{r \sim P_{\text{ref}}(\cdot | x)}\left[\exp\left(\tfrac{1}{\beta} r\right)\right],$$

where $P_{\text{ref}}(r \mid x)$ denotes the probability distribution of the initial reward $\mathcal{R}(x, y)$ induced by the reference policy $y \sim \pi_{ref}(\cdot \mid x)$.

Therefore, since the exact distribution $P_{ref}(r|x)$ is unknown, we cannot compute $Z(x)$ analytically. In the remainder of this section, we describe a Monte-Carlo approach for estimating $\log Z(x)$. Specifically, we analyze the variance (noise amplitude) of this estimate to assess the feasibility of the approach.

We estimate the variance of $\log Z(x)$ by modeling the process of generating $n$ completions from the reference model and then approximating $Z(x)$ with a finite sum. To do this analytically, we assume that $P_{ref}(r|x)$ is Gaussian, but this does not reduce the generality of our conclusions, since it can be shown that any distribution with an infinite right tail has the same properties or worse.

We can also assume that $P_{ref}(r|x)$ is Gaussian with zero mean since:

$$\log Z(x) = \log \mathbb{E}_{r \sim \mathcal{N}(\mu(x), \sigma^2(x))}\left[\exp\left(\frac{1}{\beta} r\right)\right] = \frac{\mu(x)}{\beta} + \log \mathbb{E}_{r \sim \mathcal{N}(0, \sigma^2(x))}\left[\exp\left(\frac{1}{\beta} r\right)\right],$$

so it is clear that the variance of estimated $\log \hat{Z}(x)$ does not depend on $\mu(x)$.

Now suppose we have sampled $n$ completions from a reference model and obtained $n$ samples $r_i \sim \mathbb{N}(0, \sigma^2)$. Then $\exp\left(\frac{1}{\beta} r_i\right) \sim \texttt{Lognormal}(0, \frac{\sigma^2}{\beta^2})$ and

$$\mathbb{E}\left[\exp\left(\frac{1}{\beta} r_i\right)\right] = e^{\frac{\sigma^2}{2\beta^2}}, \quad \text{var}\left(\exp\left(\frac{1}{\beta} r_i\right)\right) = \left(e^{\frac{\sigma^2}{\beta^2}} - 1\right) e^{\frac{\sigma^2}{\beta^2}}.$$

---

[4]At least for rewards that can be treated as continuous. We have derived QRPO for reward distributions with a small number of possible reward values for completeness, but this is not the focus of our work, and the noise tradeoff in this case remains an open question.

Unfortunately, the distribution of the sum of `Lognormal` variables has no explicit expression. Therefore, we will operate with the variance.

Then the estimator $\hat{Z}(x) = \frac{1}{n}\sum_{i=1}^{n}\exp\left(\frac{1}{\beta}r_i\right)$ have the variance

$$\text{var}\left(\hat{Z}(x)\right) = \text{var}\left(\frac{1}{n}\sum_{i=1}^{n}\exp\left(\frac{1}{\beta}r_i\right)\right) = \sum_{i=1}^{n}\frac{\text{var}\left(\exp\left(\frac{1}{\beta}r_i\right)\right)}{n^2} = \frac{1}{n}\left(e^{\frac{\sigma^2}{\beta^2}}-1\right)e^{\frac{\sigma^2}{\beta^2}}.$$

The last step is to calculate the variance of $\log \hat{Z}(x)$. Unfortunately, since there is no explicit expression for the distribution of $\hat{Z}(x)$, we cannot rigorously compute the variance of $\log \hat{Z}(x)$. Furthermore, the knowledge of $\text{var}\left(\hat{Z}(x)\right)$ is also insufficient to rigorously calculate the variance of $\log \hat{Z}(x)$. However, we can approximate it for large enough $n$. Indeed:

$$\frac{\mathbb{E}\left[\hat{Z}(x)\right]}{\text{std}\left(\hat{Z}(x)\right)} = \frac{e^{\frac{\sigma^2}{2\beta^2}}}{\sqrt{\frac{1}{n}\left(e^{\frac{\sigma^2}{\beta^2}}-1\right)e^{\frac{\sigma^2}{\beta^2}}}} = \frac{\sqrt{n}}{\sqrt{e^{\frac{\sigma^2}{\beta^2}}-1}},$$

then for $n \gg e^{\frac{\sigma^2}{\beta^2}}$ the distribution of $\hat{Z}(x)$ is narrow and we can use a linear approximation for $\log x$ at the point $x = \mathbb{E}\left[\hat{Z}(x)\right] = e^{\frac{\sigma^2}{2\beta^2}}$, that is, $\log x \approx e^{-\frac{\sigma^2}{2\beta^2}}x + \frac{\sigma^2}{2\beta^2} - 1$. Then finally

$$\text{var}\left(\log \hat{Z}(x)\right) \approx \text{var}\left(e^{-\frac{\sigma^2}{2\beta^2}}\hat{Z}(x) + \frac{\sigma^2}{2\beta^2} - 1\right) = e^{-\frac{\sigma^2}{\beta^2}}\text{var}\left(\hat{Z}(x)\right) = \frac{1}{n}\left(e^{\frac{\sigma^2}{\beta^2}}-1\right),$$

$$\text{std}(\log \hat{Z}(x)) = \frac{1}{\sqrt{n}}\sqrt{e^{\frac{\sigma^2}{\beta^2}}-1}.$$

## J  Noise amplitude in the transformed reward estimate

In this section, we calculate the variance for the proposed modified reward estimate $\mathcal{R}_q(x,y)$.

Suppose we have $n$ completions $y_i$ from the reference model $\pi_{ref}(\cdot \mid x)$ and the corresponding rewards $r_{ref_i} = \mathcal{R}(x,y_i)$. Then the estimate $\mathcal{R}_q(x,y)$ is computed as

$$\mathcal{R}_q(x,y) = \frac{1}{n}\sum_{i=1}^{n}\mathbb{1}\{r_i \le \mathcal{R}(x,y)\}.$$

$\mathbb{1}\{r_i \le \mathcal{R}(x,y)\}$ is a Bernoulli random variable with the variance

$$\text{var}\left(\mathbb{1}\{r_i \le \mathcal{R}(x,y)\}\right) = F_{ref}(x,\mathcal{R}(x,y))(1 - F_{ref}(x,\mathcal{R}(x,y))),$$

where $F_{ref}(x)$ is a cumulative distribution function (CDF) for the distribution of the initial reward under the reference policy (see definition 8). Then

$$\text{var}\left(\mathcal{R}_q(x,y)\right) = \frac{1}{n}F_{ref}(x,\mathcal{R}(x,y))(1 - F_{ref}(x,\mathcal{R}(x,y))),$$

$$\text{std}\left(\mathcal{R}_q(x,y)\right) = \frac{1}{\sqrt{n}}\sqrt{F_{ref}(x,\mathcal{R}(x,y))(1 - F_{ref}(x,\mathcal{R}(x,y)))},$$

Noticing that

$$\mathbb{E}\left[\mathcal{R}_q(x,y)\right] = F_{ref}(x,\mathcal{R}(x,y)),$$

we can finally rewrite the std $\left(\mathcal{R}_q(x,y)\right)$ in the following way:

$$\text{std}\left(\mathcal{R}_q(x,y)\right) = \frac{1}{\sqrt{n}}\sqrt{\mathbb{E}\left[\mathcal{R}_q(x,y)\right]\left(1 - \mathbb{E}\left[\mathcal{R}_q(x,y)\right]\right)}.$$

# K Equivalence of all quantile transformation functions strictly increasing and upper-bounded on the interval $[0, 1]$

In this section, we show that when $\beta$ is sufficiently small and the initial reward $\mathcal{R}$ can be approximated as a continuous variable, the optimal policy for a transformed reward $\tilde{\mathcal{R}}(x, y) = f\left(\mathcal{R}_q(x, y)\right)$ remains invariant under any choice of a strictly increasing continuous transformation function $f(t)$ defined on $[0, 1]$ with finite values and a non-zero left derivative at $t = 1$.[5] Formally, as $\beta$ decreases, the policy converges to the same solution regardless of which function $f$ in this family is used.

We assume mild regularity conditions, specifically $f \in C^\infty[0, 1]$.

Recall that the optimal policy 2 has the form:

$$\pi^*(y \mid x) = \frac{1}{Z(x)} \pi_{ref}(y \mid x) \exp\left(\frac{1}{\beta}\tilde{\mathcal{R}}(x, y)\right),$$

where

$$\tilde{\mathcal{R}}(x, y) = f\left(\mathcal{R}_q(x, y)\right),$$

$$Z(x) = \sum_y \pi_{ref}(y \mid x) \exp\left(\frac{1}{\beta}\tilde{\mathcal{R}}(x, y)\right).$$

We start with some standardization of the initial conditions.

## K.1 Rescale and shift

Under our assumptions:

$$f(1) = C_1 < \infty, \ C_1 > -\infty,$$
$$f'_-(1) = C_2 < \infty,$$

where $f'_-$ denotes a left derivative.

Also since $f(t)$ is strictly increasing on the interval $[0, 1]$ and its left derivative has non-zero value at $t = 1$:

$$C_2 > 0.$$

First, note that any constant shift of the function $f$ does not change the optimal policy $\pi^*(y \mid x)$. Indeed, if there is a shift $f(t) \mapsto f(t) + \alpha$ then

$$\exp\left(\frac{1}{\beta}f\left(\mathcal{R}_q(x, y)\right)\right) \mapsto \exp\left(\frac{\alpha}{\beta}\right) \cdot \exp\left(\frac{1}{\beta}f\left(\mathcal{R}_q(x, y)\right)\right),$$

$$Z(x) \mapsto \exp\left(\frac{\alpha}{\beta}\right) \cdot Z(x),$$

and the terms $\exp\left(\frac{\alpha}{\beta}\right)$ cancel each other. Hence, we can do the following transformation without changing the optimal policy:

$$f(t) \mapsto f(t) - C_1,$$

obtaining the following initialization property w.l.o.g.:

$$f(1) = 0.$$

Second, note that for any constant $\alpha$, a transformation $f(t) \mapsto \alpha f(t)$ together with $\beta \mapsto \alpha\beta$ also does not change the optimal policy. Hence, we can simultaneously transform $f(t)$ and $\beta$ in the following way without changing the problem formulation:

$$f(t) \mapsto \frac{f(t)}{C_2}, \quad \beta \mapsto \frac{\beta}{C_2}.$$

---

[5]Functions with zero left derivative at $t = 1$ correspond to another equivalence class. More formally, it can be shown that there exist classes of equivalence. The $n$-th class consists of functions whose first $n$ left derivatives are all zero at $t = 1$ but have a non-zero $n + 1$ left derivative at $t = 1$. We make a derivation only for the case of $n = 0$, i.e., non-zero first left derivative. Others can be done similarly.

After this transformation, we obtain a useful initialization property that we will use in our proof w.l.o.g:

$$f'_-(1) = 1.$$

Intuitively, this scaling argument means that if two different reward transformation functions differ only in their scale near $x = 1$, one can compensate by adjusting $\beta$. This rescaling ensures comparability of different transformation functions and does not change the essence of our invariance claim: although different scales of $f$ near $x = 1$ may alter the rate at which the policy converges, they do not affect the limiting policy itself.

Throughout the following proof, whenever $f$ does not already satisfy $f(1) = 0$ or $f'_-(1) = 1$, we perform these shift and rescaling steps first to normalize its value and derivative at $x = 1$.

## K.2  Main proof

To prove the invariance of the limiting optimal policy under any choice of the transformation function $f$ for small enough $\beta$ we show that a KL divergence between the optimal policies corresponding to any two transformation functions $f$ and $g$ among the considered family of reward transformations decreases linearly with $\beta \to 0$.

We consider the following optimal policies:

$$\pi_f^*(y \mid x) = \frac{1}{Z_f(x)} \pi_{ref}(y \mid x) \exp\left(\frac{1}{\beta} f\left(\mathcal{R}_q(x, y)\right)\right),$$

$$\pi_g^*(y \mid x) = \frac{1}{Z_g(x)} \pi_{ref}(y \mid x) \exp\left(\frac{1}{\beta} g\left(\mathcal{R}_q(x, y)\right)\right),$$

where

$$Z_f(x) = \sum_y \pi_{ref}(y \mid x) \exp\left(\frac{1}{\beta} f\left(\mathcal{R}_q(x, y)\right)\right),$$

$$Z_g(x) = \sum_y \pi_{ref}(y \mid x) \exp\left(\frac{1}{\beta} g\left(\mathcal{R}_q(x, y)\right)\right),$$

$$\mathcal{R}_q(x, y) = F_{ref}\left(x, \mathcal{R}(x, y)\right),$$

where $F_{ref}(x)$ is a cumulative distribution function (CDF) for the distribution of the initial reward under the reference policy (see definition 8).

$$D_{KL}(\pi_f^*(\cdot \mid x) \,\|\, \pi_g^*(\cdot \mid x))$$

$$= \sum_y \frac{1}{Z_f(x)} \pi_{ref}(y \mid x) \exp\left(\frac{1}{\beta} f\left(\mathcal{R}_q(x, y)\right)\right) \log \frac{\frac{1}{Z_f(x)} \pi_{ref}(y \mid x) \exp\left(\frac{1}{\beta} f\left(\mathcal{R}_q(x, y)\right)\right)}{\frac{1}{Z_g(x)} \pi_{ref}(y \mid x) \exp\left(\frac{1}{\beta} g\left(\mathcal{R}_q(x, y)\right)\right)}$$

$$= \sum_r \sum_{y:\mathcal{R}(x,y)=r} \frac{1}{Z_f(x)} \pi_{ref}(y \mid x) \exp\left(\frac{1}{\beta} f\left(F_{ref}(x, r)\right)\right)$$

$$\times \left\{ \frac{1}{\beta} f\left(F_{ref}(x, r)\right) - \frac{1}{\beta} g\left(F_{ref}(x, r)\right) + \log \frac{Z_g(x)}{Z_f(x)} \right\}$$

$$= \sum_r \frac{1}{Z_f(x)} \exp\left(\frac{1}{\beta} f\left(F_{ref}(x, r)\right)\right) \left\{ \frac{1}{\beta} f\left(F_{ref}(x, r)\right) - \frac{1}{\beta} g\left(F_{ref}(x, r)\right) + \log \frac{Z_g(x)}{Z_f(x)} \right\}$$

$$\times \underbrace{\sum_{y:\mathcal{R}(x,y)=r} \pi_{ref}(y \mid x)}_{P_{ref}(r|x)}$$

$$= \sum_r \frac{1}{Z_f(x)} \exp\left(\frac{1}{\beta} f\left(F_{ref}(x, r)\right)\right) \left\{ \frac{1}{\beta} f\left(F_{ref}(x, r)\right) - \frac{1}{\beta} g\left(F_{ref}(x, r)\right) + \log \frac{Z_g(x)}{Z_f(x)} \right\}$$

$$\times P_{ref}(r|x),$$

where $P_{\text{ref}}(r \mid x)$ denotes the probability distribution of the initial reward $\mathcal{R}(x, y)$ induced by the reference policy $y \sim \pi_{ref}(\cdot \mid x)$.

Recall that we require the distribution of the initial reward $r$ to admit a continuous approximation (such as in the case of a reward model or a coding test-case pass rate). We denote the probability density function of this continuous approximation as $p_{ref}(r \mid x)$. Then the derived sum can be turned into an integral:

$$
D_{KL}(\pi_f^*(\cdot \mid x) \,||\, \pi_g^*(\cdot \mid x))
$$

$$
= \int_{\mathbb{R}} \frac{1}{Z_f(x)} \exp\left(\frac{1}{\beta} f\left(F_{ref}(x, r)\right)\right) \quad \left\{\frac{1}{\beta} f\left(F_{ref}(x, r)\right) - \frac{1}{\beta} g\left(F_{ref}(x, r)\right) + \log \frac{Z_g(x)}{Z_f(x)}\right\}
$$

$$
\times\, p_{ref}(r \mid x) dr
$$

$$
= \int_{\mathbb{R}} \frac{1}{Z_f(x)} \exp\left(\frac{1}{\beta} f\left(F_{ref}(x, r)\right)\right) \quad \left\{\frac{1}{\beta} f\left(F_{ref}(x, r)\right) - \frac{1}{\beta} g\left(F_{ref}(x, r)\right) + \log \frac{Z_g(x)}{Z_f(x)}\right\}
$$

$$
\times\, dF_{ref}(x, r)
$$

$$
= \int_0^1 \frac{1}{\beta} \frac{1}{Z_f(x)} \exp\left(\frac{1}{\beta} f\left(z\right)\right) \quad \left\{f\left(z\right) - g\left(z\right) + \beta \log \frac{Z_g(x)}{Z_f(x)}\right\}
$$

$$
\times\, dz
$$

Since we require $f$ to be a strictly increasing smooth function defined on $[0, 1]$ with finite value and non-zero left derivative at $t = 1$ we can apply Laplace's method for an endpoint maximum to compute the asymptotics in terms of beta.

Here is the Laplace method formulation for reference:

**Theorem** (Laplace's method for an endpoint maximum). *Consider the Laplace integral*

$$
F(\lambda) = \int_a^b h(t)\, e^{\lambda f(t)}\, dt, \qquad \lambda \to \infty.
$$

*Let $I = [a, b]$ be a finite interval and assume that*

1. *$f(t)$ attains its maximum on $[a, b]$ only at the right endpoint $t = b$;*

2. *$h, f \in C([a, b])$;*

3. *$h, f \in C^\infty$ in a neighbourhood of $t = b$ and $f'(b) \neq 0$.*

*Then, as $\lambda \to \infty$,*

$$
F(\lambda) \;\sim\; e^{\lambda f(b)} \sum_{k=0}^{\infty} d_k \,\lambda^{-k-1},
$$

*with coefficients*

$$
d_k = \left(-\frac{1}{f'(t)} \frac{d}{dt}\right)^k \left(\frac{h(t)}{f'(t)}\right)\Bigg|_{t=b}.
$$

*Moreover, this expansion may be differentiated with respect to $\lambda$ arbitrarily many times.*

In our case we have

$$
a = 0, \quad b = 1, \qquad \lambda = \frac{1}{\beta}, \qquad h(t) = \frac{1}{\beta} \frac{1}{Z_f(x)} \left\{f\left(t\right) - g\left(t\right) + \beta \log \frac{Z_g(x)}{Z_f(x)}\right\}.
$$

Applying this theorem to our integral, we obtain the following asymptotics for the KL divergence:

$$
D_{KL}(\pi_f^*(\cdot \mid x) \,||\, \pi_g^*(\cdot \mid x)) = \beta \frac{1}{Z_f(x)} \log \frac{Z_g(x)}{Z_f(x)} + O(\beta^2),
$$

where we used the fact that $f(1) = g(1) = 0$ and $f'_-(1) = g'_-(1) = 1$ after rescale and shift described in Appendix K.1.

Finally, we need to show that the coefficient $\frac{1}{Z_f(x)} \log \frac{Z_g(x)}{Z_f(x)}$ has a constant leading term in its Taylor expansion with respect to $\beta$.

The partition function has the following asymptotics (see Appendix L for more details):

$$Z_f(x) = \beta + f''_-(1)\beta^2 + O(\beta^3)$$
$$Z_g(x) = \beta + g''_-(1)\beta^2 + O(\beta^3).$$

Hence

$$\frac{1}{Z_f(x)} = \frac{1}{\beta + f''_-(1)\beta^2 + O(\beta^3)}$$
$$= \frac{1}{\beta} - f''_-(1) + O(\beta)$$
$$\log\left(\frac{Z_g(x)}{Z_f(x)}\right) = \left(g''_-(1) - f''_-(1)\right)\beta + O(\beta^2)$$
$$\frac{1}{Z_f(x)} \log\left(\frac{Z_g(x)}{Z_f(x)}\right) = \left(g''_-(1) - f''_-(1)\right) + O(\beta).$$

Hence, by plugging the Taylor expansion for the coefficient $\frac{1}{Z_f(x)} \log \frac{Z_g(x)}{Z_f(x)}$ into the KL divergence asymptotics we obtain:

$$D_{KL}(\pi_f^*(\cdot \mid x) \,\|\, \pi_g^*(\cdot \mid x)) = \beta\left(g''_-(1) - f''_-(1)\right) + O(\beta^2) = O(\beta).$$

Thus, we show that for any two functions $f$ and $g$ from the family of transformation functions that are strictly increasing, continuous, defined on $[0,1]$ with finite value and non-zero left derivative at $x = 1$, under mild conditions, we asymptotically obtain the same optimal solutions. Moreover, the KL divergence between the corresponding optimal solutions decreases linearly with $\beta$, which means a very fast convergence. In practice, this means that one can take an arbitrary function from the considered family of transformation functions, given that $\beta$ is sufficiently small.

## L   Asymptotics of the partition function $Z$ for the family of quantile transformation functions strictly increasing and upper-bounded on the interval $[0,1]$

We compute the asymptotics of the partition function $Z$ with respect to $\beta$ for any choice of a strictly increasing continuous transformation function $f(t)$ defined on $[0,1]$ with finite value and non-zero left derivative at $t = 1$. We consider the case when the initial reward $\mathcal{R}$ (and thus the quantile-reward $\mathcal{R}_q$ and generalized transformed reward $\tilde{\mathcal{R}}$) can be approximated as a continuous variable.

We assume mild regularity conditions, specifically $f \in C^\infty[0,1]$. We also assume that $f$ is normalized according to the rescale and shift procedure described in Appendix K.1.

Recall that the partition function can be expressed as a moment generating function for the distribution of the reward used in the objective 1 (see Appendix D for the derivation). Here we consider objective 1 with the generalized transformed reward $\tilde{\mathcal{R}}$:

$$\tilde{Z} = \mathbb{E}_{\tilde{r} \sim \tilde{P}_{\text{ref}}(\cdot|x)}\left[\exp\left(\frac{1}{\beta}\tilde{r}\right)\right],$$

where $\tilde{P}_{\text{ref}}(r \mid x)$ denotes the probability distribution of the generalized transformed reward $\tilde{\mathcal{R}}(x, y)$ induced by the reference policy $y \sim \pi_{ref}(\cdot \mid x)$.

By applying the change-of-variables formula for expectations (also known as the "Law of the Unconscious Statistician" (LOTUS)) we obtain:

$$\tilde{Z} = \mathbb{E}_{\tilde{r} \sim \tilde{P}_{\text{ref}}(\cdot|x)}\left[\exp\left(\frac{1}{\beta}\tilde{r}\right)\right] = \mathbb{E}_{r_q \sim \text{U}[0,1]}\left[\exp\left(\frac{1}{\beta}f(r_q)\right)\right] = \int_0^1 \exp\left(\frac{1}{\beta}f(z)\right) dz.$$

Here we used the fact that the probability distribution of the quantile-reward $\mathcal{R}_q(x, y)$ induced by the reference policy $y \sim \pi_{ref}(\cdot \mid x)$ is uniform from 0 to 1 (see Appendix E for the derivation).

Since we require $f$ to be a strictly increasing smooth function defined on $[0, 1]$ with finite value and non-zero left derivative at $t = 1$ we can apply Laplace's method for an endpoint maximum to compute the asymptotics in terms of beta.

Here is the Laplace method formulation for reference:

**Theorem** (Laplace's method for an endpoint maximum). *Consider the Laplace integral*

$$F(\lambda) = \int_a^b h(t)\, e^{\lambda f(t)}\, dt, \qquad \lambda \to \infty.$$

*Let $I = [a, b]$ be a finite interval and assume that*

1. *$f(t)$ attains its maximum on $[a, b]$ only at the right endpoint $t = b$;*

2. *$h, f \in C([a, b])$;*

3. *$h, f \in C^\infty$ in a neighbourhood of $t = b$ and $f'(b) \neq 0$.*

*Then, as $\lambda \to \infty$,*

$$F(\lambda) \;\sim\; e^{\lambda f(b)} \sum_{k=0}^{\infty} d_k\, \lambda^{-k-1},$$

*with coefficients*

$$d_k = \left( -\frac{1}{f'(t)} \frac{d}{dt} \right)^k \left( \frac{h(t)}{f'(t)} \right) \Bigg|_{t=b}.$$

*Moreover, this expansion may be differentiated with respect to $\lambda$ arbitrarily many times.*

Applying this theorem to our integral (by taking $a = 0$, $b = 1$, $\lambda = \frac{1}{\beta}$ and $h(x) = 1$) we obtain the following asymptotics for the partition function:

$$\tilde{Z}(x) = \beta + f''_-(1)\beta^2 + O(\beta^3).$$

Note, that we assume the function $f$ to be normalized according to the rescale and shift procedure described in Appendix K.1 (as we require this property in the transformation functions equivalence proof K). Without this normalization the asymptotics will have a different form which can be derived similarly using the Laplace's method.

# M Detailed experimental protocol

**Table 5:** Initial models used in experiments.

| SFT-only Model | Base | Fine-tuning Data |
|---|---|---|
| Llama 8B Tülu 3 SFT (Lambert et al., 2024) | Llama-3.1-8B | Tülu 3 SFT dataset |
| Mistral 7B Instruct v0.2 (Jiang, 2024) | Mistral-7B-v0.2 | Unknown public instruction datasets |

**Table 6:** Datasets used in experiments.

| Dataset | Task | Train Size | Test Size | Filtering Criteria | Filtered |
|---|---|---|---|---|---|
| Magpie-Air (Xu et al., 2024b) | General Chat | 98,000 | 2,000 | Length $> 2048$ tokens | $< 1\%$ |
| UltraFeedback (Cui et al., 2024) | General Chat | 61,135 | 2,000 | Length $> 2048$ tokens | $< 1\%$ |
| LeetCode (Xia et al., 2025) | Coding | 2,641 | 228 | Length $> 2048$ tokens or $< 20$ test cases | $< 3\%$ |

**Table 7:** Rewards used for training and evaluation.

| Task | Reward | Description |
|---|---|---|
| General Chat | ArmoRM (Wang et al., 2024) | Reward model with sequences truncated at 2048 tokens |
| General Chat | Length-controlled ArmoRM (new) | Length bias canceled via regression as in AlpacaEval 2 (Dubois et al., 2024) |
| Coding | Python Sandbox (new) | Average pass rate over $\sim 100$ test cases per problem |

**Table 8:** Sampling parameters for reference and evaluation completions.

| Sampling Purpose | Temperature | Top-$p$ |
|---|---|---|
| Reference Completions | 1.0 | 1.0 |
| Evaluation (Llama 8B Tülu 3 SFT) | 0.6 | 0.9 |
| Evaluation (Mistral 7B Instruct v0.2) | 0.7 | 0.9 |

**Table 9:** Distribution shift settings for RL fine-tuning.

| Setting | Additional SFT on Chosen Before RL | Pairwise Preference Generation & Annotation Method |
|---|---|---|
| offline | No | Offline dataset annotations |
| off-policy-best | No | Best and worst from 6 completions (ArmoRM scores) |
| off-policy-random | No | Ranked random pair from 6 completions (ArmoRM scores) |
| SFT-offline | Yes | Offline dataset annotations |
| SFT-off-policy-best | Yes | Best and worst from 6 completions (ArmoRM scores) |
| SFT-off-policy-random | Yes | Ranked random pair from 6 completions (ArmoRM scores) |

**Table 10:** Hyperparameters for supervised and RL fine-tuning.

| Phase | Epochs | Learning Rates | Batch Size | Optimizer | LR Schedule | Gradient Clipping |
|---|---|---|---|---|---|---|
| SFT (Chat) | 1 | $5 \times 10^{-7}$ | 128 | Adam | Cosine w/ 10% warm-up | 10.0 |
| SFT (Code) | 3 | $5 \times 10^{-7}$ | 128 | Adam | Cosine w/ 10% warm-up | 10.0 |
| RL | 1 | $\{1, 3, 10\} \times 10^{-7}$ | 128 | Adam | Cosine w/ 10% warm-up | $10^8$ (effectively none) |

**Table 11:** RL fine-tuning hyperparameters: KL-regularization parameter ($\beta$) sweep and QRPO number of generated reference rewards.

| Algorithm | Chat Task | Code Task | QRPO Reference Rewards |
|---|---|---|---|
| DPO | 0.01, 0.03, 0.1 | 0.01, 0.03, 0.1 | - |
| SimPO | 2, 2.5, 10 | 2, 2.5, 10 | - |
| REBEL | 1e-6, 1e-4, 1e-2 | 1e-4, 1e-2, 1.0 | - |
| QRPO | 3e-4, 0.001, 0.003 | 0.003, 0.01, 0.03 | $n = 1$ (Magpie-Air), $n = 3$ (UltraFeedback), $n = 20$ (LeetCode) |

```python
 1  def qrpo_loss(beta, logps, ref_logps, rewards, ref_rewards):
 2      """Compute the QRPO loss for a batch of prompts.
 3      Args:
 4          beta (`torch.Tensor: (1,)`):
 5              The beta parameter for the QRPO loss.
 6          logps (`torch.Tensor: (batch_size,)`):
 7              Log probabilities of the training completions for the model.
 8          ref_logps (`torch.Tensor: (batch_size,)`):
 9              Log probabilities of the training completions for the reference model.
10          rewards (`torch.Tensor: (batch_size,)`):
11              Rewards of the training completions.
12          ref_rewards (`torch.Tensor: (batch_size, num_ref_rewards)`):
13              Rewards of the reference completions generated by the reference model.
14      Returns:
15          loss (`torch.Tensor[batch_size]`): The computed QRPO loss.
16      """
17      log_ratio = logps - ref_logps
18      quantile_rewards = (ref_rewards ≤ rewards.unsqueeze(dim=1)).float().mean(dim=1)
19      log_Z = torch.log(beta) + 1 / beta
20      loss = (quantile_rewards - beta * log_Z - beta * log_ratio) ** 2
21      return loss
```

**Figure 10:** QRPO loss reference implementation in PyTorch.

## M.1 Codebase

We release a reference implementation of QRPO based on a HuggingFace TRL (von Werra et al., 2020) trainer also supporting the baselines we trained (DPO, REBEL, SimPO) at https://github.com/CLAIRE-Labo/quantile-reward-policy-optimization. The codebase is based on the Python Machine Learning Research Template (Moalla, 2025). It contains

- All the scripts we used to train and produce all the results presented in the paper (including reference data generation, training, and plotting).

- All of our infrastructure and experiment management code for running and managing (tracking failures, etc.) experiments at scale on a SLURM cluster.

- A reference implementation for a scalable code sandbox on SLURM clusters with container runtimes, that does not require elevated privileges.

- The specification of our software environment for reproducibility (OCI container image, `pip` dependencies, etc.). We acknowledge that few users have access to the NVIDIA GH200 platform to reproduce our exact environment and numbers throughout the training pipeline, so this specification mainly serves as a reference.

## M.2 Initial SFT-only Models

We consider popular middle-scale models that have been instruction fine-tuned (IFT/SFT) but have not undergone RL fine-tuning.[6] We choose `allenai/Llama-3.1-Tulu-3-8B-SFT` (Llama 8B Tülu 3 SFT) (Lambert et al., 2024), a fine-tuning of the `meta-llama/Llama-3.1-8B` base model on the Tülu 3 SFT dataset (Lambert et al., 2024), and `mistralai/Mistral-7B-Instruct-v0.2` (Mistral 7B Instruct v0.2) (Jiang, 2024).

---

[6]This is important as our experiments emulate the RL fine-tuning stage after the supervised fine-tuning stage in a post-training pipeline, and more importantly because reporting improvements on only specific benchmarks over already (usually closed-source) RL fine-tuned models is a biased view on the performance profile of these models which have been carefully tuned to balance multiple benchmarks. E.g., taking Llama-3-8B-Instruct and improving its AlpacaEval 2 scores by RL fine-tuning it on a narrow chat dataset is not a faithful way to report an algorithmic improvement as it does not show where the model has regressed.

## M.3 Datasets

For the general chat task, we use two popular alignment datasets, extensively used in previous work (Meng et al., 2024; Gao et al., 2024) consisting primarily of synthetic data.

`Magpie-Align/Magpie-Air-DPO-100K-v0.1` (Magpie-Air) (Xu et al., 2024b) is a strong alignment dataset containing 98000 training samples (and 2000 testing samples) with synthetic instructions and completions generated by Llama-3-8B-Instruct targeting mainly information seeking, and other categories such as creative writing, advice seeking, planning, and math. The chosen (resp. rejected) completions have been selected by picking the best (resp. worse) completions in a set of 5 completions rated by the `RLHFlow/ArmoRM-Llama3-8B-v0.1` reward.

`HuggingFaceH4/ultrafeedback_binarized` (UltraFeedback) (Cui et al., 2024) consists of 61135 training samples (and 2000 testing samples) with instructions targeting instruction following, truthfulness, honesty, helpfulness, and completions generated by a large pool of models with different capabilities (between 7B and 70B open models and closed models including GPT4), annotated by GPT-4. This is the pre-processed version of the original UltraFeedback dataset that was used to train Zephyr-7B-$\beta$ (Tunstall et al., 2024), where for each prompt, the completion with the highest overall score is taken as the chosen completion, and one of the remaining three at random as the rejected one.

For the coding task, we use `newfacade/LeetCodeDataset` (LeetCode) (Xia et al., 2025), which contains 2641 training samples (and 228 testing samples) comprising easy, medium, and hard LeetCode problems. Each problem has a median of 100 test cases and contains a correct solution that we use for additional SFT before performing the RL fine-tuning phase.

For all the datasets, in both the train and test splits, we filter out the samples where the length of the prompt plus the completion is longer than 2048 tokens. For the LeetCode dataset, we additionally filter out the problems which have less than 20 test cases. This results in less than 1% of filtering for Magpie-Air and UltraFeedback, and less than 3% of filtering for LeetCode. This is to match our training and reward model context length of 2048 to avoid prompts for which the model often generates long sequences that get truncated at training time (off-policy) and by the reward model and would not give a meaningful reward signal.

We additionally split the test set in each dataset into two equal splits: a validation subset for hyperparameter selection and a test subset to report the final numbers. For the LeetCode dataset, we maintain the distribution of easy, medium, and hard problems in the subsets (stratification by the difficulty label).

## M.4 Rewards

For the general chat task we use the `RLHFlow/ArmoRM-Llama3-8B-v0.1` (ArmoRM) (Wang et al., 2024) reward model with a maximum sequence length of $2048$. It is used to compute the rewards directly used by REBEL and QRPO in all distribution settings and to label the chosen and rejected preferences for DPO and SimPO in the off-policy setting. This also holds for Magpie-Air in the offline setting as its chosen and rejected annotations have been annotated with ArmoRM as well by Xu et al. (2024b). For UltraFeedback the offline chosen and rejected annotations are not annotated by the ArmoRM, but most of the best models for the baselines turned out to be from the off-policy setting due to the low quality of the UltraFeedback offline completions.

Although ArmoRM has a verbosity component to reduce length bias, it still suffers from out-of-distribution length bias. i.e., its length bias has been minimized on specific outputs (original UltraFeedback), but is still present when computing the rewards on chats that diverge from the UltraFeedback outputs, which we did observe in our experiments. We therefore also report a *length-controlled* reward whose coefficients are computed for each model based on a regression that cancels the length component as done in AlpacaEval 2 (Dubois et al., 2024) (see Appendix M.5 for more details on the length-controlled reward computation).

For LeetCode, we use a Python sandbox for executing LLM-generated code in a safe environment with exclusive and controlled hardware limits, which we built for this project. The reward for a solution to a problem is the average number of test cases passed in the suite of tests for the problem. Each problem has a minimum of 20 tests and a median of 100 tests, and we cap the number of tests to run for each problem to the first 100 tests for computational efficiency. We use this average pass rate reward instead of a binary reward indicating whether all tests passed for two main reasons: (i)

because the return value of the solution for edge cases is often not specified, still those appear in the test cases with arbitrary targets causing high rewards to collapse when binarized (e.g., passing the test cases with a rate of 0.99 because of a single failed test on an under-specified edge case would be converted to a 0 binary reward). (ii) because we assume a reward that can be treated as continuous in the theory of QRPO.[7]

## M.5 Length-controlled reward (LC-Reward) computation

For every prompt $x$ in an evaluation split (validation or test), we first collect completions produced by the reference policy $\pi_{ref}$ and compute the prompt-wise mean and standard deviation of the original reward $\mathcal{R}$ and the token length $L$ corresponding to the reference policy:

$$\mu_R^{\text{ref}}(x) = \mathbb{E}_{y \sim \pi_{ref}(\cdot|x)}\big[\mathcal{R}(x,y)\big], \qquad \sigma_R^{\text{ref}}(x) = \text{std}_{y \sim \pi_{ref}(\cdot|x)}\big[\mathcal{R}(x,y)\big],$$
$$\mu_L^{\text{ref}}(x) = \mathbb{E}_{y \sim \pi_{ref}(\cdot|x)}\big[L(y)\big], \qquad \sigma_L^{\text{ref}}(x) = \text{std}_{y \sim \pi_{ref}(\cdot|x)}\big[L(y)\big].$$

These prompt-wise statistics absorb two major confounders: (i) the intrinsic difficulty of the query, captured by the average reward $\mu_R^{\text{ref}}(x)$ and its variance $\sigma_R^{\text{ref}}(x)$, and (ii) the target answer length implicitly specified by the prompt, captured by $\mu_L^{\text{ref}}(x)$ and $\sigma_L^{\text{ref}}(x)$.

**Step 1: prompt-wise normalization**  For each completion $y$ produced by a trained model $\pi_\theta$ we compute the normalized reward and completion length using the normalization coefficients for its prompt:

$$R_{\text{norm}}(x,y) = \frac{\mathcal{R}(x,y) - \mu_R^{\text{ref}}(x)}{\sigma_R^{\text{ref}}(x)}, \qquad L_{\text{norm}}(x,y) = \frac{L(y) - \mu_L^{\text{ref}}(x)}{\sigma_L^{\text{ref}}(x)}.$$

**Step 2: fitting the length coefficient for a trained model**  Given a trained model's outputs $y$ on the entire evaluation split (validation or test), we regress the normalized reward on the normalized length by fitting a linear model:

$$R_{\text{norm}}(x,y) = k\, L_{\text{norm}}(x,y) + b + \varepsilon,$$

where $k$ captures the *global* linear dependence of ArmoRM scores on completion length for this model in the evaluation split and $b$ absorbs any fixed shift.

**Step 3: debiasing the rewards for a trained model**  Finally, we subtract the length contribution (bias) learned for the trained model and transform the result back to the original reward scale:

$$\mathcal{R}_{\text{LC}}(x,y) = \big(R_{\text{norm}}(x,y) - k\, L_{\text{norm}}(x,y)\big)\sigma_R^{\text{ref}}(x) + \mu_R^{\text{ref}}(x)$$

Written directly in terms of the original magnitudes this is:

$$\boxed{\mathcal{R}_{\text{LC}}(x,y) = \mathcal{R}(x,y) - k\, \frac{L(y) - \mu_L^{\text{ref}}(x)}{\sigma_L^{\text{ref}}(x)}\, \sigma_R^{\text{ref}}(x)}$$

By construction $\mathcal{R}_{\text{LC}}$ has zero first-order correlation with length while preserving the prompt-wise mean and variance of the original reward distribution, ensuring that improvements measured with LC-Rewards reflect qualitative gains rather than verbosity.

## M.6 Sampling for reference completions and evaluations

When sampling reference completions, we use a temperature of 1 and top-$p$ of 1 to keep the generations faithful to the distribution of the reference policy. This keeps the implementation faithful to the theoretical derivation of QRPO and at the same time maintains diversity in the reference completions and rewards. We did observe in early experiments that sampling with a lower temperature and smaller top-$p$ resulted in less diversity of reference rewards and a worse evaluation performance.

---

[7]We have also derived a version of QRPO for rewards with a small number of possible values in Appendix F which would be compatible with the binary reward in this case. We defer this setting to future work.

We defer the exploration of the hyperparameters related to the generation of the reference completion to future work.

When sampling evaluation completions, we use the recommended temperature and top-$p$ for each model, that is a temperature of 0.6 and top-$p$ of 0.9 for Llama 8B Tülu 3 SFT and a temperature of 0.7 and top-$p$ 0.9 for Mistral 7B Instruct v0.2. The parameters of Llama 8B Tülu 3 SFT are taken from its default generation configuration file and the parameters of Mistral 7B Instruct v0.2 are a recommended choice within API servers.[8]

All the sampling is done with vLLM (Kwon et al., 2023) and we keep all the other sampling parameters as default. In early experiments we generated up to 200 completions per prompt, and since this is an embarrassingly parallel problem, our codebase includes a scalable sampling pipeline with data parallelism to effectively scale this task to a cluster scale.

To generate completions for the LeetCode dataset, we modify the original prompt to make the model aware of the initialization code that runs before its suggested solution. This code includes generic imports and definitions of helper classes (see Figures 11, 12, and 13). The imports and definitions are used in the template solution code that the model gets in its prompt and in the test suite, and are therefore critical in its crafting of the problem solution. The original initialization code is not constant across problems and we found it to lack robustness and to be subject to bad practices such as having several wildcard import statements. We therefore combined and refactored all initialization prefixes into a single unified one shown in Figures 12 and 13.

## M.7 Distribution shift settings

It is well known that although policy fitting methods can be used to fit the optimal policy with any data distribution, they do exhibit different levels of performance depending on the distribution shift between the model being trained and the completions in the training dataset. For example, it can be recommended to perform SFT on the chosen completions when the completions in the preference dataset are not from the model being aligned. We therefore let the distribution shift setting be a hyperparameter and select 6 different distribution shift settings to put all algorithms in their best setting. We consider the *SFT-chosen* and *no-SFT* cases where the model is first fine-tuned on the chosen completions or not (our initial models are already instruction fine-tuned, this is an extra fine-tuning to reduce distribution shift). Subsequent to these two settings we have the *offline* setting (data from the dataset) and two *off-policy* settings, where we replace the chosen and rejected completions in the dataset (keeping the prompts) with the completions generated by the model before training (thus $2 \times (1 + 2) = 6$ settings). We refer to the off-policy settings by (i) *off-policy-best* when we follow Xu et al. (2024b) in picking the chosen (resp. rejected) from the best (resp. worst) out of 6 completions scored by the ArmoRM reward model, and by (ii) *off-policy-random* when we only pick them by comparing and annotating 2 random completions from the 6 completions. Unlike popular belief, we observe that DPO obtains better results on off-policy-random than on off-policy-best in multiple cases (see Table 12 for example).

For the LeetCode dataset, as it is small and does not contain preferences, we only use the *off-policy-random* setting to generate 20 completions per prompt and group them in 10 random pairs that we annotate with the code execution reward to make preferences and increase the dataset size by 10 folds (each prompt gets 10 pairs).

## M.8 Hardware and software acceleration

We train all the models (SFT and RL fine-tuning) with the same distributed training setup. We distribute training with HuggingFace Accelerate (Gugger et al., 2022) using the DeepSpeed (Aminabadi et al., 2022) plugin at ZeRO stage 1 (optimizer state partitioning) over 8 GPUs from 2 nodes with 4 NVIDIA GH200 each.

Our SFT trainer and RL fine-tuning trainer (same trainer supporting DPO, SimPO, REBEL, and QRPO) are based on the HuggingFace TRL (von Werra et al., 2020) SFT and DPO trainers.

---

[8]The data was available on `https://openrouter.ai/`, but the website changed interface and does not show this data anymore. These are also the parameters suggested when asking Le Chat.

## M.9 Baseline algorithms and hyperparameters

We perform full model fine-tuning with a batch size of 128 (128 pairs of completions for RL fine-tuning and 128 individual completions for SFT). For both SFT and RL fine-tuning we train using the Adam optimizer and a cosine learning rate schedule with 10% warm-up steps.

For SFT (when fine-tuning on the chosen completions) we use a learning rate of $5e-7$ and train for one epoch in the chat task and three epochs in the code task (smaller dataset) The SFT loss is computed on all the tokens in the sequence (prompt and completion). This is the default in the TRL trainer. We apply gradient clipping by norm with value 10.0 (to prevent potential spikes, though the norm stabilizes below 10.0 very early in training).

For RL fine-tuning, we sweep over three learning rates $[1e-7, 3e-7, 1e-6]$ and train for one epoch. We run three $\beta$ values for each algorithm, following a log scale, unless specific hyperparameters are recommended by the authors of the respective algorithms. For the chat task, for DPO we sweep over $[0.01, 0.03, 0.1]$ which is the same magnitude as in Meng et al. (2024) and are also commonly used in the literature, for REBEL we sweep over $[1e-6, 1e-4, 1e-2]$ as done by Gao et al. (2024) in a similar experimental setting using UltraFeedback and ArmoRM, for SimPO we sweep over $[2, 2.5, 10]$ as recommended by Meng et al. (2024) keeping the margin at 0.5 as recommended by their public codebase to maintain the same budget of hyperparameters for all methods, and for QRPO we sweep over $[3e-4, 0.001, 0.003]$ which we found to be a good range in preliminary experiments on another chat dataset. The code task is harder and we expect a worse offline-to-online generalization requiring to aim for a more conservative optimal policy and change in the rewards. Therefore we adjust the range of $\beta$ values used in the sweep for each method depending on the impact of $\beta$ on the method to obtain similar impacts: for REBEL we increase $\beta$ by 2 orders of magnitude and we sweep over $[1e-4, 1e-2, 1.0]$, one order to be more conservative and one order to match the difference between rewards which increases from the chat task by around one order of magnitude since REBEL is sensitive to the reward scale to preserve the signal-to-regularization ratio (higher $\beta$ values have also been used by Gao et al. (2024) in their summarization experiment); For QRPO we increase $\beta$ by only one order of magnitude and sweep over $[0.003, 0.01, 0.03]$, we only need one order to be more conservative and do not need to take into account the change in reward scale as the quantile reward is invariant to it; For DPO we keep the same values for $\beta$, since this parameter is not tied to an external reward scale (model learns an internal reward implicitly based on preferences). Moreover, we do not adjust for additional conservatism, as $\beta = 1$ is too conservative. Finally for SimPO, the training completions are at least twice shorter therefore $\beta$ is already effectively increased in the loss and the values do not follow a log scale so we can't increase by an order of magnitude and thus we keep the same values.

To estimate the percentile rewards for QRPO, we use $n = 1$ reference model rewards for Magpie-Air, $n = 3$ for UltraFeedback, and $n = 20$ for LeetCode. Note that when using $n = 1$, the quantile reward is obtained by comparing the reward of the training sample to the reward of the single reference reward and yields a binary quantile reward of 0 or 1.

We use the same RL fine-tuning trainer for all methods, changing only the training loss with the configuration file, so although QRPO only needs one completion instead of a completion pair, we consider the pair as a single sample in the batch so that all methods use the same number of training steps when consuming one epoch. QRPO applies its loss independently on each completion of the pair and can be seen as effectively using double the batch size, but is actually using exactly the same number of completions.

## M.10 Evaluation methods

For each training run, we save five equally spaced checkpoints and compute the online rewards of completions generated by the checkpoints on prompts from the evaluation split of the dataset used in the run. The rewards are based on the ArmoRM reward model for the chat task and the code execution reward for the code task. We repeat the online evaluation three times to obtain estimates of the mean performance and its standard deviation.

We split the evaluation dataset into a validation split (selection) and a testing split (reporting). For each baseline and task, we use the mean online reward in evaluations on the *validation* split to select the best checkpoint across hyperparameters and checkpoints (learning rate, beta, distribution shift setting, checkpoint number) for a given algorithm, model, and dataset, and report mean performance on the *test*

split and its standard deviation. We start by selecting the best checkpoint across hyperparameters for each distribution shift setting. Then we perform a more careful selection among the best checkpoints for each distribution shift setting as we believe that the training data distribution shift can qualitatively affect different characteristics of the trained model, not limited to the online reward: We first shortlist checkpoints that are in the 98%-CI (one-sided) of the best online reward value among the checkpoints. Then, we refine the shortlist to checkpoints that are in the 80%-CI (one-sided) of the best online length-controlled reward (LC-Reward) value. This procedure allows us to take into account both online Reward and LC-Reward to make a more robust selection. If we have more than one checkpoint present in the final shortlist, we choose one with the better online reward.

For downstream evaluations, on the chat task we further evaluate the best checkpoint of every algorithm on AlpacaEval 2.0 (Li et al., 2023; Dubois et al., 2024) and report the length-controlled win rate against GPT-4 preview (gpt-4-1106-preview denoted as `gpt4_turbo` in the AlpacaEval repository) as the final performance of an algorithm on a (model, dataset) configuration. AlpacaEval 2.0 is a widely used benchmark by the community, assessing broad conversational skills across a wide range of queries with a total of 805 questions. On the code task, the reward on the test split of the evaluation set can already be considered as downstream as they measure the performance of the model on the popular LeetCode coding task.

### M.11 Statistical significance

The claims in this work are supported by experiments that include LLM training and evaluation. There are three main sources of randomness present at two levels. (i) At the training level, our results can be influenced by the reference model generation randomness used to compute the quantile rewards for QRPO, the random shuffling of the data samples in the dataloader, and the hardware acceleration non-determinism. (ii) At the evaluation level, our results can be influenced by the sampling randomness during the evaluation of trained models and again by the hardware acceleration non-determinism. Unfortunately, it is computationally prohibitive to repeat the training several times for each setup to estimate the noise in the results, thus the first source of randomness cannot be addressed directly. However, we address the noise introduced by sampling from the models during evaluation by repeating each sampling procedure three times and estimating the average value and a standard deviation (we cannot repeat the experiments on different hardware to control for that source of randomness). All conclusions drawn from the obtained results were made by taking into account the obtained noise estimates. Furthermore, we employ a 3-split strategy to train, select hyperparameters, and report performance which has not been employed in previous offline alignment work. This is possible in our case because we compute online rewards during evaluation for all checkpoints and therefore can perform a robust hyperparameter selection.

### M.12 Common implementation pitfalls and experiment protocol limitations

We have identified several pitfalls in alignment algorithms implementations and experimental protocols while experimenting with various fine-tuned models and fine-tuning libraries and attempting to replicate previous published results. We provide a non-exhaustive list below aimed to raise awareness about these details that result in non-negligible performance differences. In particular, examples mentioned in this section are not meant to point out issues in specific codebases. We recommend the open-source community to adopt more robust testing for these cases. We have observed that the codebases with the most bugs are those that try to support the largest number of use cases and users, introducing frequent bugs with the frequent addition of new features, and we encourage the community to weigh a bit more the robust testing end of the tradeoff.

**End-of-sequence (EOS) token** The EOS token is part of the sequence the model predicts to a have a sound language model distribution, however we have observed several implementation issues with regard to it. For example we have found that:

- There was a bug in SFT Trainer of the TRL library (von Werra et al., 2020) where when the padding token was set to be the same as the EOS token as shown in their examples, the EOS token would be masked out in the loss like the padding token, leading to models never predicting and end of sequence during the SFT and then at inference time generating non-ending sequences. This bug has since been fixed in TRL (#3200, #3328).

- Popular models diverge in the EOS used for pre-training and post-training with their Hug-gingFace config remaining with the pre-training EOS and leading them to generate non-ending sequences out of the box. (E.g., the Magpie-Align models trained with the Axolotl library)(EOS and chat template).
- An extra EOS token is added to the sequence after applying the chat template if it already contains an EOS in the DPO implementation of TRL. This bug is still present in TRL at the time of writing. (Added EOS after chat template.)

**Gradient and loss accumulation across devices and accumulation steps**    There is an increased complexity to aggregate the loss of all micro-batches across devices and accumulation time steps when training in a distributed setting and using gradient accumulation to simulate a larger batch. For a fixed batch size any combination of these dimensions of splitting the batch size should yield the same gradient. Yet we have observed several bugs with regard to this aggregation which leads to a biased gradient and incorrect reported gradient norm in several libraries. For example,

- In the HuggingFace Transformers library (Wolf et al., 2020) there was a bug leading to the gradient norm to scale with the number of accumulation steps, which is solved at the time of writing (#35808).
- There is still new code being merged in the TRL library for which the gradient norm is not accurate (#3317).
- The incorrect loss and gradient accumulation bug has had a long history of being patched and then appearing again (Unsloth blogpost, #35207, #3574).

**Arbitrary sampling parameters at evaluation**    Some benchmarks or models do not specify the sampling parameters for their evaluations. For example, AlpacaEval (Li et al., 2023) does not specify a decoding configuration, resulting in different works reporting performance on AlpacaEval with different decoding strategies even when using the same base model making the comparison across works very hard. For example, for the same Llama 3 8B Xu et al. (2024b) use greedy decoding with a repetition penalty on AlpacaEval while Meng et al. (2024) perform a grid search over multiple sampling temperatures and report their results with a temperature of 0.9.

**Unfair hyperparameter budgets**    We believe that new methods should compare to baselines while maintaining the same hyperparameter budget. This is because if a new method introduces a new hyperparameter that increases the computational budget to find its optimum, baselines could as well find better optima by searching over more hyperparameters. However, we find that this is not always followed. For example, while Meng et al. (2024) perform a comprehensive hyperparameter search for their baselines, they still use four times more hyperparameters for SimPO than for DPO.

**Unclear hyperparameter selection**    In offline alignment, at least with the current algorithms, there is no accurate metric to predict online performance on downstream evaluations. For example, we have observed that the DPO loss, margins, or accuracy are not reliable metrics to predict online performance. This has also been observed by Rafailov et al. (2024) for other policy fitting algorithms. Furthermore, the best hyperparameters cannot be selected based on the online performance on the downstream evaluations, as this practice would be subject to a selection bias. Therefore, it is important for offline alignment work to report their hyperparameter selection protocol. Yet, we have found that several research papers do not document their hyperparameter selection protocol. This also leads to reproducibility issues and discrepancies between papers that seemingly use the same experimental protocol (datasets, hyperparameters, etc.), but still report different numbers.

## Example formatted prompt from LeetCode

```
You are an expert Python programmer. You will be given a question (problem
specification) and will generate a correct Python program that matches the
specification and passes all tests.

### Question:
Given an array of integers nums and an integer target, return indices of the
two numbers such that they add up to target. You may assume that each input
would have exactly one solution, and you may not use the same element twice.
You can return the answer in any order.

Example 1:

Input: nums = [2,7,11,15], target = 9
Output: [0,1]
Explanation: Because nums[0] + nums[1] == 9, we return [0, 1].

Example 2:

Input: nums = [3,2,4], target = 6
Output: [1,2]

Example 3:

Input: nums = [3,3], target = 6
Output: [0,1]

Constraints:

2 <= nums.length <= 104
-109 <= nums[i] <= 109
-109 <= target <= 109
Only one valid answer exists.

Follow-up: Can you come up with an algorithm that is less than O(n2) time
complexity?

### Format:
Your code will run after the following definitions and imports, which may or
may not be needed.
```python
(see Figures 7 and 8 ...)
```
You will use the following starter code to write the solution to the problem
and enclose your code within delimiters.
You have to add the missing imports to the starter code to run correctly.
```python
# Additional imports.

# End of additional imports.
class Solution:
    def twoSum(self, nums: List[int], target: int) -> List[int]:

```

### Answer: (use the provided format with backticks)
```

**Figure 11:** Example formatted prompt from LeetCode from which reference solution are generated. We have modified the original prompts by adding the highlighted text to make a model aware of the code that runs before its suggested solution.

**Definitions and imports for helper classes (part 1)**

```python
import collections
import itertools
import functools
import math
import string
import random
import bisect
import re
import operator
import heapq
import queue

from typing import List, Tuple, Dict, Any, Union, Optional
from queue import PriorityQueue
from itertools import combinations, permutations
from functools import lru_cache
from collections import defaultdict, OrderedDict, deque, Counter

inf = float('inf')

class ListNode:
    def __init__(self, val=0, next=None):
        self.val = val
        self.next = next

def list_node(values: list):
    if not values:
        return None
    head = ListNode(values[0])
    p = head
    for val in values[1:]:
        node = ListNode(val)
        p.next = node
        p = node
    return head

def is_same_list(p1, p2):
    if p1 is None and p2 is None:
        return True
    if not p1 or not p2:
        return False
    return p1.val == p2.val and is_same_list(p1.next, p2.next)
```

**Figure 12:** Definitions and imports for helper classes (part 1). These are used in the prompt (Figure 11) and prepended to the code generated by a model.

**Definitions and imports for helper classes (part 2)**

```python
class TreeNode:
    def __init__(self, val=0, left=None, right=None):
        self.val = val
        self.left = left
        self.right = right

def tree_node(values: list):
    if not values:
        return None
    root = TreeNode(values[0])
    i = 1
    queue = deque()
    queue.append(root)
    while queue:
        node = queue.popleft()
        if i < len(values) and values[i] is not None:
            node.left = TreeNode(values[i])
            queue.append(node.left)
        i += 1
        if i < len(values) and values[i] is not None:
            node.right = TreeNode(values[i])
            queue.append(node.right)
        i += 1
    return root

def is_same_tree(p, q):
    if not p and not q:
        return True
    elif not p or not q:
        return False
    elif p.val != q.val:
        return False
    else:
        return is_same_tree(p.left, q.left) and is_same_tree(p.right, q.right)

class Node:
    def __init__(self, val=0, left=None, right=None, random=None):
        self.val = val
        self.left = left
        self.right = right
        self.random = random

class CategoryHandler:
    def haveSameCategory(self, a: int, b: int) -> bool:
        pass
```

**Figure 13:** Definitions and imports for helper classes (part 2). These are used in the prompt (Figure 11) and prepended to the code generated by a model.

# N  Detailed empirical results

(a) learning rate = 1e-7

(b) learning rate = 3e-7

(c) learning rate = 1e-6

(d) Best learning rate on a validation set for each number of reference rewards.

**Figure 14:** Scaling curves in Figure 3 for different learning rates. For a fixed $\beta$ and learning rate, we still select the best checkpoint according to a validation set and report the performance on a test set, as described in our experimental protocol in Appendix M.

**Table 12:** Llama 8B Tülu 3 SFT, Magpie-Air dataset. The best models selected across distribution shifts are marked with * according to rewards and LC-rewards as described in the selection protocol in Appendix M. We underline the values determining the selection shortlist using the validation split (valid).

| Method | Setting | Reward (valid/test) | LC-Reward (valid/test) | $\beta$ | LR | Step |
|---|---|---|---|---|---|---|
| **Base** | | $0.1493_{\pm0.0003}$ / $0.1519_{\pm0.0010}$ | $0.1495_{\pm0.0006}$ / $0.1518_{\pm0.0011}$ | | | |
| | SFT | $\underline{0.1612}_{\pm0.0004}$ / $0.1649_{\pm0.0008}$ | $\underline{0.1614}_{\pm0.0002}$ / $0.1648_{\pm0.0008}$ | | 5e-7 | 764 |
| **DPO** | offline | $0.1871_{\pm0.0000}$ / $0.1903_{\pm0.0002}$ | $0.1750_{\pm0.0012}$ / $0.1780_{\pm0.0008}$ | 0.03 | 1e-6 | 320 |
| | offpolicy–best | $0.1734_{\pm0.0001}$ / $0.1761_{\pm0.0001}$ | $0.1671_{\pm0.0002}$ / $0.1706_{\pm0.0003}$ | 0.01 | 3e-7 | 320 |
| | offpolicy–random | $0.1774_{\pm0.0001}$ / $0.1804_{\pm0.0005}$ | $0.1682_{\pm0.0009}$ / $0.1716_{\pm0.0006}$ | 0.01 | 1e-6 | 480 |
| | SFT → offline* | $\underline{0.1912}_{\pm0.0001}$ / $0.1943_{\pm0.0002}$ | $\underline{0.1869}_{\pm0.0007}$ / $0.1904_{\pm0.0003}$ | 0.01 | 1e-6 | 160 |
| | SFT → offpolicy–best | $0.1874_{\pm0.0000}$ / $0.1904_{\pm0.0002}$ | $0.1871_{\pm0.0006}$ / $0.1903_{\pm0.0003}$ | 0.01 | 1e-6 | 160 |
| | SFT → offpolicy–random | $0.1872_{\pm0.0007}$ / $0.1907_{\pm0.0002}$ | $0.1900_{\pm0.0005}$ / $0.1922_{\pm0.0001}$ | 0.01 | 1e-6 | 480 |
| **SimPO** | offline | $0.1493_{\pm0.0003}$ / $0.1519_{\pm0.0010}$ | $0.1495_{\pm0.0006}$ / $0.1518_{\pm0.0011}$ | 10 | 1e-6 | 0 |
| | offpolicy–best | $0.1830_{\pm0.0001}$ / $0.1858_{\pm0.0002}$ | $0.1714_{\pm0.0007}$ / $0.1755_{\pm0.0007}$ | 2.5 | 3e-7 | 160 |
| | offpolicy–random | $0.1822_{\pm0.0002}$ / $0.1857_{\pm0.0004}$ | $0.1735_{\pm0.0037}$ / $0.1740_{\pm0.0008}$ | 10 | 3e-7 | 320 |
| | SFT → offline* | $\underline{0.1949}_{\pm0.0004}$ / $0.1976_{\pm0.0002}$ | $\underline{0.1943}_{\pm0.0004}$ / $0.1975_{\pm0.0003}$ | 10 | 1e-6 | 160 |
| | SFT → offpolicy–best | $0.1918_{\pm0.0001}$ / $0.1941_{\pm0.0002}$ | $0.1914_{\pm0.0002}$ / $0.1936_{\pm0.0001}$ | 10 | 3e-7 | 320 |
| | SFT → offpolicy–random | $0.1912_{\pm0.0002}$ / $0.1939_{\pm0.0003}$ | $0.1910_{\pm0.0003}$ / $0.1938_{\pm0.0005}$ | 2.5 | 3e-7 | 764 |
| **REBEL** | offline | $0.1830_{\pm0.0009}$ / $0.1861_{\pm0.0001}$ | $0.1702_{\pm0.0005}$ / $0.1747_{\pm0.0024}$ | 1e-4 | 1e-6 | 160 |
| | offpolicy–best | $0.1777_{\pm0.0002}$ / $0.1813_{\pm0.0003}$ | $0.1677_{\pm0.0003}$ / $0.1723_{\pm0.0006}$ | 1e-6 | 1e-7 | 640 |
| | offpolicy–random | $0.1744_{\pm0.0003}$ / $0.1776_{\pm0.0002}$ | $0.1677_{\pm0.0007}$ / $0.1707_{\pm0.0007}$ | 1e-4 | 1e-6 | 320 |
| | SFT → offline* | $\underline{0.1914}_{\pm0.0006}$ / $0.1937_{\pm0.0011}$ | $\underline{0.1867}_{\pm0.0007}$ / $0.1889_{\pm0.0012}$ | 1e-6 | 3e-7 | 764 |
| | SFT → offpolicy–best | $0.1855_{\pm0.0002}$ / $0.1884_{\pm0.0001}$ | $0.1847_{\pm0.0001}$ / $0.1883_{\pm0.0001}$ | 1e-4 | 1e-6 | 160 |
| | SFT → offpolicy–random | $0.1827_{\pm0.0002}$ / $0.1859_{\pm0.0003}$ | $0.1824_{\pm0.0005}$ / $0.1860_{\pm0.0003}$ | 1e-4 | 1e-6 | 480 |
| **QRPO** | offline | $0.1855_{\pm0.0000}$ / $0.1879_{\pm0.0001}$ | $0.1741_{\pm0.0005}$ / $0.1835_{\pm0.0016}$ | 3e-4 | 1e-6 | 480 |
| | offpolicy–best | $0.1762_{\pm0.0005}$ / $0.1788_{\pm0.0004}$ | $0.1719_{\pm0.0013}$ / $0.1754_{\pm0.0007}$ | 0.001 | 1e-7 | 320 |
| | offpolicy–random | $0.1772_{\pm0.0001}$ / $0.1804_{\pm0.0009}$ | $0.1781_{\pm0.0019}$ / $0.1787_{\pm0.0013}$ | 0.001 | 1e-7 | 320 |
| | SFT → offline* | $\underline{0.1948}_{\pm0.0000}$ / $0.1972_{\pm0.0003}$ | $\underline{0.1975}_{\pm0.0004}$ / $0.2005_{\pm0.0004}$ | 3e-4 | 1e-6 | 160 |
| | SFT → offpolicy–best | $0.1810_{\pm0.0002}$ / $0.1851_{\pm0.0001}$ | $0.1831_{\pm0.0014}$ / $0.1867_{\pm0.0006}$ | 0.001 | 1e-6 | 320 |
| | SFT → offpolicy–random | $0.1759_{\pm0.0017}$ / $0.1790_{\pm0.0005}$ | $0.1962_{\pm0.0052}$ / $0.1994_{\pm0.0036}$ | 0.001 | 1e-6 | 764 |

**Table 13:** Mistral 7B Instruct v0.2, Magpie-Air dataset.

| Method | Setting | Reward (valid/test) | LC-Reward (valid/test) | $\beta$ | LR | Step |
|---|---|---|---|---|---|---|
| **Base** | | $0.1574_{\pm0.0004}$ / $0.1598_{\pm0.0006}$ | $0.1572_{\pm0.0004}$ / $0.1598_{\pm0.0008}$ | | | |
| | SFT | $\underline{0.1611}_{\pm0.0005}$ / $0.1633_{\pm0.0006}$ | $\underline{0.1610}_{\pm0.0004}$ / $0.1633_{\pm0.0004}$ | | 5e-7 | 764 |
| **DPO** | offline | $0.1764_{\pm0.0001}$ / $0.1783_{\pm0.0003}$ | $0.1768_{\pm0.0023}$ / $0.1775_{\pm0.0011}$ | 0.03 | 3e-7 | 764 |
| | offpolicy–best | $0.1803_{\pm0.0002}$ / $0.1820_{\pm0.0003}$ | $0.1761_{\pm0.0005}$ / $0.1777_{\pm0.0012}$ | 0.03 | 3e-7 | 160 |
| | offpolicy–random | $0.1791_{\pm0.0001}$ / $0.1810_{\pm0.0000}$ | $0.1787_{\pm0.0012}$ / $0.1795_{\pm0.0010}$ | 0.01 | 3e-7 | 320 |
| | SFT → offline* | $\underline{0.1872}_{\pm0.0002}$ / $0.1901_{\pm0.0001}$ | $\underline{0.1873}_{\pm0.0003}$ / $0.1898_{\pm0.0003}$ | 0.03 | 3e-7 | 320 |
| | SFT → offpolicy–best | $\underline{0.1869}_{\pm0.0001}$ / $0.1891_{\pm0.0002}$ | $0.1844_{\pm0.0004}$ / $0.1856_{\pm0.0012}$ | 0.03 | 3e-7 | 160 |
| | SFT → offpolicy–random | $0.1859_{\pm0.0004}$ / $0.1883_{\pm0.0002}$ | $0.1865_{\pm0.0006}$ / $0.1879_{\pm0.0013}$ | 0.03 | 3e-7 | 480 |
| **SimPO** | offline | $0.1630_{\pm0.0001}$ / $0.1654_{\pm0.0006}$ | $0.1630_{\pm0.0001}$ / $0.1653_{\pm0.0008}$ | 10 | 1e-7 | 160 |
| | offpolicy–best | $0.1678_{\pm0.0003}$ / $0.1704_{\pm0.0005}$ | $0.1679_{\pm0.0003}$ / $0.1712_{\pm0.0005}$ | 2 | 1e-7 | 160 |
| | offpolicy–random | $0.1786_{\pm0.0004}$ / $0.1807_{\pm0.0003}$ | $0.1720_{\pm0.0008}$ / $0.1739_{\pm0.0004}$ | 2 | 3e-7 | 160 |
| | SFT → offline | $\underline{0.1855}_{\pm0.0001}$ / $0.1880_{\pm0.0001}$ | $\underline{0.1853}_{\pm0.0002}$ / $0.1876_{\pm0.0002}$ | 10 | 1e-7 | 160 |
| | SFT → offpolicy–best | $0.1778_{\pm0.0005}$ / $0.1802_{\pm0.0003}$ | $0.1764_{\pm0.0007}$ / $0.1786_{\pm0.0006}$ | 10 | 1e-7 | 160 |
| | SFT → offpolicy–random* | $\underline{0.1859}_{\pm0.0005}$ / $0.1884_{\pm0.0001}$ | $\underline{0.1851}_{\pm0.0006}$ / $0.1879_{\pm0.0012}$ | 10 | 3e-7 | 160 |
| **REBEL** | offline | $0.1727_{\pm0.0008}$ / $0.1734_{\pm0.0006}$ | $0.1789_{\pm0.0021}$ / $0.1821_{\pm0.0032}$ | 1e-4 | 3e-7 | 764 |
| | offpolicy–best | $0.1675_{\pm0.0007}$ / $0.1719_{\pm0.0003}$ | $0.1806_{\pm0.0004}$ / $0.1831_{\pm0.0011}$ | 1e-4 | 1e-6 | 160 |
| | offpolicy–random | $0.1759_{\pm0.0001}$ / $0.1784_{\pm0.0003}$ | $0.1737_{\pm0.0003}$ / $0.1764_{\pm0.0001}$ | 1e-4 | 3e-7 | 160 |
| | SFT → offline* | $\underline{0.1845}_{\pm0.0001}$ / $0.1864_{\pm0.0002}$ | $\underline{0.1868}_{\pm0.0006}$ / $0.1884_{\pm0.0002}$ | 1e-4 | 3e-7 | 320 |
| | SFT → offpolicy–best | $0.1757_{\pm0.0002}$ / $0.1771_{\pm0.0001}$ | $0.1811_{\pm0.0019}$ / $0.1818_{\pm0.0003}$ | 1e-4 | 1e-7 | 480 |
| | SFT → offpolicy–random | $0.1832_{\pm0.0001}$ / $0.1854_{\pm0.0002}$ | $0.1895_{\pm0.0003}$ / $0.1913_{\pm0.0010}$ | 1e-4 | 3e-7 | 320 |
| **QRPO** | offline | $0.1761_{\pm0.0003}$ / $0.1782_{\pm0.0007}$ | $0.1784_{\pm0.0004}$ / $0.1798_{\pm0.0006}$ | 0.003 | 1e-6 | 764 |
| | offpolicy–best | $0.1732_{\pm0.0004}$ / $0.1757_{\pm0.0002}$ | $0.1731_{\pm0.0003}$ / $0.1762_{\pm0.0005}$ | 0.003 | 1e-6 | 480 |
| | offpolicy–random | $0.1657_{\pm0.0004}$ / $0.1695_{\pm0.0005}$ | $0.1640_{\pm0.0009}$ / $0.1676_{\pm0.0017}$ | 0.003 | 3e-7 | 764 |
| | SFT → offline* | $\underline{0.1863}_{\pm0.0001}$ / $0.1886_{\pm0.0002}$ | $\underline{0.1866}_{\pm0.0002}$ / $0.1893_{\pm0.0003}$ | 3e-4 | 3e-7 | 320 |
| | SFT → offpolicy–best | $0.1822_{\pm0.0003}$ / $0.1849_{\pm0.0004}$ | $0.1822_{\pm0.0003}$ / $0.1849_{\pm0.0003}$ | 0.003 | 1e-7 | 764 |
| | SFT → offpolicy–random | $0.1771_{\pm0.0002}$ / $0.1800_{\pm0.0003}$ | $0.1778_{\pm0.0004}$ / $0.1810_{\pm0.0006}$ | 0.003 | 1e-6 | 764 |

**Table 14:** Llama 8B Tülu 3 SFT, UltraFeedback dataset.

| Method | Setting | Reward (valid/test) | LC-Reward (valid/test) | $\beta$ | LR | Step |
|---|---|---|---|---|---|---|
| **Base** | | $\underline{0.1278}_{\pm 0.0012}$ / $0.1293_{\pm 0.0005}$ | $\underline{0.1279}_{\pm 0.0012}$ / $0.1294_{\pm 0.0006}$ | | | |
| | SFT | $0.1201_{\pm 0.0012}$ / $0.1219_{\pm 0.0007}$ | $0.1202_{\pm 0.0012}$ / $0.1219_{\pm 0.0008}$ | | 5e-7 | 476 |
| **DPO** | offline | $0.1445_{\pm 0.0002}$ / $0.1452_{\pm 0.0005}$ | $0.1445_{\pm 0.0001}$ / $0.1455_{\pm 0.0006}$ | 0.01 | 1e-6 | 300 |
| | offpolicy–best* | $\underline{0.1480}_{\pm 0.0004}$ / $0.1491_{\pm 0.0001}$ | $\underline{0.1480}_{\pm 0.0004}$ / $0.1493_{\pm 0.0001}$ | 0.01 | 3e-7 | 300 |
| | offpolicy–random | $0.1470_{\pm 0.0002}$ / $0.1485_{\pm 0.0003}$ | $0.1470_{\pm 0.0002}$ / $0.1489_{\pm 0.0002}$ | 0.01 | 1e-6 | 400 |
| | SFT → offline | $0.1439_{\pm 0.0003}$ / $0.1443_{\pm 0.0003}$ | $0.1438_{\pm 0.0002}$ / $0.1443_{\pm 0.0002}$ | 0.01 | 1e-6 | 300 |
| | SFT → offpolicy–best | $0.1442_{\pm 0.0002}$ / $0.1452_{\pm 0.0002}$ | $0.1443_{\pm 0.0002}$ / $0.1451_{\pm 0.0002}$ | 0.01 | 1e-6 | 300 |
| | SFT → offpolicy–random | $0.1430_{\pm 0.0004}$ / $0.1436_{\pm 0.0003}$ | $0.1430_{\pm 0.0002}$ / $0.1436_{\pm 0.0001}$ | 0.01 | 1e-6 | 300 |
| **SimPO** | offline | $0.1516_{\pm 0.0002}$ / $0.1528_{\pm 0.0005}$ | $0.1512_{\pm 0.0005}$ / $0.1535_{\pm 0.0006}$ | 2 | 1e-6 | 200 |
| | offpolicy–best* | $\underline{0.1525}_{\pm 0.0004}$ / $0.1539_{\pm 0.0002}$ | $\underline{0.1524}_{\pm 0.0004}$ / $0.1535_{\pm 0.0009}$ | 2 | 1e-6 | 200 |
| | offpolicy–random | $0.1507_{\pm 0.0004}$ / $0.1522_{\pm 0.0001}$ | $0.1505_{\pm 0.0003}$ / $0.1515_{\pm 0.0008}$ | 2 | 1e-6 | 476 |
| | SFT → offline | $\underline{0.1518}_{\pm 0.0004}$ / $0.1535_{\pm 0.0004}$ | $0.1515_{\pm 0.0002}$ / $0.1523_{\pm 0.0004}$ | 2 | 1e-6 | 200 |
| | SFT → offpolicy–best | $0.1483_{\pm 0.0004}$ / $0.1500_{\pm 0.0003}$ | $0.1475_{\pm 0.0005}$ / $0.1493_{\pm 0.0002}$ | 2 | 3e-7 | 476 |
| | SFT → offpolicy–random | $0.1501_{\pm 0.0003}$ / $0.1508_{\pm 0.0005}$ | $0.1490_{\pm 0.0003}$ / $0.1502_{\pm 0.0006}$ | 10 | 1e-6 | 200 |
| **REBEL** | offline | $\underline{0.1468}_{\pm 0.0005}$ / $0.1478_{\pm 0.0003}$ | $0.1468_{\pm 0.0005}$ / $0.1473_{\pm 0.0004}$ | 1e-4 | 3e-7 | 200 |
| | offpolicy–best | $\underline{0.1474}_{\pm 0.0004}$ / $0.1487_{\pm 0.0001}$ | $0.1473_{\pm 0.0004}$ / $0.1489_{\pm 0.0004}$ | 1e-6 | 3e-7 | 200 |
| | offpolicy–random* | $\underline{0.1478}_{\pm 0.0001}$ / $0.1487_{\pm 0.0005}$ | $\underline{0.1478}_{\pm 0.0001}$ / $0.1488_{\pm 0.0004}$ | 1e-6 | 1e-6 | 100 |
| | SFT → offline | $0.1448_{\pm 0.0004}$ / $0.1465_{\pm 0.0005}$ | $0.1443_{\pm 0.0002}$ / $0.1461_{\pm 0.0004}$ | 1e-4 | 1e-6 | 200 |
| | SFT → offpolicy–best | $0.1403_{\pm 0.0002}$ / $0.1413_{\pm 0.0001}$ | $0.1404_{\pm 0.0003}$ / $0.1417_{\pm 0.0002}$ | 1e-4 | 1e-6 | 300 |
| | SFT → offpolicy–random | $0.1413_{\pm 0.0002}$ / $0.1427_{\pm 0.0003}$ | $0.1413_{\pm 0.0002}$ / $0.1427_{\pm 0.0004}$ | 1e-4 | 1e-6 | 300 |
| **QRPO** | offline | $0.1454_{\pm 0.0005}$ / $0.1466_{\pm 0.0001}$ | $0.1394_{\pm 0.0010}$ / $0.1426_{\pm 0.0025}$ | 0.003 | 1e-7 | 476 |
| | offpolicy–best | $\underline{0.1503}_{\pm 0.0007}$ / $0.1508_{\pm 0.0005}$ | $0.1505_{\pm 0.0004}$ / $0.1514_{\pm 0.0003}$ | 3e-4 | 3e-7 | 100 |
| | offpolicy–random | $\underline{0.1502}_{\pm 0.0008}$ / $0.1505_{\pm 0.0006}$ | $0.1506_{\pm 0.0007}$ / $0.1505_{\pm 0.0002}$ | 3e-4 | 3e-7 | 100 |
| | SFT → offline* | $\underline{0.1497}_{\pm 0.0003}$ / $0.1504_{\pm 0.0008}$ | $\underline{0.1517}_{\pm 0.0001}$ / $0.1556_{\pm 0.0017}$ | 0.001 | 3e-7 | 100 |
| | SFT → offpolicy–best | $0.1430_{\pm 0.0002}$ / $0.1438_{\pm 0.0002}$ | $0.1431_{\pm 0.0003}$ / $0.1438_{\pm 0.0002}$ | 0.001 | 1e-6 | 200 |
| | SFT → offpolicy–random | $0.1416_{\pm 0.0009}$ / $0.1429_{\pm 0.0003}$ | $0.1432_{\pm 0.0007}$ / $0.1429_{\pm 0.0017}$ | 0.003 | 3e-7 | 100 |

**Table 15:** Mistral 7B Instruct v0.2, UltraFeedback dataset.

| Method | Setting | Reward (valid/test) | LC-Reward (valid/test) | $\beta$ | LR | Step |
|---|---|---|---|---|---|---|
| **Base** | | $\underline{0.1335}_{\pm 0.0004}$ / $0.1349_{\pm 0.0009}$ | $\underline{0.1333}_{\pm 0.0004}$ / $0.1348_{\pm 0.0009}$ | | | |
| | SFT | $0.1243_{\pm 0.0008}$ / $0.1258_{\pm 0.0013}$ | $0.1237_{\pm 0.0011}$ / $0.1254_{\pm 0.0015}$ | | 5e-7 | 476 |
| **DPO** | offline | $0.1368_{\pm 0.0001}$ / $0.1381_{\pm 0.0002}$ | $0.1366_{\pm 0.0002}$ / $0.1379_{\pm 0.0004}$ | 0.01 | 1e-7 | 476 |
| | offpolicy–best | $\underline{0.1475}_{\pm 0.0002}$ / $0.1484_{\pm 0.0002}$ | $0.1456_{\pm 0.0001}$ / $0.1484_{\pm 0.0004}$ | 0.03 | 1e-6 | 100 |
| | offpolicy–random | $0.1441_{\pm 0.0001}$ / $0.1458_{\pm 0.0002}$ | $0.1433_{\pm 0.0002}$ / $0.1449_{\pm 0.0002}$ | 0.03 | 3e-7 | 300 |
| | SFT → offline | $0.1402_{\pm 0.0005}$ / $0.1417_{\pm 0.0005}$ | $0.1410_{\pm 0.0005}$ / $0.1413_{\pm 0.0005}$ | 0.01 | 1e-7 | 300 |
| | SFT → offpolicy–best* | $\underline{0.1469}_{\pm 0.0003}$ / $0.1480_{\pm 0.0007}$ | $\underline{0.1459}_{\pm 0.0003}$ / $0.1465_{\pm 0.0008}$ | 0.03 | 1e-6 | 100 |
| | SFT → offpolicy–random | $0.1441_{\pm 0.0001}$ / $0.1454_{\pm 0.0004}$ | $0.1430_{\pm 0.0003}$ / $0.1440_{\pm 0.0006}$ | 0.01 | 3e-7 | 476 |
| **SimPO** | offline | $0.1382_{\pm 0.0001}$ / $0.1392_{\pm 0.0001}$ | $0.1364_{\pm 0.0002}$ / $0.1395_{\pm 0.0001}$ | 2.5 | 3e-7 | 476 |
| | offpolicy–best | $0.1433_{\pm 0.0003}$ / $0.1450_{\pm 0.0003}$ | $0.1443_{\pm 0.0005}$ / $0.1450_{\pm 0.0023}$ | 10 | 3e-7 | 200 |
| | offpolicy–random | $0.1419_{\pm 0.0003}$ / $0.1431_{\pm 0.0004}$ | $0.1419_{\pm 0.0005}$ / $0.1432_{\pm 0.0017}$ | 2.5 | 3e-7 | 476 |
| | SFT → offline | $0.1415_{\pm 0.0005}$ / $0.1426_{\pm 0.0002}$ | $0.1408_{\pm 0.0005}$ / $0.1416_{\pm 0.0001}$ | 2 | 1e-7 | 400 |
| | SFT → offpolicy–best* | $\underline{0.1460}_{\pm 0.0002}$ / $0.1472_{\pm 0.0005}$ | $\underline{0.1467}_{\pm 0.0003}$ / $0.1478_{\pm 0.0007}$ | 10 | 3e-7 | 200 |
| | SFT → offpolicy–random | $0.1435_{\pm 0.0005}$ / $0.1441_{\pm 0.0004}$ | $0.1445_{\pm 0.0005}$ / $0.1447_{\pm 0.0004}$ | 10 | 3e-7 | 476 |
| **REBEL** | offline | $0.1364_{\pm 0.0002}$ / $0.1384_{\pm 0.0006}$ | $0.1372_{\pm 0.0006}$ / $0.1393_{\pm 0.0003}$ | 1e-4 | 1e-7 | 100 |
| | offpolicy–best | $0.1376_{\pm 0.0003}$ / $0.1397_{\pm 0.0001}$ | $0.1368_{\pm 0.0003}$ / $0.1392_{\pm 0.0002}$ | 1e-4 | 3e-7 | 100 |
| | offpolicy–random | $\underline{0.1426}_{\pm 0.0002}$ / $0.1444_{\pm 0.0001}$ | $0.1426_{\pm 0.0003}$ / $0.1442_{\pm 0.0001}$ | 1e-4 | 1e-6 | 100 |
| | SFT → offline | $0.1386_{\pm 0.0006}$ / $0.1395_{\pm 0.0005}$ | $0.1397_{\pm 0.0001}$ / $0.1396_{\pm 0.0005}$ | 1e-4 | 1e-7 | 200 |
| | SFT → offpolicy–best | $0.1398_{\pm 0.0000}$ / $0.1409_{\pm 0.0001}$ | $0.1398_{\pm 0.0000}$ / $0.1408_{\pm 0.0001}$ | 1e-4 | 1e-7 | 200 |
| | SFT → offpolicy–random* | $\underline{0.1437}_{\pm 0.0007}$ / $0.1457_{\pm 0.0007}$ | $\underline{0.1445}_{\pm 0.0008}$ / $0.1466_{\pm 0.0006}$ | 1e-4 | 1e-6 | 100 |
| **QRPO** | offline | $0.1413_{\pm 0.0002}$ / $0.1424_{\pm 0.0003}$ | $0.1413_{\pm 0.0003}$ / $0.1427_{\pm 0.0003}$ | 0.003 | 1e-6 | 476 |
| | offpolicy–best* | $\underline{0.1458}_{\pm 0.0006}$ / $0.1469_{\pm 0.0007}$ | $\underline{0.1460}_{\pm 0.0005}$ / $0.1470_{\pm 0.0007}$ | 0.003 | 1e-6 | 200 |
| | offpolicy–random | $0.1335_{\pm 0.0004}$ / $0.1349_{\pm 0.0009}$ | $0.1333_{\pm 0.0004}$ / $0.1348_{\pm 0.0009}$ | 0.001 | 1e-6 | 0 |
| | SFT → offline | $0.1411_{\pm 0.0001}$ / $0.1421_{\pm 0.0007}$ | $0.1418_{\pm 0.0002}$ / $0.1422_{\pm 0.0008}$ | 0.003 | 3e-7 | 400 |
| | SFT → offpolicy–best | $0.1421_{\pm 0.0002}$ / $0.1429_{\pm 0.0004}$ | $0.1422_{\pm 0.0003}$ / $0.1431_{\pm 0.0004}$ | 0.003 | 1e-6 | 200 |
| | SFT → offpolicy–random | $0.1243_{\pm 0.0008}$ / $0.1258_{\pm 0.0013}$ | $0.1237_{\pm 0.0011}$ / $0.1254_{\pm 0.0015}$ | 0.001 | 1e-6 | 0 |

**Table 16:** Llama 8B Tülu 3 SFT, LeetCodeDataset dataset. The datasets contain 10 preference pairs per prompt, generated from 20 reference completions annotated with the test-case pass rate reward. QRPO uses the 20 reference completions and their rewards to compute the quantile reward.

| Method | Setting | Reward (valid/test) | $\beta$ | LR | Step |
|--------|---------|---------------------|---------|-----|------|
| **Base** | | $\underline{0.206}_{\pm 0.019}$ / $0.209_{\pm 0.005}$ | | | |
| | SFT | $\underline{0.215}_{\pm 0.006}$ / $0.188_{\pm 0.011}$ | | 5e-7 | 100 |
| **DPO** | SFT $\rightarrow$ offpolicy–random | $\underline{0.317}_{\pm 0.006}$ / $0.302_{\pm 0.014}$ | 0.01 | 1e-6 | 80 |
| **SimPO** | SFT $\rightarrow$ offpolicy–random | $\underline{0.280}_{\pm 0.037}$ / $0.223_{\pm 0.014}$ | 10 | 3e-7 | 160 |
| **REBEL** | SFT $\rightarrow$ offpolicy–random | $\underline{0.301}_{\pm 0.017}$ / $0.261_{\pm 0.018}$ | 0.01 | 1e-6 | 160 |
| **QRPO** | SFT $\rightarrow$ offpolicy–random | $\underline{0.316}_{\pm 0.020}$ / $0.327_{\pm 0.010}$ | 0.01 | 1e-6 | 200 |

**(a)** Llama 8B on Magpie-Air (Figure 4)

**(b)** Mistral 7B on Magpie-Air

**(c)** Llama 8B on UltraFeedback

**(d)** Mistral 7B on UltraFeedback

**Figure 15:** Length bias curves in Figure 4 for all of the model and dataset configurations. Each figure shows the length bias of the best checkpoint of each algorithm after training the given initial model on the given dataset. Trend lines capture a first order fit of the scatter points for each algorithm and the central point on the line indicates the average normalized completion length for the algorithm. The legend also indicates a Spearman rank correlation.

