# OpenReview forum: "Quantile Reward Policy Optimization: Alignment with Pointwise Regression and Exact Partition Functions"
_NeurIPS.cc/2025/Conference — NeurIPS 2025 poster_

### Official Review · Reviewer_Ghhc · 2025-06-29

**Clarity:** 3
**Significance:** 2
**Originality:** 2
**Rating:** 5
**Confidence:** 4

**Summary:**

The paper proposes a LLM alignment algorithm in the 'policy fitting' (i.e. DPO style) class of methods, that uses pointwise absolute rewards and offline data. Quantile Reward Policy Optimization (QRPO) first transforms each raw reward into its percentile within the reference model's reward distribution, yielding a transformed reward on $[0, 1]$. Due to this transform, the KL-regularized RL objective admits a closed-form log-likelihood target and removes noisy Monte-Carlo partition function estimates that certain prior methods suffered from. The authors propose a straightforward supervised regression between this quantile reward and the model's log-probability ratio to the reference policy. The method eliminates the need for a critic, pairwise preferences and depending on the setting the need for a reward model (using instead verifiable rewards, unit test pass rates, single human scores). The authors run experiments on instruction-following, maths and coding tasks, reporting that QRPO matches or surpasses DPO and other baselines with lower length bias. The learning signal can however plateau once the learned policy overtakes the cached reference completions, and reward signal magnitude is effectively discarded.

**Questions:**

1. Have you quantified how quickly gradients vanish when the learned policy's reward surpasses all reference completions' rewards (i.e. $\mathcal{R}(x, y') \le \mathcal{R}(x, y)\ \forall y' \sim \pi_{\text{ref}}(y' \vert x)$? Adding this detail, or further illustrating how QRPO might be used in more online settings (that is, periodically updating the reference and re-sampling the empirical CDF) would clarify its practicality in long-training regimes or tasks firmly outside the initial reference model's distribution.
2. How do both RLOO and GRPO compare to QRPO, seeing as they seem to occupy similar design spaces to (they accept point-wise rewards and address gradient variance reduction)? I am not expecting you to re-run experiments with these on account of the additional work, but if you can that would be great, or failing that a short discussion of their characteristics (compute requirements, ability to run offline or online, expected performance, etc) would be a nice addition to the paper.
3. What is the impact of discarding the scale of the reward? In many environments, the pointwise reward contains magnitude information that can help provide richer learning signals, for instance differentiating a good from a great completion (distance to goal, safety penalties, code performance). As I understand it, the price of the quantile transform used to get a closed-form partition function is that we discard the reward magnitude information and instead only rank it relative to the reference model's completions (i.e. the policy is merely pushed to outrank its own reference model). Could you comment further on the implications of this; for example washing out extreme negatives (to the detriment of safety) or extreme positives (to the detriment of learning speed).

**Ethical Concerns:**

["NO or VERY MINOR ethics concerns only"]

**Final Justification:**

My initial evaluation of the work contained some misunderstanding about the originality and significance of the work. This was clarified in the discussion, and I consider this work to be a useful and novel offline alignment algorithm for cases where only pointwise rewards are available. The paper is technically sound with a good evaluation in the offline setting. While I do believe that the paper would be improved by addressing a couple of points which the authors consider out of scope and defer to future work (expanding on how and when it is useful to apply a transform on top of the quantile reward, as well as performing an evaluation online to match the claims that this method can work effectively in online settings as well as offline), I also believe that the contributions made as the paper stands are substantial enough to recommend acceptance.

**Limitations:**

While the two main limitations with the work are brought up (the quantile reward re-shaping the reward distribution, offline only focus), this is done so with a brevity that may not be commensurate with their significance.

While a more rigorous investigation of these may be postponed for future work, I do believe it would be valuable to discuss the potential drawbacks of the re-shaped reward distribution and loss of reward magnitude in the main text in some greater depth, dwelling on potential mitigation strategies for practitioners.

**Paper Formatting Concerns:**

Figure 2 perhaps ought to be typeset or re-drawn with vector graphics rather than hand drawn.

**Quality:**

4

**Strengths And Weaknesses:**

# Quality
The paper contains a good section on preliminary work, with a clean and relatively easy to follow derivation from the KL-regularized RL objective to their closed form regression loss. The auxiliary proofs in the appendix for the uniform CDF and analytic partition function on which the work rests are clean and re-use standard probability results.

The experimental work is careful and high-quality, and appears to follow rigorous methodology to appropriately tune baselines, re-use appropriate reward models, and report numbers with uncertainty estimates throughout.

Given the prevalence of 'policy improvement' methods such as GRPO and RLOO, which are prominently mentioned in Table 1, and serve similar purposes as post-training algorithms that use pointwise rewards and work online (and increasingly offline too), I believe these would make valuable additions to the comparisons in the paper. While I recognize the additional work and perhaps scope creep - and hence I won't consider this point strongly in my score - it would be very interesting to see how QRPO compares with GRPO in the Llama 8B LeetCode run reported in Table 2. This seems particularly relevant owing to the similarities one might draw between the computational requirements of both methods: with the group rollouts in GRPO and and $n$ completions $y_{i, j} \sim \pi_{\text{ref}}(\cdot \vert x_{i})$ used in QRPO.

# Clarity

The paper is for the most part clearly written and easy to follow. Figure 2 is somewhat hard to read and would benefit from being typeset properly.

# Significance

The proposed method does introduce some genuinely useful properties: dropping the pairwise preference data requirement of many previous direct alignment algorithms is beneficial for applications or data pipelines that only measure a pointwise reward signal (thumbs up/down, likert scale), while the offline nature of the algorithm may help to amortize pre-computation costs, and being less prone to length bias is also beneficial. However, there already exist prior alignment algorithms with these qualitative properties for practitioners. Much of the introduction and 3-axis taxonomy is seemingly spent to explain why this work is unique: indeed, offline policy-fitting, with absolute scalar rewards and no auxiliary networks (for estimating $Z$, a critic, or otherwise) *is* useful, but it is only narrowly unique in the alignment space.

However, the reported performance is at times only marginally superior to DPO or SimPO which slightly limits the significance of the work. Moreover, the exclusive focus on offline policy fitting comparisons, while in the authors' own words "*PF methods are losing applicability*" in the face of progress such as agent reasoning on verifiable reward tasks also limits the significance of the work: while the work addresses the use of absolute reward signals, it seems likely that online data collection will continue to play a significant role in post-training as the field appears to be moving from aligning chatbots to training reasoning-style agents.

# Originality

The main original contribution in QRPO appears to be the "quantile reward trick" (Eqns 7 -- 10), where the quantile transformation makes the exponentiated reward's partition function prompt-independent, which turns the KL-regularized RL optimum into a closed-form regression target.  While there are similar sounding techniques in other areas of ML (e.g. probability-integral / CDF transforms are used in density estimation methods like normalizing flows so the base distribution's log-density has a tractable normalizer, or quantile regression in QR-DQN), the approach in this paper is original and solves a problem in direct alignment algorithms. Moreover, the approach set out in this paper (i.e. converting an arbitrary scalar reward into a bounded, uniform signal whose partition function is a constant) could be re-used elsewhere where we face energy-based or KL objectives with nuisance normalizers.

**Nits:**
- line 56: "effective on a scale" -> "effectiive at scale"
- line 301: "we choose to compare to DPO"
- Paragraph at line 496: repeated sentence
- Eq 20: d(F_{ref}(r)) -> F_{ref}(r, x)

---

> ### Author Rebuttal · Authors · 2025-07-31
>
> We sincerely thank the reviewer for the detailed and constructive feedback, as well as for highlighting the strengths of our work.
>
> Below, we address each point raised:
> ___
> ## Vital role of offline RL algorithms for post-training pipelines
> We specifically focused on the offline regime because we believe that a major reason for the cost inefficiency of online RL post-training is the **inability to scale** by leveraging diverse offline data (i.e., previously collected samples or offline datasets). This is currently one of the **main bottlenecks** in RL post-training pipelines. For example, recent large-scale RL post-training efforts (e.g., Grok-4) required compute budgets **comparable to pre-training**, largely due to the sequential and expensive nature of online data generation.
>
> Thus, even though online data collection will likely continue to play a significant role in post-training pipelines, the effective scaling of training requires an offline-capable algorithm that can significantly amortize this data generation cost by reusing the data generated in previous iterations.
> ___
> ## Policy improvement algorithms cannot be a viable option for offline training
> Policy improvement methods (e.g., PPO, GRPO, RLOO) are **inherently online** algorithms. While slight off-policy training is possible via importance sampling corrections, this approach introduces high-variance gradient estimates, making optimization unstable at strong off-policiness. More importantly, **true offline** training is **still infeasible** with importance sampling because it requires knowing the sampling probabilities of each data point under the behavior policy, which is typically unknown for offline datasets collected from arbitrary sources.
>
> Therefore, we would be grateful if the reviewer could provide examples for their claim that "there already exist prior alignment algorithms with these qualitative properties for practitioners" (in the context of offline pointwise algorithms).
>
> In addition, could the reviewer elaborate on "is only narrowly unique in the alignment space"?
> Given that it covers an essential gap in stable, offline (or highly off-policy) pointwise methods, which can largely impact the scaling of RL pipelines.
>
> ---
>
> ## Results interpretation
> In some cases, performance can seem marginally superior to DPO or SimPO for some metrics, but we elaborate on their significance, noting that we primarily rely on AlpacaEval 2, which evaluates downstream generalization on an unseen dataset with GPT-4 as a judge and is a more robust proxy for human preferences than the reward model. Importantly, we focus on its Length-Controlled (LC) score as it both mitigates length bias and shows the highest correlation with human judgments.
> Then come ArmoRM rewards, which correlate with AlpacaEval2 but can be "hacked" by trained policies, making them less reliable. Among average reward scores, LC-ArmoRM scores remain more informative since they reduce verbosity bias.
>
> With this metric priority in mind,
> - On Llama 8B SFT, QRPO consistently outperforms all baselines. On the high-quality Magpie-Air dataset, the LC-AlpacaEval2 gap between QRPO and SimPO is of the same magnitude as the well-established DPO→SimPO gap, which is widely regarded as a substantial improvement in general chat tasks, confirming that QRPO delivers a significant gain.
> - On the Mistral Instruct model, we likely observe capacity saturation. After extensive hyperparameter tuning, DPO, SimPO, and QRPO achieve **nearly identical scores** on UltraFeedback, with differences within the noise level. Importantly, our results for DPO and SimPO significantly surpass those originally reported in the SimPO paper (Meng et. al., 2024), indicating we have pushed all methods considerably further than previous studies. Thus, the observed saturation likely reflects that the limits of Mistral Instruct’s performance were reached, rather than a limitation specific to QRPO.
> - On Magpie-Air with Mistral, similar saturation effects appear, with DPO and SimPO exploiting length bias for non-significant AlpacaEval gains. LC-AlpacaEval scores for QRPO remain higher, consistent with its improved robustness to verbosity bias (Fig. 4).
> ---
> ## Answers to the questions
> **Q1**
>
> First, note that **gradient vanishing does not occur simply due to quantile reward saturation ($\mathcal{R}_q(x, y)=1$)**. Even when $\mathcal{R}_ q=1$, gradients remain non-zero unless the ratio $\pi_{\theta}/\pi_{\text{ref}}=1/\beta$ is exactly reached (from plugging $\mathcal{R}_q=1$ into the optimal policy expression and approximating $Z_q(x)\approx \beta e^{1/\beta}$ for small $\beta$). Thus, gradient vanishing coincides strictly with policy convergence, not reward saturation.
>
> However, widespread quantile reward saturation (many samples achieving $\mathcal{R}_q=1$) can still reduce the magnitude of the reward signal, potentially slowing training. Clarifying when this can occur:
> * In the **offline setting**, quantile rewards are computed from a static dataset that is crafted by the user. If this dataset contains samples that would saturate the quantile, then a preliminary SFT on this data can be performed to reduce the gap/distribution shift of the initial checkpoint/reference policy for QRPO.
> * In the **online setting**, improved policy outputs may frequently exceed reference completions, causing a quantile reward saturation.
>
> However, **this saturation coincides with (or follows) convergence to the optimal policy**. This is because the optimal policy under the quantile reward remains **supported within the reference distribution**, due to the “best-of-N” nature of this optimal distribution (Appendix M).
>
> As a result, the policy cannot easily escape the support of $\pi_{\text{ref}}$, and the model is unlikely to produce completions that consistently outperform all reference completions. Hence, full quantile reward saturation typically does not occur before convergence.
> We also validate this in practice; our experiments never exceeded an average quantile reward of 0.95–0.98, preserving informative gradients.
>
> Moreover, one can still improve beyond the optimal policy given a reference by either lowering $\beta$ or performing several iterations
> periodically updating the reference model to the most recent trained policy and re-estimating the quantile CDF.
> This is a common recipe used with policy fitting methods, for example, used with DPO for the Llama 3 models.
> ___
> **Q2**
>
> We do not consider RLOO and GRPO to occupy the same design space as QRPO, as neither supports fully offline training, which is a key property of QRPO. Nevertheless, it is useful to compare their characteristics in the online setting.
>
> We opted for a rigorous experimental protocol that required 50k GPU hours to complete to show the effectiveness of QRPO in offline/off-policy regimes and support its unique contribution.
> As the reviewer acknowledges, adding a thorough evaluation of online and semi-online regimes is outside our scope and would require 2 times more resources due to online sampling costs. We therefore provide a theoretical comparison:
> 1. Theoretical Similarities:
> - QRPO’s gradient in the online case is equivalent to a **policy gradient with a baseline**, making it closely related to GRPO and RLOO.
> - The main difference lies in the reward transformation: QRPO optimizes a quantile-based reward, while GRPO and RLOO use the raw signal.
> - Both approaches can be brought closer by either applying an inverse-CDF transform (Eq. 12) to QRPO’s quantiles to approximate the raw scale or, conversely, by using quantile rewards in GRPO.
>
> 2. Differences: Optimal Baseline in QRPO
> - QRPO provides a closed-form baseline, $\beta \log{Z_ q}$, which equals the expected reward under the optimal policy:
> $\beta \log{Z_q} = \mathbb{E}_ {y \sim \pi^ * (\cdot \mid y)} [r(x,y)], \quad r(x,y) = \mathcal{R}_ q (x,y) - \beta \log{\frac{\pi_ \theta(y \mid x)}{\pi_ {ref}(y \mid x)}}$. This guarantees minimum gradient variance as the learned policy approaches the optimum, while in GRPO and RLOO baseline is typically an estimator with additional variance.
> - This baseline is what allows QRPO to work both online and offline, bridging the policy improvement gradient and the policy fitting gradient.
>
> 3. Online/Offline Compute Efficiency:
>  - In the offline case, QRPO is unmatched, the quantile rewards and the reference model probabilities can be precomputed, therefore only requiring to have the training policy during training. These are the same requirements as SFT and represent ~3x less memory compared to GRPO (at least online) which needs the reference model, sampler, and training policy.
> - Online:
>   - QRPO can precompute $n$ reference completions per prompt once (unless the reference is updated), reusing them throughout training.
>   - GRPO and RLOO require fresh completions at every gradient step, leading to considerably higher compute costs due to repeated online sampling.
> ___
> **Q3**
>
> We distinguish two distinct interpretations of the question: (i) preserving different reward ranges for different prompts, (ii) extracting the information from outliers.
> - (i): QRPO allows for a reward range preserving by applying the additional transform on top of a quantile reward (as proposed in Eq. 12). This transformation can be prompt-specific, e.g., it can be a multiplication by the std of the reference model rewards for each prompt.
> - (ii): QRPO is indeed insensitive to the outliers with respect to the reference rewards distribution (i.e., it can't distinguish them from normal boundary values). However, one can try to mitigate this issue by picking a special transformation function (eq. 12). For instance, if one needs to account for negative outliers (i.e., safety penalties), one can use a log transformation, thus heavily penalizing any sample with near 0 quantile reward.
>
> Finally, we note that other algorithms based on normalization (e.g., GRPO) also discard the scale information and reduce sensitivity to outliers.

---

> > ### Comment · Reviewer_Ghhc · 2025-08-04
> >
> > I thank the authors for their detailed response and clarifications on points of misunderstanding.
> >
> > First, on the role of offline RL algorithms for post-training pipelines, I agree with the authors' characterisation that offline RL algorithms play a vital role in RLHF-style alignment work. In this view, the alignment algorithm might be seen to 'filter' the pre-trained LLM's output according to some preferences.
> >
> > However, I will gently push back on the negative characterisation of online RL algorithms as being burdened by cost, and viewing this often significant post-training cost of e.g. Grok-4 as a failure of online methods. I would instead suggest that these methods present a path towards scalable, open-ended synthetic data generation and a way to effectively scale compute. This is why I was keen to see QRPO evaluated in an online setting. I might also highlight that LLMs are increasingly seeing adoption not just as chatbots -- where the objective is indeed to optimise the human feedback for conversational abilities -- but in agent systems with tools and multi-step rollouts for which we don't often have an offline dataset to hand.
> >
> > While I understand that the authors see this online evaluation as being out of scope and something to defer to future work, I do believe that this is a shame since QRPO no doubt holds promise in these settings. For instance, current methods such as PPO and GRPO require on-policy samples, and this slows down and adds complexity to current RL pipelines due to frequent weight synchronisation requirements with inference servers. As the authors point out, we may try to relax this on-policy requirement with importance sampling corrections, however this may increase noise.
> >
> > To comment on the questions, for Q1 I thank the authors for clarifying the conditions under which reward saturation may occur, and I agree that by adopting an iterative scheme with QRPO, one would in all likelihood be able to improve the policy beyond the optimal policy implied by the initial fixed reference.
> >
> > On Q2, I find the theoretical comparison compelling and maintain that it is a shame that the paper did not devote any experiments to this case, since indeed the ability to deviate off-policy with QRPO is one of its strong points.
> >
> > For Q3, thank you for pointing to Eq. 12 and highlighting the flexibility this affords. However once again, the deferral of any substantive use of this to later work is slightly disappointing.
> >
> > While I maintain that the paper would be improved by addressing these deferred analyses in this paper instead of future work (off policy, transformations $f$ from Eq. 12), I do find that the introduction of the quantile reward transformation and the pointwise policy fitting approach is sufficiently substantive to recommend acceptance.

---

> > > ### Author Response · Authors · 2025-08-05
> > >
> > > We thank the reviewer for engaging in the discussion and for carefully assessing our rebuttal.
> > >
> > > We certainly agree with the reviewer on the benefits of online algorithms for scaling open-ended synthetic generation to unlock new capabilities not present in offline datasets.  What we further emphasize is that **offline algorithms will play a crucial role within these online pipelines as tasks become harder and longer**: with more extended agentic tasks, the delay between data collection and policy training increases. So, as the reviewer notes, this either requires trainers to halt while waiting for online data or results in increased off-policy learning where samples ready for training at $ \pi_ t $ are generated by the policy at time $\pi_ {t-n}$. The former makes learning inefficient, and the latter makes it unstable with current importance sampling methods.
> > > **Offline algorithms are therefore essential for allowing online pipelines to scale efficiently by enabling learners to learn from delayed samples**. Moreover, offline algorithms also facilitate policy learning in multi-agent systems by enabling agents to learn from one another.
> > >
> > > In this sense, offline algorithms should **not be viewed as limited to “RLHF-style alignment work** where the objective is to optimize human feedback for conversational abilities”. Instead, they serve as enablers for scaling online pipelines.
> > >
> > > We thus acknowledge that it is a pity that we did not investigate QRPO in an online pipeline in this work, especially given the discussion we've had comparing the gradient of QRPO to policy improvement algorithms. However, we would like to emphasize that this is **not due to a reduced scope** but rather a deliberate choice to deliver a **focused and rigorous scientific study** within the constraints:
> > >
> > > - **Offline is arguably a harder learning setting than online** (as there is no online feedback for correction); thus, we chose to first demonstrate QRPO’s performance offline. (For reference, DPO was initially introduced as an offline-only algorithm.)
> > > - **Offline algorithms are not limited to RLHF-style alignment**, and we have demonstrated **QRPO’s effectiveness in technical tasks through our coding experiment**.
> > > - **“If it works offline, it is likely to work better online.”** This intuition holds for policy fitting algorithms; for instance, online DPO outperforms DPO. Our early experiments with QRPO showed that **adding online data to offline training led QRPO to a 26% greater reward improvement** compared to pure offline training. While this does not replace direct comparisons to GRPO, it provides additional evidence supporting the impact of QRPO.
> > > - Conducting a **thorough study with comprehensive evaluation protocols and extensive hyperparameter tuning is both challenging and costly**, particularly for an academic lab. We focused on rigorous scientific methodology rather than showcasing a few large wins over poorly tuned baselines. (Tuning baselines for an online setting that does not saturate is prohibitive for us, considering the number of hyperparameters and the scale of models to obtain meaningful results.)
> > > - With the **new theory, offline results, the scaling experiment** necessary for investigating quantile reward estimation noise, and the **length bias experiment** showing benefits over preference algorithms, we are already struggling to fit everything into the allowed pages, and believe the paper has a full scope worthy of being evaluated independently of the promises that QRPO brings for the future. We will add the valuable discussions we had with the reviewer to facilitate and accelerate the future impact of QRPO.
> > >
> > > In conclusion, while we agree that the paper could be further improved with a broader scope, **we hope to have demonstrated that the current scope is of high quality and offers contributions substantial enough for acceptance beyond borderline at the conference**. We warmly welcome any further feedback or recommendations to enhance our work.

---

### Official Review · Reviewer_ipQ5 · 2025-07-01

**Clarity:** 3
**Significance:** 3
**Originality:** 3
**Rating:** 5
**Confidence:** 4

**Summary:**

This paper introduces Quantile Reward Policy Optimization (QRPO), a novel approach for offline alignment optimization. The key challenge it addresses is the estimation of the partition function in KL-regularized RL objectives. To circumvent this issue, QRPO leverages a quantile reward transformation, eliminating the need for explicit integration. Empirical results on 8B-scale models demonstrate its effectiveness in both chatting and code generation tasks.

**Questions:**

- Equation (9) – Uniform Partition Function?

The partition function in Equation (9) becomes uniform across prompts (independent of *x*), which seems counterintuitive—given that reward distributions vary significantly by prompt, a fixed baseline could lead to gradient imbalance during optimization. Without prompt-dependent normalization, does this formulation fail to properly stabilize training?

- Equation (11) – Necessity of the Proposed Objective?

   - A simpler baseline could minimize MSE: $(r(x, y) - \log \pi_{\theta}(y|x))^2$, leading to a standard RL objective.

   - Why is QRPO’s design necessary? What advantages does it provide over this simpler approach?

- Evaluation & Performance Claims (Table 3)

    - The results appear noisy—sometimes DPO outperforms QRPO.

   - Can you provide a quantitative analysis of QRPO’s SOTA performance and the model behaviors? For example,

       -  Under what conditions does QRPO differ significantly from other methods?

       - What specific behaviors in the responses lead to its improvements?

- Code Generation & Mathematical Reasoning

   - For code generation: What specific problems does QRPO solve that other methods cannot?

  -  For mathematical reasoning tasks:

     - Have you evaluated QRPO on such tasks?

      - Since KL regularization is often unnecessary here, does QRPO still provide benefits?

**Ethical Concerns:**

["NO or VERY MINOR ethics concerns only"]

**Final Justification:**

The authors have addressed my concerns and the method seems to provide a new formulation for learning the optimal policy.

**Limitations:**

Yes

**Quality:**

3

**Strengths And Weaknesses:**

**Advantages**
- The paper is well-written and easy to follow in most parts.

- The empirical results show promising performance improvements.

**Disadvantages**

- Lack of Motivation & Baseline Comparison

   - There is insufficient justification for why Equation (6) must be explored in an offline setting.

   - Direct offline optimization methods (e.g., policy gradient with KL regularization) could be viable alternatives—why is QRPO necessary?

*Theoretical Gaps*

- The approximation of the partition function introduces bias—does the proposed objective still converge to the optimal policy?

- A theoretical analysis of the approximation error and convergence properties is missing.

---

> ### Author Rebuttal · Authors · 2025-07-30
>
> We thank the reviewer for their time and effort in providing detailed feedback on our submission.
>
> Below, we address each point raised:
> ___
>
> ## Motivation
>
> We specifically focused on the offline regime because we believe that a major reason for the cost inefficiency of RL post-training is the **inability to scale by leveraging diverse offline data** (i.e., previously collected online samples or offline datasets). This is currently one of the main bottlenecks in RL post-training pipelines. For example, recent large-scale RL post-training efforts (e.g., Grok-4) required compute budgets **comparable to pre-training**, largely due to the sequential and expensive nature of online data generation, making such pipelines difficult to scale.
>
> To bridge this gap, there is a strong need for an algorithm that can effectively leverage offline data, while also:
> - Operating directly on pointwise absolute rewards, like the SoTA policy gradient methods, and
> - Maintaining a simple and stable optimization objective (not requiring costly partition function estimation or auxiliary model-based approximations)
> ___
> ## Why QRPO is Needed
>
> 1. The policy improvement algorithms (policy gradient, PPO, GRPO) are the main reason for this online inefficiency and cannot use offline data. It is well known that applying PPO-like methods with offline data leads to divergence.
> 2. The current _policy fitting_ (DPO, SimPO, REBEL) algorithms can use offline data, but do not meet the requirements above. They optimize a relative reward signal via pairwise comparisons, which is not a good replacement for pointwise rewards as mentioned in our background section, due to suboptimal updates, probability mass drifts, and limited relative signal with verifiable reward settings.
>
> To the best of our knowledge, QRPO is the first offline method combining pointwise absolute rewards and stable optimization.
> ___
> ## Theoretical Aspects
>
> ### **No approximation (hence no bias) in** $Z_q(x)$
>
> In QRPO, the partition function $Z_q(x)$ is computed **analytically and exactly** due to the quantile reward transformation. Thus, QRPO introduces **no approximation or bias**, ensuring a stable and principled optimization without extra modeling overhead.
>
> ### **The only stochastic element is the empirical quantile reward** ${\mathcal R}_q$
>
> For each prompt $x$ we estimate
>
> $$
> \hat{\mathcal R}_ q(x,y)=\frac{1}{n}\sum_ {i=1}^{n} \mathbf 1 \bigl\\{ \mathcal R(x,y_i)\le \mathcal R(x,y) \bigr\\},
> \qquad y_ i \sim \pi_ {\text{ref}}(\cdot|x).
> $$
>
> * **Unbiasedness.**
>   $\mathbb E[\hat{\mathcal R}_q(x,y)|x,y]=\mathcal R_q(x,y)$.
>
> * **Consistency and rate.**
>   By the Glivenko–Cantelli theorem the empirical CDF converges uniformly:
>
>   $$
>   \sup_r\bigl|\hat F_{n}(x,r)-F_{\text{ref}}(x,r)\bigr| \xrightarrow[n\to\infty]{a.s.} 0.
>   $$
>
>   The Dvoretzky–Kiefer–Wolfowitz inequality gives a finite-sample bound
>   $ \Pr \bigl(|\hat{\mathcal R}_q-\mathcal R_q|>\varepsilon\bigr)\le 2e^{-2n\varepsilon^2}$.
>
> Hence, the only error term we add is **zero-mean “label noise’’ that shrinks with $n$.**
>
> ### **Unbiased quantile noise keeps the optimum intact**
>
> The empirical quantile adds only zero‑mean, additive noise to the true target.
> In an MSE objective, such noise contributes a constant variance term, leaving the arg‑min, and thus the desired policy, unchanged (standard least‑squares result).
> Because gradients remain unbiased with finite variance, SGD with the usual diminishing step‑size still converges to that same optimum.
>
> We will include this clarification in the paper to be more explicit about the QRPO convergence properties.
> ___
> ## Answers to the questions
>
> ### **Q1**
>
> The partition function is indeed independent of the prompt; however, the normalization **is prompt-dependent**.
> The normalization in QRPO comes from the quantile transformation that recenters rewards with respect to the performance of the reference policy for **each prompt** separately, allowing $Z$ to be prompt-independent.
> ___
> ### **Q2**
>
> We interpret the reviewer’s proposed baseline as $(r(x,y) - \beta \log(\frac{p_ \theta (x \mid y)}{p_ {ref} (x \mid y)}))^2$, since otherwise no KL regularization term would be present.
>
> This loss can recover the correct **policy gradient** in an **online setting**, however, in the **offline setting**, it leads to biased solutions.
> For example, consider a toy environment with:
>
> * Actions $a_1, a_2$ with rewards $r_1 = 1, r_2 = 2$
> * Reference policy probabilities $\pi_\text{ref}(a_1)=0.8, \pi_\text{ref}(a_2)=0.2$
> * Offline dataset uniformly sampling both actions.
>
> Optimizing Equation (1) yields the true optimum:
>
> $$
> p^ * (a_1)=0.595,\quad p^ * (a_2)=0.405
> $$
>
> while different squared-loss formulations give:
>
> | Objective                                                           | $p(a_1)$    | $p(a_2)$    |
> | - | - | - |
> | $(r - \log p)^2$ *(initial loss)*                                   | 0.422     | 0.578     |
> | $(r - \beta \log(p/p_\text{ref}))^2$ *(likely intended variant)*    | 0.542     | 0.458     |
> | **QRPO Loss** | **0.595** | **0.405** |
> ___
> ### **Q3**
>
> **Results Interpretation**
>
> **1. Metrics Priority**
>
> * Our primary evaluation metric is **AlpacaEval 2**, as it measures downstream performance on a dataset of unseen prompts and uses GPT-4 as a judge, providing a more reliable proxy for human preferences than the training reward model.
> * Within AlpacaEval 2, we emphasize the **Length-Controlled (LC) score**, shown to have the highest correlation with human judgments (Dubois et al., 2024) and to mitigate length bias that many algorithms exploit.
> * ArmoRM rewards (measured on a test set) are generally correlated with LC-AlpacaEval2 but we've observed that they are sometimes **hacked** by trained policies in a non-trivial way. While we manually reviewed outputs to filter obvious hacks, reward scores alone are less reliable.
> * For ArmoRM rewards, we similarly value the LC (Length-Controlled) score more since it mitigates length bias
>
> **2. Significance of Improvements**
>
> * On Llama 8B SFT, QRPO consistently outperforms all baselines. On the high-quality Magpie-Air dataset, the LC-AlpacaEval2 gap between QRPO and SimPO is of the same magnitude as the well-established DPO→SimPO gap, which is widely regarded as a substantial improvement in general chat tasks, confirming that QRPO delivers a significant gain.
> * On the Mistral Instruct model, we likely observe capacity saturation. Despite extensive hyperparameter tuning (same budget across methods), DPO, SimPO, and QRPO achieve **nearly identical scores** on Ultrafeedback, with differences within the noise level. Importantly, our results for DPO and SimPO significantly surpass those originally reported in the SimPO paper (Meng et. al., 2024), indicating we have pushed all methods considerably further than previous studies. Thus, the observed saturation likely reflects that the limits of Mistral Instruct’s performance were reached, rather than a limitation specific to QRPO.
> * On Magpie-Air with Mistral, similar saturation effects appear, with DPO and SimPO exploiting length bias for slight (non-significant) AlpacaEval gains. LC-AlpacaEval scores for QRPO remain higher, consistent with its improved robustness to verbosity bias (Figure 4).
>
> **Qualitative Analysis**
>
> To robustly identify the specific behaviors that contribute to QRPO’s improvements, a thorough qualitative analysis (e.g., an extensive blinded case study with a large sample of examples) would be required. Such an analysis is beyond the scope of this work, where we focused on quantitative metrics that are known to correlate well with human preferences.  Additionally, since we exhaustively tuned all algorithms, observed differences in responses are subtle and difficult to characterize qualitatively without careful and systematic examination. We would appreciate it if the reviewer could clarify whether we correctly understood this question.
> ___
> ### **Q4**
> **Qualitative Analysis**
>
> Similar to what we noted for chat tasks, such characteristics cannot be robustly identified within our scope. We are open to further clarification from the reviewer.
>
> **Mathematical Reasoning and Effect of KL Penalty**
>
> 1. Role of KL Regularization
> * Prior work (Liu et al., 2025) observes that a KL penalty stabilizes training and leads to stronger solutions when the reference policy is periodically updated to recent checkpoints.
> * Gao et al. (2022) demonstrate that KL regularization primarily acts like early stopping, without shaping the training trajectory. These results support the view that periodic reference policy resets are sufficient to mitigate potential KL-related issues, making QRPO applicable to reasoning tasks as well.
>
> 2. Applicability to Reasoning Tasks
> * We have not tested QRPO datasets because 7–8B models are limited in reasoning ability, making comparisons between methods potentially uninformative.
> * We chose to evaluate QRPO on code generation instead of mathematical reasoning because this is still a highly technical domain demonstrating QRPO's ability to handle tasks requiring structured reasoning and precision, while being easier and more robust to verify, requiring less computational budget (no judge), and more suitable for 7–8B models.
>
> [1] Pal, Arka, et al. "Smaug: Fixing failure modes of preference optimisation with dpo-positive." arXiv preprint arXiv:2402.13228 (2024)
>
> [2] Liu, Mingjie, et al. "Prorl: Prolonged reinforcement learning expands reasoning boundaries in large language models." arXiv preprint arXiv:2505.24864 (2025).
>
> [3] Gao, Leo, John Schulman, and Jacob Hilton. "Scaling laws for reward model overoptimization." International Conference on Machine Learning. PMLR, 2023.

---

> > ### Comment · Reviewer_ipQ5 · 2025-08-05
> >
> > Thank you for the clarification.
> >
> > Could you clarify the toy offline example you provided? I did not fully understand how QRPO achieves the optimal solution in this case. I am confused by the fact that the optimal policy in QRPO does not seem to depend on the data collection policy or the hyperparameters. Is this true, or is there a theoretical justification for this result that I might have missed?
> >
> > Regarding the baseline, based on my experience, offline policy gradient (PG) methods/variants can work effectively if the learning rate is small and the training epochs are limited—conditions that are typically true in LLMs. For reference, please see works like [1].
> >
> > [1] Qu, Yuxiao, et al. "Recursive introspection: Teaching language model agents how to self-improve." Advances in Neural Information Processing Systems 37 (2024): 55249-55285.

---

> > > ### Author Response · Authors · 2025-08-06
> > >
> > > Thank you for engaging in the discussion! We are happy to provide further clarification.
> > >
> > > ## Clarification on the Optimal Solution
> > > In both the toy example and our paper, we aim to optimize the KL-regularized objective defined in Equation (1). The **theoretical optimum** of this objective can be computed analytically in closed form, as shown in Equation (2) (this is a known result; see the DPO paper by Rafailov et al., 2023, for an example derivation). For the toy setup, the optimal policy takes the form:
> > >
> > > $$
> > > \pi^*(a_i) = \frac{1}{Z} \pi_{ref}(a_i) \exp{\left( \frac{1}{\beta} r_i \right)}; \ \ \ \text{where} \ Z = \sum_{a_i} \pi_{ref}(a_i) \exp{\left( \frac{1}{\beta} r_i \right)}.
> > > $$
> > >
> > > QRPO minimizes a squared regression loss that directly targets this optimal solution for all actions present in the dataset; the loss is literally $[\beta\log \pi^*(a_i) - \beta\log \pi_\theta(a_i)]^2$, so it simply matches the probabilities from the model with the optimal ones. In this sense, it **is independent of the data distribution** and depends only on whether the actions appear in the data at all.
> > >
> > > In contrast, Reward Weighted Regression (RWR) and Advantage Weighted Regression (AWR) methods (such as those referenced by the reviewer) optimize a weighted supervised regression under a sampling policy (usually the expert), which is effectively equivalent to fitting a policy that **depends explicitly on the sampling distribution**: $\pi^*(a_i) = \frac{1}{Z} \mu(a_i) \exp{\left( \frac{1}{\beta} r_i \right)}$, where $\mu(a_i)$ is the sampler distribution.
> > >
> > > ___
> > > ## Clarification on the distinction between QRPO and Reward Weighted Regression methods
> > >
> > > To clarify the differences, RWR essentially minimizes the divergence $D_{KL}(\pi^* (\cdot \mid y) || \pi_\theta(\cdot \mid y))$, where $\pi^*(y \mid x) = \frac{1}{Z(x)} \mu(y \mid x) \exp{\left( \frac{1}{\beta} R(x, y) \right)}$, and $\mu(y \mid x)$ is a sampler distribution. This is equivalent to and, in practice, optimized with the following optimization objective (Peng et al., 2019, Eq. 39):
> > >
> > > $$
> > > \underset{\pi}{\operatorname{argmax}} \mathbb{E}_ {x \sim \mathcal{D}} \ \mathbb{E}_ {y \sim \mu(\cdot \mid x)} \bigg[ \log(\pi_\theta(y \mid x)) \exp{ \frac{1}{\beta} \mathcal{R}(x,y)} \bigg].
> > > $$
> > >
> > > AWR and similar methods use the same formulation but replace the reward with an advantage estimate $A(x, y) = \mathcal{R}(x,y) - V^\mu(x)$ or use any other way of centering the signal (e.g., the work mentioned by the reviewer).
> > >
> > > In fact, using the taxonomy introduced in our paper, these methods can also be described as "policy fitting" approaches, but they differ from QRPO in two key ways:
> > > - **Fitting approach**: RWR/AWR perform distribution matching via forward KL, while QRPO performs direct regression on log-probabilities.
> > > - **Sampler-dependent optimal policy**: The target policy in RWR/AWR depends on the data collection policy $\mu$ due to the distribution matching objective, whereas QRPO targets an optimal policy that depends only on $\pi_{ref}$ thanks to the regression.
> > >
> > > Essentially, RWR-like methods **do not optimize the KL-constrained objective defined in Equation (1)**. Since our work is specifically focused on this objective, a comparison would not be theoretically accurate.
> > >
> > > Furthermore, the motivation of Eq. (1) is to optimize the reward while staying close to $\pi_{ref}$. In contrast, the motivation of RWR methods is to imitate the sampler policy while weighting its actions by the rewards.
> > > This introduces several points where our alignment setting is better solved by QRPO and DPO-style methods than RWR methods. For example, QRPO and DPO can take offline examples that show the behavior not to adopt, and they will be downweighted with a negative gradient. However, in RWR, if these samples are in the dataset, it means they're part of the sampling policy and should be positively reinforced, only with a smaller reward than the others.
> > >
> > > ---
> > > Finally, if you need further clarification on the results in the toy example, and if the Area Chair finds it acceptable, we are happy to provide a short code snippet that finds the optimal policy for each loss, showing that only QRPO finds the right policy.
> > >
> > > We thank you again for engaging in the discussion. Please let us know if you need further clarification.
> > >
> > >
> > >
> > > [1] Rafailov, Rafael, et al. "Direct preference optimization: Your language model is secretly a reward model." Advances in neural information processing systems 36 (2023): 53728-53741.
> > >
> > > [2] Peters, Jan, Katharina Mulling, and Yasemin Altun. “Relative entropy policy search.” Proceedings of the AAAI Conference on Artificial Intelligence. Vol. 24. No. 1. 2010.
> > >
> > > [3] Peng, Xue Bin, et al. "Advantage-weighted regression: Simple and scalable off-policy reinforcement learning." arXiv preprint arXiv:1910.00177 (2019).

---

> > > > ### Comment · Reviewer_ipQ5 · 2025-08-07
> > > >
> > > > Thank you for the clarification. This discussion has been highly enlightening and has shifted my perspective on designing learning schemes. While I believe a more principled approach to defining terms like "divergence" and "data-collection distribution" would be fundamental—offering deeper theoretical insights—I remain uncertain about why KL-divergence is so heavily defined in terms of the learner's versus the teacher's distribution. But, this likely goes beyond the scope of the current paper. I greatly appreciate the authors' detailed response and, based on their explanation, would like to revise my recommendation accordingly.

---

### Official Review · Reviewer_A3fZ · 2025-07-05

**Clarity:** 3
**Significance:** 2
**Originality:** 3
**Rating:** 5
**Confidence:** 3

**Summary:**

This paper proposes “Quantile Reward Policy Optimization” (QRPO) -- a new algorithm / framework for optimizing the KL constrained objective of RL for LLMs, by analytically solving for the partition function under a quantile based reward.  This furthur can be used in both offline and online settings which is an advantage over strictly online methods like PPO and RLOO. The authors report results against multiple baselines on the AlpacaEval2 and UltraFeedback dataset showing the method outperforming baselines.

**Questions:**

- How sensitive is the algorithm to online vs offline data-distribution?
- Do the authors have a hypothesis about whyQRPO is more robust to length-hacking?
- I believe RLOO is left out of the reference citations eventhough it is referenced in the first figure

See weakneses for a list of additional questions / suggestions.

**Ethical Concerns:**

["NO or VERY MINOR ethics concerns only"]

**Limitations:**

yes

**Quality:**

3

**Strengths And Weaknesses:**

Strengths:
- Well motivated method and theoretically intuitive
- The paper is generally well written and clear -- especially on describing the subtle difference between widely used algorithms.

Weaknesses:
- As outlined in the limitations,  there aren't any ablations with the use of online data in training beyond Figure 3 which looks at generating data in one timestep, with a single behaviour policy. However, is it described as one of the key novelity points of the algorithm since it bridges the gap between online and offline algorithms. Thus, I think abalations giving insight into the dynamics of an online training setup would strengthen the paper.
- As transforiming the reward is one of the key insights of the paper, it could benefit from a more extensive discussion of the reward distributions experimented on both for ArmorRM under different datasets, and the LeetCode test case distribution. In practice, how Gaussian were the original rewards? and how uniform were the transformed rewards?

---

> ### Author Rebuttal · Authors · 2025-07-30
>
> We thank the reviewer for the thorough review and appreciate the detailed comments and pertinent questions.
>
> We address the raised concerns below:
>
> **W1: online ablation**
> - We had preliminary ablations in our early experiments where we added online data to offline data in QRPO training on the Magpie-Air dataset, and we observed a 26% greater reward improvement compared to pure offline training.
> - We thank the reviewer for the suggestion and will include this preliminary online–offline ablation result in the appendix to provide additional empirical support to the broader impact of QRPO.
> - For us, this was rather a consistency check and an expected behavior, as in general for policy fitting methods, online data is expected to help (Guo et. al., 2024).
> - We chose to focus on the offline regime (varying distribution shifts) as this is where the main difficulty arises: where existing policy improvement methods (PPO, etc.) completely fail and where QRPO makes its significant contribution compared to the rest of the policy fitting methods (DPO, etc.) that use pairwise data.
> - In fact, the inability to scale the RL training by leveraging offline data (offline or off-policy from previous iterations) is a key bottleneck in current RL post-training pipelines, for example, requiring the post-training of Grok-4 RL  to consume compute comparable to pre-training, largely due to costly online data sampling.
> - Hence, to "bridge the gap" between online and offline algorithms, we focused on validating the ability of QRPO to keep the characteristic of using pointwise rewards from the online-only policy improvement methods and enabling training with offline/off-policy data.
> - Our rigorous experimental protocol to evaluate QPRO for offline/off-policy regimes already required more than 50k GPU hours to complete, which is already very large for an academic lab, and we expect practitioners to expand on the preliminary online-offline ablation results that we had (which require 2 times more resources due to online sampling costs) to further highlight the online and semi-online performance of QRPO.
>
> **W2: reward distribution discussion**
> - QRPO does not rely on the specific distribution of the initial reward. The quantile reward has a uniform distribution independently of the shape of the initial reward distribution (see proof in Appendix E; there are no assumptions on the initial distribution shape).
> - We've verified that quantile rewards indeed follow a uniform distribution for all models, datasets, and tasks. We will add the corresponding plots to the appendix.
> - ArmorRM rewards for the general chat task are nearly Gaussian on both the Magpie-Air and UltraFeedback datasets. However, for some prompts, it is skewed, and this is why it is not correct to assume it to always be Gaussian and use a closed-form expression for the partition function Z(x) of an assumed distribution without applying a quantile transformation.
> - On coding dataset, since the reward is limited to the range from 0 to 1 (test case pass rate), the distribution is often non-Gaussian
>
> **Q1: sensitivity to data distribution**
> - We have discussed the online vs. offline in our answer to **W1**.
> - But what we find also interesting, and have experimented with thoroughly in the scaling ablation (Figure 3), is the sensitivity to offline vs. off-policy (that can be treated as relatively close to online):
>   - In our offline regime, data comes from a stronger model, or could be from human annotators, resulting in high-quality samples. Since these samples typically lie in the right tail of the reference model distribution (used for quantile transformation), the quantile reward estimates were less sensitive to the number of reference completions used for CDF estimation. Hence, with offline data, a small number of reference completions suffices for a good signal (we used 1 on Magpie-Air), but scaling reference completions yields limited gains.
>   - However, in the off-policy regime, however, data is generated from the reference model, so there is no distribution shift between the training data and reference model completions. This makes the noise in quantile reward estimates decrease significantly when increasing the number of reference completions. Hence, with off-policy data, scaling reference data pre-computation leads to larger performance improvements.
>
> **Q2: length bias mitigation**:
> - We hypothesize that the reduced length bias observed in QRPO (and similarly in REBEL) is primarily due to the methods being able to use the scalar ArmoRM reward signal, as this reward model was trained to penalize verbosity and avoid length exploitation.
> - Notably, while DPO and SimPO also rely on the same underlying reward model to generate preference pairs, the conversion of scalar rewards into binary preferences appears to discard part of the anti-verbosity signal, making these methods more susceptible to length-hacking.
>
> **Q3: RLOO citation**:
> Thank you for catching this oversight. We will ensure that the appropriate citation for RLOO is included in the revised version of the paper.
>
> We appreciate the reviewer’s constructive feedback, and we think their questions improved our work. We remain open to any further suggestions.
>
> [1] Guo, Shangmin, et al. "Direct language model alignment from online ai feedback." arXiv preprint arXiv:2402.04792 (2024).

---

### Official Review · Reviewer_u3X5 · 2025-07-23

**Clarity:** 4
**Significance:** 3
**Originality:** 4
**Rating:** 5
**Confidence:** 3

**Summary:**

QRPO is a novel alignment algorithm that is simple and works offline, but also allows the utilization of pointwise absolute rewards. The key idea is the use of a quantile transformation to make the partition function analytically tractable. The authors demonstrate consistent improvements over DPO, REBEL, and SimPO on both conversation and coding datasets.

**Questions:**

See weakness.

**Ethical Concerns:**

["NO or VERY MINOR ethics concerns only"]

**Limitations:**

yes

**Quality:**

4

**Strengths And Weaknesses:**

**Strengths**
- The paper is well-structured and clearly written, explaining the methodology and results effectively.
- The quantile transformation is elegant and nicely bridges the gap between policy fitting and policy improvement methods. This enables offline methods to effectively leverage absolute reward models. The proof is rigorous. There is solid empirical evidence supporting the effectiveness of QRPO as compared to the baselines.

**Weaknesses**
- Experiments focus only on 7B-8B models. Given the importance of scale in modern LLM alignment, evaluation on larger models would significantly strengthen the claims.
- QRPO's efficacy hinges on having a reasonably good reference sampler. If the reference policy is poorly calibrated, quantile estimates may misrank high‑quality outputs. A small experiment varying reference accuracy would clarify this sensitivity.

---

> ### Author Rebuttal · Authors · 2025-07-31
>
> We appreciate the reviewer’s thoughtful and encouraging review and address the raised points below.
>
> ---
>
> ### **W1: Only 7B–8B Experiments**
>
> - We agree with the reviewer that given the potential impact of QRPO, experiments with larger models would make the claims stronger and accelerate confident adoption.
> - However, providing a **thorough scientific experimental protocol** such as the one we followed to formulate our conclusions would have been **computationally prohibitive with larger-scale models**.
> - We opted to put our budget and efforts into such a protocol, which at the 8B scale is already significant and allows comparison with previous papers (notably SimPO), **rather than conduct a more limited number of experiments with larger-scale models**.
> - Our protocol includes diverse datasets and models, and appropriately tuned baselines (over 200 trained models for each baseline), and used over 50k GPU hours, which is already very high for an academic project.
>
> ---
>
> ### **W2: Calibration of the Reference Policy**
>
> - The reference policy is typically taken to be the initial training model, so the calibration can be seen with respect to the initial training checkpoint.
> - If the training data is **purely offline and strictly much better than the initial checkpoint**, then we can perform **SFT on this data first to reduce the distribution shift** and make the initial checkpoint for QRPO, thus the reference policy. more "calibrated" to allow effective training.
> - This is what we have observed in our experiments in Table 6, for example, where models trained on Magpie-Air were **more performant by having a preliminary SFT on the chosen answer** (whose quantile is around 0.91) before QRPO. (Followed by the models trained offline without the preliminary SFT, and then the SFT models—so the performance is not due to SFT alone).
> - When the **data is collected iteratively** from the trained model, this data can have **quantiles that progressively become in the top quantile** of the reference model/initial checkpoint.
> - However, this saturation coincides with **convergence to the optimal policy**. This is because the optimal policy under the quantile reward remains supported ("calibrated") within the reference distribution, due to the “best-of-N” nature of this optimal distribution (Appendix M).
> - As a result, the policy cannot easily escape the support of the reference model, and the model is unlikely to produce completions that consistently outperform all reference completions. **Hence, there is no dramatic miscalibration before convergence**.
> - We also validate this in practice; our experiments never exceeded an average quantile reward of 0.95–0.98.
> - Moreover, when gains may start to diminish, observe that the **reference policy can be iteratively replaced by the most recent/performant checkpoint during training**, with new reference completions to estimate the quantile reward, which would be better "calibrated". This is a common recipe used with policy fitting methods. For example, iterative DPO was used to train the Llama 3 models.
> - We thank the reviewer for raising this point, and we will add this discussion to the paper to make it easier for practitioners to understand how to use QRPO.
>
>
> We appreciate the reviewer’s insightful feedback and thoughtful points. We also remain receptive to any further suggestions or recommendations that may help strengthen the quality and impact of our work.

---

### Official Review · Reviewer_n4LT · 2025-07-23

**Clarity:** 4
**Significance:** 4
**Originality:** 4
**Rating:** 4
**Confidence:** 3

**Summary:**

This paper presents Quantile Reward Policy Optimization, a novel policy fitting method for fine-tuning large language models  that sidesteps a longstandong obstacle in  alignment: the intractability of the partition function in KL-regularized objectives when using absolute rewards. It introduces an elegant and computationally efficient solution by applying a quantile transformation to the reward distribution, rendering the partition function analytically tractable in closed form. Empirical results demonstrate QRPO’s superiority across chat and code generation tasks when compared to strong baselines like DPO, REBEL, and SimPO.

**Questions:**

1. Have you explored any other transformations with regards that allow the framework to a multi level preference reward signal?

**Ethical Concerns:**

["NO or VERY MINOR ethics concerns only"]

**Final Justification:**

I have gone over the replies of the reviewers. They have answered the questions to my satisfaction, and I maintain my positive score.

**Limitations:**

Due to my limited exposure to applied RL I am unable to analyze the limitations of the empirical results.

**Paper Formatting Concerns:**

None.

**Quality:**

4

**Strengths And Weaknesses:**

1. From a theoretical standpoint, the most remarkable contribution of this work is the closed-form expression for the partition function Z(x) after applying a quantile transformation to the reward. The tractability of Z(x), which typically renders absolute reward regression intractable, is a longstanding bottleneck in regularized RL. This is an elegant and rigorous contribution that simplifies the RL fine-tuning objective while preserving its core structure.

2. The authors extend the novel theoretical insight and they validate the method comprehensively with experiments on two major alignment datasets and a code generation task, showing that QRPO outperforms state-of-the-art methods under comparable settings.

Since I am from a primarily a theory background, I am unable to evaluate the robustness of the empirical results in a rigorous manner.

---

> ### Author Rebuttal · Authors · 2025-07-30
>
> We thank the reviewer for their time and for providing a thoughtful and thorough review, particularly from the theoretical perspective.
>
> ---
>
> ### Strengths and Weaknesses
>
> **Longstanding Bottleneck and Broader Impact**
>
> We appreciate the acknowledgment of our theoretical contributions, especially the closed-form expression for the partition function. As highlighted, this addresses a longstanding bottleneck in the field and was **also described as "elegant and rigorous" by Reviewer u3X5.**
>
> Additionally, Reviewer Ghhc noted the **broader potential impact**:
>
> > “The approach set out in this paper (i.e., converting an arbitrary scalar reward into a bounded, uniform signal whose partition function is a constant) could be reused elsewhere where we face energy-based or KL objectives with nuisance normalizers.”
>
>
> **Empirical Evaluation**
>
> Regarding the empirical evaluation, which the reviewer mentioned was outside their primary area of expertise, we believe the comments from other reviewers offer a helpful, complementary perspective.
>
> * Reviewer Ghhc:
>
>   > "The experimental work is careful and high-quality, and appears to follow rigorous methodology to appropriately tune baselines, re-use appropriate reward models, and report numbers with uncertainty estimates throughout."
>
> * Reviewer u3X5:
>
>   > "Solid empirical evidence supporting the effectiveness of QRPO as compared to the baselines."
>
> ---
>
> ### Questions
>
> **Multi-Level Preference Reward Signal**
>
> This is indeed an interesting opportunity. Multi-level (multi-dimensional) rewards could be integrated into QRPO using several conventional strategies (linear projection into a single scalar reward, independent training on each reward, followed by model merging, or training with a linear combination of individual reward losses).
>
> However, there is also an opportunity for a novel approach within QRPO. Specifically, we can recover the quantile of a training completion given the reference completions with a non-linear mapping that takes into account all the relative information between the completions.
> I.e., the completions would not be ranked independently, but all would be ranked relative to each other.
> For example, we can use a carefully prompted LLM-as-judge with all the reference completions together with training completions to obtain quantile rewards.
>
> We will add this discussion to the paper, which we believe will significantly benefit practitioners in effectively leveraging QRPO and inspire new directions and open questions for future research.
>
> ---
>
> We greatly value the feedback and questions from the reviewer, and we remain open to any additional ideas or recommendations that could further enhance the quality and impact of our work.

---

### Comment · Area_Chair_7xzs · 2025-08-05
**Please read the authors’ rebuttal and join the discussion**

Dear Reviewers,

Thank you for your valuable contributions.

The authors have provided rebuttal to your reviews. Please carefully read their responses  as soon as possible, and indicate whether your concerns have been addressed. You are also encouraged to engage in discussion with fellow reviewers.

Note that simply **submitting "Mandatory Acknowledgement" without posting any feedback is NOT allowed**. *Let's be the kind of reviewers we’d appreciate for our own submissions*.

Best,
AC

---

### Note · Authors · 2025-08-13

Dear Area Chairs and Reviewers,

We thank you for your time and effort, and take the opportunity to provide a summary of the final reviews from our perspective, which we hope reflects a shared consensus.

**Reviewer n4LT** gave excellent individual scores for Quality, Clarity, Significance, and Originality, and praised our contributions. We clarified their question on using QRPO with a multi-level preference reward, which they found clear. The "borderline accept" overall rating, despite the strong individual scores, remains unclear. The reviewer mentioned their limited expertise to assess the empirical results -- we believe this reflects confidence level rather than the work’s merit.

**Reviewer u3X5** praised the elegance of QRPO, the rigor of its proofs, and the strength of its empirical evidence, giving excellent scores for Quality, Clarity, and Originality. We acknowledged their point on scaling to larger models, noting that our budget prioritized a thorough 8B-scale evaluation for scientific rigor and comparability with prior work, and addressed the questions.

**Reviewer A3fZ** found QRPO well-motivated and theoretically intuitive, and highlighted the paper’s clarity. We acknowledged the request for broader online training evaluation, providing preliminary online-offline results and explaining our focus on the offline regime, and clarified the questions.

**Reviewer ipQ5** initially raised concerns about the motivation and necessity of QRPO and its theoretical properties. We provided detailed clarifications on the advantages of QRPO over existing offline methods, explained its convergence guarantees, and illustrated key points through a toy example. The reviewer described these clarifications as “highly enlightening” and noted that they had shifted their perspective on designing learning schemes, leading to a revision of their score.

**Reviewer Ghhc** noted the originality of QRPO and the high-quality experimental work, while questioning whether existing methods share its properties and requesting comparisons with online methods. We clarified the paper’s positioning, highlighting the unique and essential properties of QRPO not met by any other method, and presented a theoretical comparison with online counterparts. The reviewer acknowledged the vital role and uniqueness of QRPO in offline training and found the theoretical comparison in the online regime compelling, noting that QRPO “no doubt holds promise” in this setting. They revised their score.

---

### Decision · Program_Chairs · 2025-09-17

**Decision:**

Accept (poster)

**Comment:**

This paper presents Quantile Reward Policy Optimization (QRPO), a RL method that aim to better leverage pointwise reward signals. The main challenge is the intractable partition function, and QRPO proposes to apply a quantile transformation to obtain a tractable partition function. The derivations are clear, and the method’s motivation and distinction from related work were clarified during the author–reviewer discussion, with the authors emphasizing robustness benefits of quantile weighting and theoretical guarantees. Reviewers appreciated the solid theoretical foundation but raised concerns about the modest empirical gains and limited experimental scope, which the authors addressed with additional ablations and explanations. Quantile transformation in QRPO is insightful, although estimate the quantile itself may still require "relative" computation among rewards. Overall, I recommend accept for this paper.